# Socially meaningful visual context either enhances or inhibits vocalisation processing in the macaque brain

Mathilda Froesel [1] ✉, Maëva Gacoin[1], Simon Clavagnier[1], Marc Hauser[2], Quentin Goudard[1] & Suliann Ben Hamed [1] ✉

Social interactions rely on the interpretation of semantic and emotional information, often from multiple sensory modalities. Nonhuman primates send and receive auditory and visual communicative signals. However, the neural mechanisms underlying the association of visual and auditory information based on their common social meaning are unknown. Using heart rate estimates and functional neuroimaging, we show that in the lateral and superior temporal sulcus of the macaque monkey, neural responses are enhanced in response to species-specific vocalisations paired with a matching visual context, or when vocalisations follow, in time, visual information, but inhibited when vocalisation are incongruent with the visual context. For example, responses to affiliative vocalisations are enhanced when paired with affiliative contexts but inhibited when paired with aggressive or escape contexts. Overall, we propose that the identified neural network represents social meaning irrespective of sensory modality.

Brain structure and function have evolved in response to social relationships, both within and between groups, in all mammals. For example, across species, brain size and gyrification has been shown to increase with average social group size[1–3], as well as meta-cognitive abilities[4]. Within a given species, functional connectivity within the so-called social brain has been shown to be stronger in macaques living in larger social groups[5]. In this context, successful social interactions require the proper interpretation of social signals[6], whether visual (body postures, facial expressions, inter-individual interactions) or auditory (vocalisation).

In humans, the core language system is amodal, in the sense that our phonology, semantics and syntax function in the same way whether the input is auditory (speech) or visual (sign). In monkeys and apes, vocalisations are often associated with specific facial expressions and body postures[7]. This raises the question of whether and how auditory and visual information are integrated to interpret the meaning of a given situation, including emotional states and functional behavioural responses. For example, macaque monkeys scream as an indication of fear, triggered by potential danger from conspecifics or heterospecifics. In contrast, macaques coo during positive social interactions, involving approach, feeding and group movement[8,9]. To what extent, does hearing a scream generate a visual representation of the individual(s) involved in such an antagonistic situation, as opposed to a positive social situation? Does seeing an antagonistic situation set up an expectation that screams, but not coos, will be produced?

Face, voice, and social scene processing in monkeys have been individually explored, to some extent, from the behavioural[10–13] and neuronal points of view[14–35]. Audio–visual integration during naturalistic social stimuli has recently been shown in specific regions of the monkey face-patch system[36], the voice-patch system[37–40], as well as in the prefrontal voice area[41]. However, beyond combining sensory information, social perception also involves integrating contextual, behavioural and emotional information[42,43]. In this context, how macaque monkeys associate specific vocalisations with specific social visual scenes based on their respective meaning has scarcely been explored. Our goal is to help fill this gap.

This study used video-based heart rate monitoring and functional magnetic resonance in awake behaving monkeys to show that rhesus

[1]Institut des Sciences Cognitives Marc Jeannerod, UMR5229 CNRS Université de Lyon, 67 Boulevard Pinel, 69675 Bron Cedex, France. [2]Risk-Eraser, LLC, PO Box 376, West Falmouth, MA 02574, USA. ✉e-mail: mathilda.froesel@isc.cnrs.fr; benhamed@isc.cnrs.fr

monkeys (*Macaca mulatta*) systematically associate the meaning of a vocalisation with the meaning of a visual scene. Specifically, they associate affiliative facial expressions or social scenes with corresponding affiliative vocalisations, aggressive facial expressions or social scenes with corresponding aggressive vocalisations, and escape visual scenes with scream vocalisations. In contrast, vocalisations that are incompatible with the visual information are fully suppressed, indicating a top-down regulation over the processing of sensory input.

## Results

In the following, we investigate whether and how macaques associate visual and auditory stimuli based on their semantic content, and we characterize the neuronal bases underlying this audio–visual integration. We obtained neural and autonomic data from two macaques using functional magnetic resonance brain imaging and video-based heart rate tracking. We designed six variants of a unique task in which we systematically manipulated the general semantics or meaning of the context as specified by visual information and presented as independent runs in the sessions. Each context, and so each independent run, combined visual stimuli of identical social content with either semantically congruent or incongruent monkey vocalisations presented together with the visual stimuli or not. The semantic context was set by the social content of the visual stimuli presented within a given variant of the task. As a result, auditory stimuli could be readily identified as congruent or incongruent with the context defined by the visual stimuli even when presented alone. On each block of trials, the

monkeys could be exposed to either visual stimuli only (Vi), auditory congruent stimuli only (AC), auditory incongruent stimuli only (AI), audio-visual congruent stimuli (VAC) or audio-visual incongruent stimuli (VAI), in a block design (Fig. 1A). Importantly, paired contexts shared the same auditory stimuli, but opposite social visual content (Fig. 1B), thus opposite semantic content and meaning. All contexts were presented randomly in independent runs and at least once during each scanning session. We report group fMRI and group heart-rate analyses. All reported statistics are based on non-parametric tests.

### Auditory whole brain activations depend on semantic congruence with visual context

Combining the F+ and F− face contexts (Fig. 2, see Supplementary Fig. 1A for individual monkey maps and Supplementary Fig. 2), which include faces expressing lipsmacks or aggressive threats, we find in the visual contrast, robust bilateral activation ($p < 0.05$ FWE) in the extrastriate cortex, along the superior temporal sulcus (STS) as well as in the prefrontal cortex, as expected from previous studies[17,24,44]. Activations were also observed in the posterior part of the fundus of the intraparietal sulcus at an uncorrected level ($p < 0.001$). Supplementary Fig. 3 represents these activation patterns overlaid with the CIVM non-human primate atlas parcellation and corresponding percentage signal change (%SC) for each area described in Supplementary Table 1 for the visual, auditory congruent and auditory incongruent vs. fixation contrasts. Please note that receiving coils were placed so as to optimize temporal and prefrontal cortex signal-to-noise ratio (SNR). As a result,

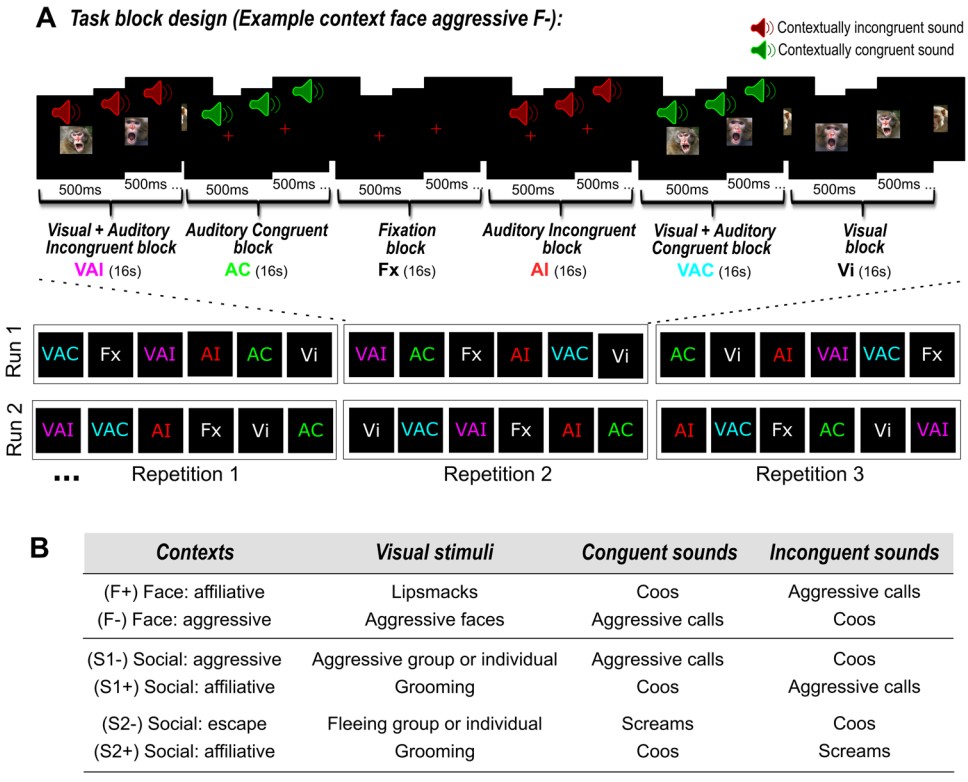

**A** *Task block design (Example context face aggressive F-):*

🔊)) Contextually incongruent sound
🔊)) Contextually congruent sound

| Visual + Auditory Incongruent block **VAI** (16s) | Auditory Congruent block **AC** (16s) | Fixation block **Fx** (16s) | Auditory Incongruent block **AI** (16s) | Visual + Auditory Congruent block **VAC** (16s) | Visual block **Vi** (16s) |

Run 1: VAC Fx VAI AI AC Vi | VAI AC Fx AI VAC Vi | AC Vi AI VAI VAC Fx

Run 2: VAI VAC AI Fx Vi AC | Vi VAC VAI Fx AI AC | AI VAC Fx AC Vi VAI

... Repetition 1          Repetition 2          Repetition 3

**B**

| Contexts | Visual stimuli | Congruent sounds | Incongruent sounds |
|---|---|---|---|
| (F+) Face: affiliative | Lipsmacks | Coos | Aggressive calls |
| (F−) Face: aggressive | Aggressive faces | Aggressive calls | Coos |
| (S1−) Social: aggressive | Aggressive group or individual | Aggressive calls | Coos |
| (S1+) Social: affiliative | Grooming | Coos | Aggressive calls |
| (S2−) Social: escape | Fleeing group or individual | Screams | Coos |
| (S2+) Social: affiliative | Grooming | Coos | Screams |

**Fig. 1 | Description of experimental design and the six different contexts used in the study. A** Experimental design. Example of an aggressive face (F−) context. Each run was composed of three randomized repetitions of six different blocks of 16 s. The six blocks could be either visual stimuli only (Vi), auditory congruent stimuli only (AC), auditory incongruent stimuli only (AI), audio-visual congruent stimuli (VAC) or audio-visual incongruent stimuli (VAI), or fixation with no sensory stimulation (Fx). Block presentation was pseudo-randomized and counterbalanced so that, across all repetitions and all runs of given context, each block was, on average, preceded by the same number of blocks from the other conditions. Initial blocks were either a visual block (Vi, VAC, VAI), or a fixation block followed by

a visual block (Vi, VAC or VAI), such that context was set by visual information early on in each run. Each sensory stimulation block contained a rapid succession of 500 ms stimuli. Each run started and ended with 10 seconds of fixation.
**B** Description of contexts. Six different contexts were used. Each context combined visual stimuli of identical social content with either semantically congruent or incongruent monkey vocalisations. Pairs of contexts shared the same auditory stimuli, but opposite social visual content (F+ vs. F−; S1+ vs. S1−; S2+ vs. S2−). Each run corresponded to one of the semantic contexts described above. Visual stimuli were extracted from videos collected by the Ben Hamed lab, as well as by Marc Hauser on Cayo Santiago, Puerto Rico.

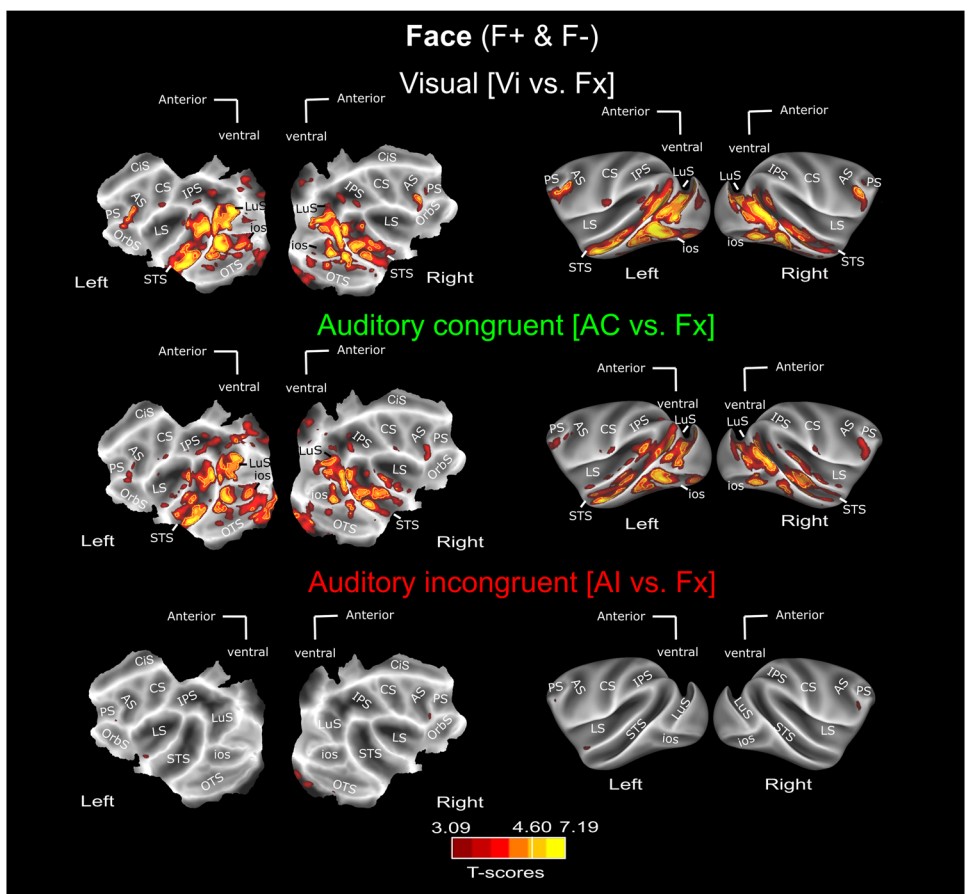

**Fig. 2 | Whole-brain activation FACE contexts (F+ & F−): main contrasts.** Whole-brain activation maps of the F+ (face affiliative) and F− (face aggressive) runs, cumulated over both monkeys, for the visual (white, Vi vs. Fx), auditory congruent (green, AC vs. Fx) and auditory incongruent (red, AI vs. Fx). Note that the AC and AI conditions contain exactly the same sound samples (coos and aggressive calls). Darker shades of red indicate level of significance at $p < 0.001$ uncorrected, $t$-score > 3.09. Lighter shades of yellow and brown outlines indicate level of significance at $p < 0.05$ FWE correction, $t$-score > 4.6, DF [1, 5200]. ios Inferior Occipital Sulcus, LS Lateral Sulcus, STS Superior Temporal Sulcus, CiS Cingulate Sulcus, LuS Lunate Sulcus, IPS Intraparietal Sulcus, PS Precentral Sulcus, CS Central Sulcus, AS Arcuate Sulcus, OrbS Orbital Sulcus. See Supplementary Fig. S1 for individual monkey data. Corresponding size effects are presented is Supplementary Figs. S2, S6 and main Fig. 6.

no activations can be seen in the occipital cortex (see temporal SNR maps in Supplementary Fig. 4 and precise mean and std signal evaluation in occipital cortex and STS; Please note that in spite of these lower SNR in the occipital cortex, %SC based on an atlas defined ROIs are occasionally significant for the Visual vs. Fixation contrast, and (less so) for the Auditory congruent vs. Fixation contrast, in V1, V2, V3 and V4: see Supplementary Tables 1–3). The congruent auditory versus fixation contrast, which combined aggressive calls and coos from the two different contexts, leads to activation within the inferior bank of the lateral sulcus, both at corrected ($p < 0.05$ FWE) and uncorrected levels ($p < 0.0001$), as described in previous studies[22,26,29]. Importantly, this contrast also leads to the same robust bilateral activations as the visual contrast: the extra-striate cortex, along the superior temporal sulcus (STS) ($p < 0.05$ FWE), as well as in the prefrontal and intraparietal cortex ($p < 0.0001$ uncorrected). Percent signal change at local peak activations in the lateral sulcus and superior temporal sulcus are presented in Supplementary Fig. 2. Supplementary Fig. 5 (left) represents the distribution of AC − AI/AC + AI (Supplementary Fig. 5A) and AC − V/AC + V (Supplementary Fig. 5B) modulation indexes across ROIs, thus precisely quantifying the effect strength. These activations are significantly higher than those observed for the incongruent vocalisations, whether the congruent auditory stimuli are coos (Fig. 3B, Supplementary Figs. 2, 6 for the effect strengths of the t-score maps) or aggressive calls (Fig. 3C), although congruent coos led to significantly higher activations than congruent aggressive calls (Supplementary

Fig. 6). Supplementary Fig. 7 represents the activation patterns of Fig. 3 overlaid with the CIVM non-human primate atlas parcellation and corresponding percentage signal change (%SC) for each area are described in Supplementary Table 2 for the visual, auditory congruent and auditory incongruent vs. fixation contrasts. In contrast, when we present the exact same aggressive calls and coos, the incongruent auditory versus fixation contrast leads to minimal activation, if any (Fig. 2, see Supplementary Fig. 1 for individual monkey data and Supplementary Figs. 2, 6 for the effect strengths of the t-score maps). Again, this doesn't depend on whether the incongruent sounds are aggressive calls (Fig. 3D) or coos (Fig. 3E). This pattern of activation therefore confirms that auditory activation does not depend on the acoustic morphology or function of the vocalisation. Rather, it depends on whether the vocalisations are congruent or not to the semantic content of the visual stimuli.

These observations are reproduced in a different set of contexts, in which the visual stimuli involve social scenes (grooming, aggression or escape) with either semantically congruent or incongruent vocalisations (Fig. 4 for all social contexts on group data, see Supplementary Fig. 1B for individual monkey data, Supplementary Fig. 8 for S+ and S− group data social contexts presented independently, and Supplementary Figs. 9, 10 for effect strengths in representative ROIs of the t-score map). Supplementary Fig. 11 represents these activation patterns overlaid with the CIVM non-human primate atlas parcellation; corresponding percentage signal change (%SC) for each area is

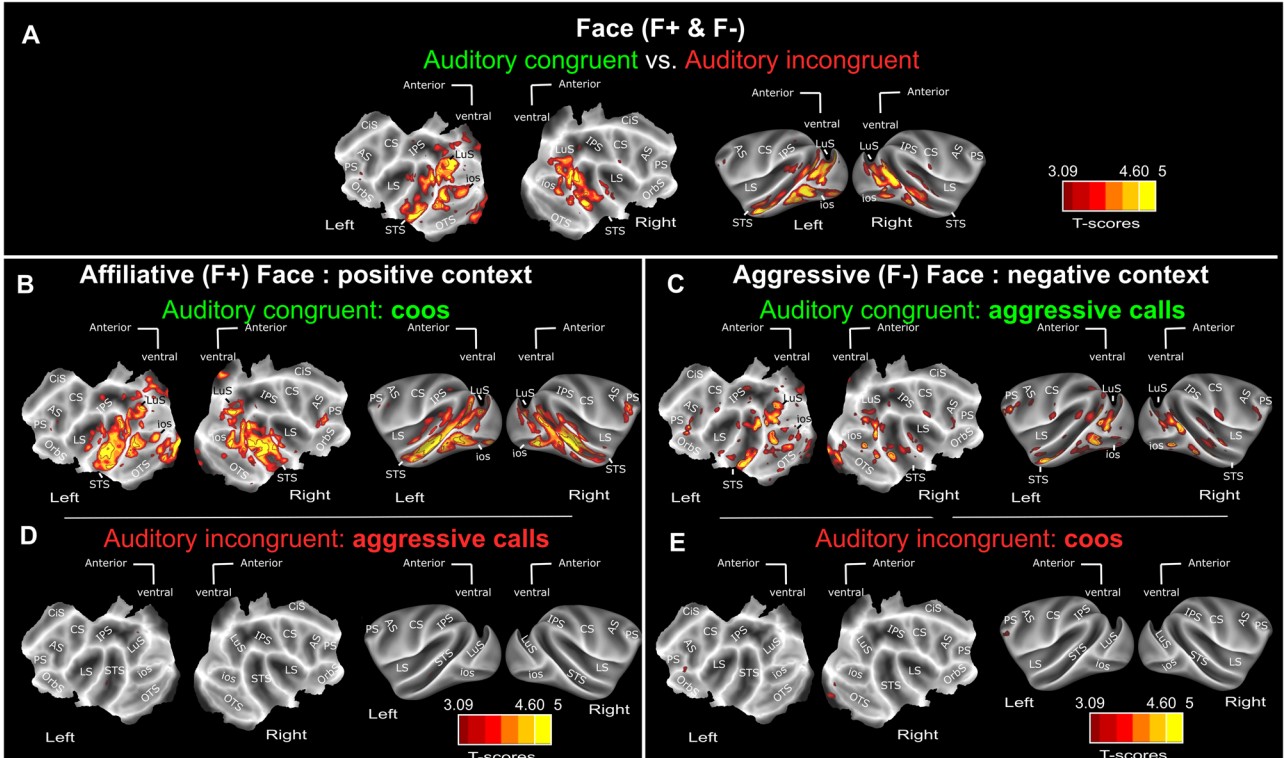

**Fig. 3 | Auditory activations depend on semantic congruence with visual context. A** Whole-brain activation maps of the F+ (face affiliative) and F− (face aggressive) runs, for the auditory congruent vs auditory incongruent (relative to the visual context) contrast. Whole-brain activation map for the F+ (face affiliative) (**B**) auditory congruent (coos, dark green, AC vs. Fx) and (**D**) auditory incongruent (aggressive calls, dark red, AI vs. Fx) conditions. Whole-brain activation map for the F− (face aggressive) (**C**) auditory congruent (aggressive calls, green, AC vs. Fx) and (**E**) auditory incongruent (coos, red, AI vs. Fx) conditions. Darker shades of red indicate level of significance at $p < 0.001$ uncorrected, $t$-score > 3.09. Lighter shades of yellow and brown outlines indicate level of significance at $p < 0.05$ FWE correction, t-score > 4.6, DF = [1, 2604] for F+ and F− DF = [1, 2618] and DF [1, 5200] for Face (F+ & F−). ios Inferior Occipital Sulcus, LS Lateral Sulcus, STS Superior Temporal Sulcus, CiS Cingulate Sulcus, LuS Lunate Sulcus, IPS Intraparietal Sulcus, PS Precentral Sulcus, CS Central Sulcus, AS Arcuate Sulcus, OrbS Orbital Sulcus.

described in Supplementary Table 3 for the visual, auditory congruent and auditory incongruent vs. fixation contrasts. To further quantify the effect strength of congruency on auditory processing, we computed an AC − AI/AC + AI modulation index (Supplementary Fig. 5A) for both face and social contexts. In both lateral and superior temporal sulci and both types of contexts, this index reveals a significantly higher activation for auditory congruent vocalisation than auditory incongruent stimuli. It is worth noting that, in 66% of the instances, both AI and AC conditions are preceded by blocks involving visual stimulation (Vi, VAC and VAI). Because this was the case for both AI and AC conditions, the absence of auditory activations in the AI vs. Fx contrast and the presence of temporal and occipital activations in the AC vs. Fx contrast cannot be interpreted as a trace of the activations resulting from the previous blocks. Instead, this pattern of responses should be considered as a process that results from the structure of the task. Indeed, the AC − AI/AC + AI modulation index progressively grows stronger within any given run, as visual stimulation reinforces context-related information. This supports the idea that the observed enhancement of AC relative to AI is context-dependent (Fig. 5A). In addition, this modulation index is not significantly different whether the auditory stimuli were presented right after a block containing visual information or separated in time from it (Fig. 5B).

Taken together, these results indicate that audio-visual semantic associations are implemented in a specific cortical network involved in the processing of both visual face and social stimuli as well as voice stimuli. This network is composed of prefrontal and temporal areas, but also, of visual striate and extrastriate visual areas (see Supplementary note attached to Supplementary Tables 1–3. An important question is thus whether these neuronal computations impact the

behaviour or the physiology of the monkeys. In the following section, we investigate how heart rate changes in response to auditory-visual stimuli that are either congruent or incongruent with the social situation.

## Heart rate variations depend on semantic congruence with visual context

In this study, monkeys were required to fixate the centre of the screen while the different auditory and visual stimuli were presented. As a result, it was not possible to analyse whether gaze is spontaneously affected by the different stimulus categories. It was, however, possible to analyse heart-rate variation using a video-based method developed by our team[45]. Figure 6 focuses on heart rate variation in response to the auditory sound categories in the different contexts. Heart rate responses, described in Fig. 6 of Froesel et al. 2020, are typically slow to build up (several seconds). As a result, quantifications of heart rate information were carried out in the second half of the block (last 8 s).

We observe a main context effect on heart rate measures (Fig. 6A, Friedman non-parametric test, $X_{2(253)} = 437.8$, $p < 0.001$), such that overall heart rate (HR) varies in response to a specific sound, as a function of the type of run being used. Differences in HR are observed between face runs and the two types of social runs, most probably due to the identity of the visual and auditory stimuli, and how they are processed by the monkeys. While this pattern is interesting, we focus here on the observed differences in HR between the positive and negative contexts of runs involving identical stimuli. For the paired contexts (F+ /F− and S1+/S1) both types of sounds (i.e. coos and aggressive calls) are associated with higher heart rate in the positive contexts than in the negative contexts (Wilcoxon paired non-

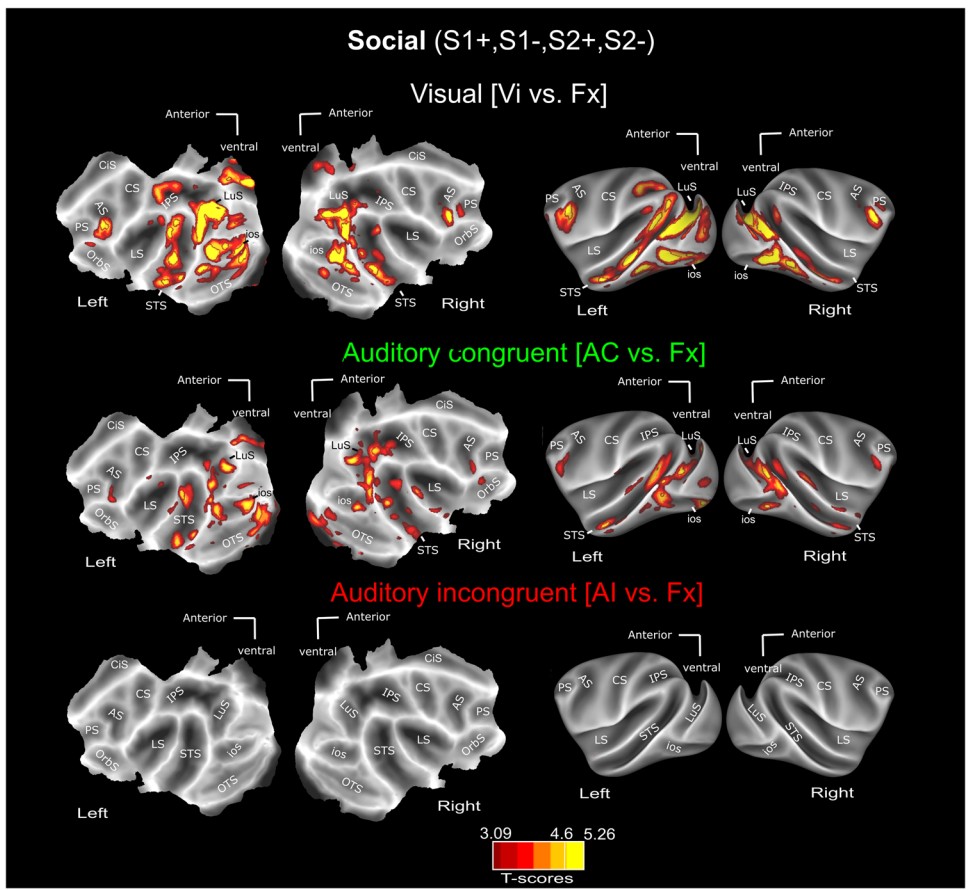

**Fig. 4 | Whole-brain activation Social contexts (S1+, S1−, S2+ & S2−): main contrasts.** Whole-brain activation maps of the S1+, S2+ (social affiliative 1 & 2), S1− (social aggressive) and S2− (social escape) runs, cumulated over both monkeys, for the visual (white, Vi vs. Fx), auditory congruent (green, AC vs. Fx) and auditory incongruent (red, AI vs. Fx). Note that the AC and AI conditions contain exactly the same sound samples (coos, aggressive calls and screams). Darker shades of red indicate level of significance at $p < 0.001$ uncorrected, $t$-score 3.09. Lighter shades of yellow and brown outlines indicate level of significance at $p < 0.05$ FWE correction, $t$-score > 4.6, DF = [1, 10344]. ios Inferior Occipital Sulcus, LS Lateral Sulcus, STS Superior Temporal Sulcus, CiS Cingulate Sulcus, IPS Intraparietal Sulcus, PS Precentral Sulcus, CS Central Sulcus, AS Arcuate Sulcus, LuS Lunate Sulcus, OrbS Orbital Sulcus.

parametric test, aggressive calls between the F+ and F− contexts: $Z = 13.77$, $p < 0.001$, Cohen's d: 16.3 and S1+ and S1−: $Z = 13.82$, $p < 0.001$, Cohen's d: 91.6; Coos between the F+ and F−: $Z = 13.87$, $p < 0.001$, Cohen's d: 47.05, S1+ and S1−: $Z = 13.78$, $p < 0.001$ Cohen's d: 17.42). Screams are also associated with higher heart rate in the positive context than in the negative context (S2+/S2−: Wilcoxon paired non-parametric test, $Z = 13.77$, $p < 0.001$ Cohen's d: 4.01). A reverse effect is observable for coos in the negative context containing screams (S2+/S2−), i.e. heart rate is higher in the negative context than in the positive context (Wilcoxon paired non-parametric test, $Z = 13.78$, $p < 0.001$, Cohen's d: 5.987). Although heart rate measures vary from one context to the other, in all contexts, congruent auditory stimuli (Fig. 6A, green) is systematically associated with lower heart rates than incongruent auditory stimuli (Fig. 6A, red, Friedman non-parametric test, Face: $X_{2(253)} = 271.442$, $p < 0.001$; Social 1:, $X_{2(253)} = 295.34$, $p < 0.001$; Social 2:, $X_{2(253)} = 174.66$, $p < 0.001$, Wilcoxon paired non-parametric test: F+: $Z = 13.98$, $p < 0.001$, Cohen's d: 4.5, F−: $Z = 9.77$, $p = 0.012$, Cohen's d: 3.9; S1+: $Z = 13.76$, $p < 0.001$, Cohen's d 19.7, S1−: $Z = 13.72$, $p < 0.001$, Cohen's d 18.66, S2+: $Z = 13.82$, $p < 0.001$, Cohen's d: 8.1, S2−: $Z = 13.77$, $p < 0.001$, Cohen's d: 2.92). This effect is more pronounced for the social contexts (S1+/S1− and S2+/S2−) than for the face contexts (Fig. 6B, F+/F−, Wilcoxon, $F = 17.45$, $p < 0.001$, Cohen's d: 1.81). This suggests an intrinsic difference between the processing of faces and social scenes. This effect is also more pronounced for contexts involving affiliative visual stimuli (F+, S1+ and S2+) than for contexts involving aggressive or escape visual stimuli (Fig. 6B. F−, S1−

and S2−, Wilcoxon non-parametric test, $F = 13.20$, $p < 0.001$, Cohen's d: 1.73). This latter interaction possibly reflects an additive effect between the semantics and emotional valence of the stimuli. Indeed, affiliative auditory stimuli are reported to decrease heart rate relative to aggressive or alarm stimuli[46]. As a result, emotionally positive stimuli would enhance the semantic congruence effect, while emotionally negative stimuli would suppress the semantic congruence effect. Overall, these observations indicate that semantic congruence is perceptually salient, at least implicitly. Importantly, the temporal dynamics of heart rate changes appear to mirror hemodynamic signal modulation in the identified functional network. Because changes in heart rate might affect measured fMRI responses[47], we re-ran the analyses presented in Figs. 2–4 using heart rate as a regressor of non-interest in addition to head motion and eye position (Supplementary Fig. 12). Observed activations remained unchanged, thus indicating that the reported activations are not an artefact of changes in heart rate. In order to further estimate the degree of coupling between heart rate and brain activations, we run a GLM using heart rate as a regressor of interest. No activations could be observed including at uncorrected levels.

**Visual auditory gradients across the lateral sulcus (LS) and superior temporal sulcus (STS)**

While LS demonstrates stronger activation for socially congruent auditory stimuli relative to visual stimuli, the STS appears to be equally activated by both sensory modalities. To better quantify this effect, we

## Evolution of AC-AI/AC+AI modulation index as a function of

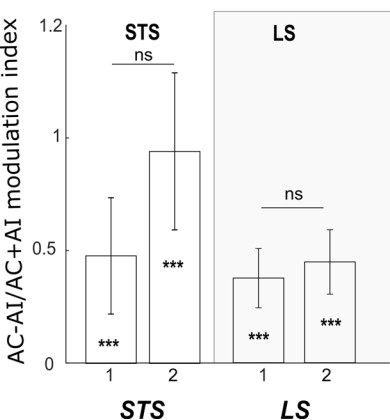

**A Repetition in run**

**B Distance from last visual block**

**Fig. 5 | Distribution of AC-AI/AC+AI modulation index as a function of repetition in run and distance from last visual block.** Distribution of modulation index of percentage signal change (%SC) for the AC condition relative to fixation baseline compared to the AI condition relative to fixation baseline (AC − AI/AC + AI), as a function of repetition order in the run (**A**) or as a function of the distance from the last visual block (**B**), for each of the STS and LS, and each of the face and social runs, computed on individual ROIs across all runs. In (**A**), 1: first occurrence of AC or AI, 2: second occurrence, 3: third occurrence. In (**B**), 1: AC or AI just following a block with visual stimuli presentations, 2: AC or AI presented two blocks away from a block with visual stimuli presentations. Statistical differences relative to baseline or across conditions are indicated as follows: ***, $p < 0.001$; **, $p < 0.01$; *, $p < 0.05$; n.s., $p > 0.05$ (Wilcoxon two-sided non-parametric test: (**A**) STS 1: $n = 14$, $Z = 3.21$, $p = 1.6e\text{-}06$; 2: $n = 14$, $Z = 3.41$, $p = 6.4e\text{-}04$; 3: $n = 14$, $Z = 4.78$, $p = 1.7e\text{-}06$; 1–2: $Z = 1.58$, p = 0.11; 1–3: $Z = 4.16$, $p = 0.003$; 2–3: $Z = 1.81$, $p = 0.06$. LS: 1: $n = 10$, $Z = 1.57$, $p = 0.11$; 2: $n = 10$, $Z = 2.38$, $p = 0.02$; 3: $n = 10$, $Z = 4.38$, $p = 0.01$; 1–2: $Z = 1.77$, $p = 0.07$; 1–3: $Z = 2.3$, $p = 0.02$; 2–3: $Z = 0.86$, $p = 0.38$. **B** STS: 1: $n = 196$, $Z = 3.26$, $p = 6.7e\text{-}12$; 2: $n = 196$, $Z = 3.62$, $p = 6.9e\text{-}12$; 1–2: $Z = 1.58$, $p = 0.19$; LS: 1: $Z = 3.18$, $p = 5.1e\text{-}12$; 2: $Z = 3.28$, $p = 6.7e\text{-}10$; 1–2: $Z = 0.05$, $p = 0.8$). Data are presented as median ± s.e.

define regions of interest (ROIs, 1.5 mm spheres) at local peak activations in the auditory congruent (AC vs Fx) contrast, in the face contexts (Fig. 7A, see Supplementary Fig. 13 for a precise localization of each of these local maxima on corresponding brain anatomy). These peaks match peak activations in the social contexts auditory congruent (AC vs Fx) contrast. This latter social context contrast reveals two additional peaks in the right LS which were used to define two additional ROIs (right LS4 and LS6). Overall, 8 ROIs are thus defined in the right STS, 6 in the left STS, 4 in the left LS and 6 in the right LS. The numbering of these ROIs was adjusted so as to match mirror positions across hemispheres. Figure 7B presents median percentage signal change (%SC) for each independent ROI, in the left and right hemispheres, on each of the face and social contexts. Overall, STS ROIs and LS ROIS had similar %SC profiles across the face and social contexts (LS: FACE $_{(9,320)}$ = 0.585, $p = 0.867$; SOCIAL $F_{(9,702)}$ = 1.008, $p = 0.432$ and STS: FACE $F_{(13, 507)}$ = 1.283, $p = 0.225$; SOCIAL $F_{(13,1014)}$ = 1.629; $p = 0.078$). No interhemispheric difference could be noted (LS: FACE $F_{(1,40)}$ = 0.136; $p = 0.714$; SOCIAL: $F_{(1,78)}$ = 0.727; $p = 0.396$ and STS: FACE $F_{(1, 40)}$ = 0.014; $p = 0.906$; SOCIAL: $F_{(1,78)}$ = 0.544; $p = 0.463$). Note that these observations are preserved when ROIs are defined in an independent set of data identifying face-related activation local maxima from a purely visual task (see Supplementary Fig. 14 and its associated note).

In the STS, in both of the face (F+ and F−) and social contexts (S1+, S1−, S2+ and S2−), %SC in the visual condition relative to fixation across all ROIs is not significantly different from %SC in the auditory congruent condition relative to fixation although a trend can be noted, (Fig. 8, left, Wilcoxon two-sided non-parametric test: FACE: AC vs V: $Z = 1.68$, $p = 0.09$. SOCIAL: AC vs V: $Z = 2.4$, $p = 0.051$). The STS thus appears equally responsive to visual and auditory social stimuli (%SC of all contexts are significantly different from fixation %SC, Wilcoxon non-parametric test, FACE: AC: $Z = 16.14$, $p < 0.001$; V: $Z = 19.35$, $p < 0.001$; SOCIAL: AC: $Z = 11.49$, $p < 0.01$; V: $Z = 14.87$, $p < 0.001$). In

contrast, in the LS, %SC in the visual condition relative to fixation across all ROIs is significantly different from %SC in the auditory congruent condition relative to fixation, (Fig. 8, left, two-sided Wilcoxon non-parametric test, FACE: AC vs V: $Z = 3.97$, $p < 0.01$; SOCIAL: AC vs V: $Z = 4.7$, p < 0.01). This result therefore suggests a strong auditory preference for LS (%SC of all auditory congruent are significantly different from fixation, Wilcoxon non-parametric test, FACE: $Z = 11.65$, $p < 0.001$; SOCIAL: $Z = 5.86$, $p < 0.01$), although LS is also significantly activated by the visual stimuli in the face context (V: $Z = 4.84$, $p < 0.01$). Last, V and AC activations were significantly weaker in the social context relative to the face context (AC: STS: $Z = 7.17$, $p < 0.001$; LS: $Z = 4.9$, $p < 0.001$; V: STS: $Z = 6.54$, $p < 0.001$; LS: 4.32 $p < 0.001$). This is most probably due to the fact that both visual (faces vs. social scenes) and auditory stimuli (coos + aggressive calls vs. coos + aggressive calls + screams) were different between the two contexts. This could have resulted in low level sensory differences in stimulus processing due to differences in spatial and auditory frequency content. Alternatively, these differences might have generated a different engagement from the monkeys in the task for faces and scenes. Yet, another possibility is that the non-human primate brain does not process in exactly the same way the association of social auditory stimuli with facial expressions and with scenes. This will have to be further explored. Overall, therefore, LS appears preferentially sensitive to auditory stimuli whereas the STS appears more responsive to visual than auditory stimuli. In Supplementary Fig. 5B, we show the modulation index of AC versus Vi for both sulci and type of context.

**Visual-auditory integration in the STS during the social contexts**
When processed in the brain, sensory stimuli from different modalities are combined such that the neuronal response to their combined processing is different from the sum of the neuronal responses to each one of them. This process is called multisensory integration[48] and is more pronounced when unimodal stimuli are ambiguous or difficult to

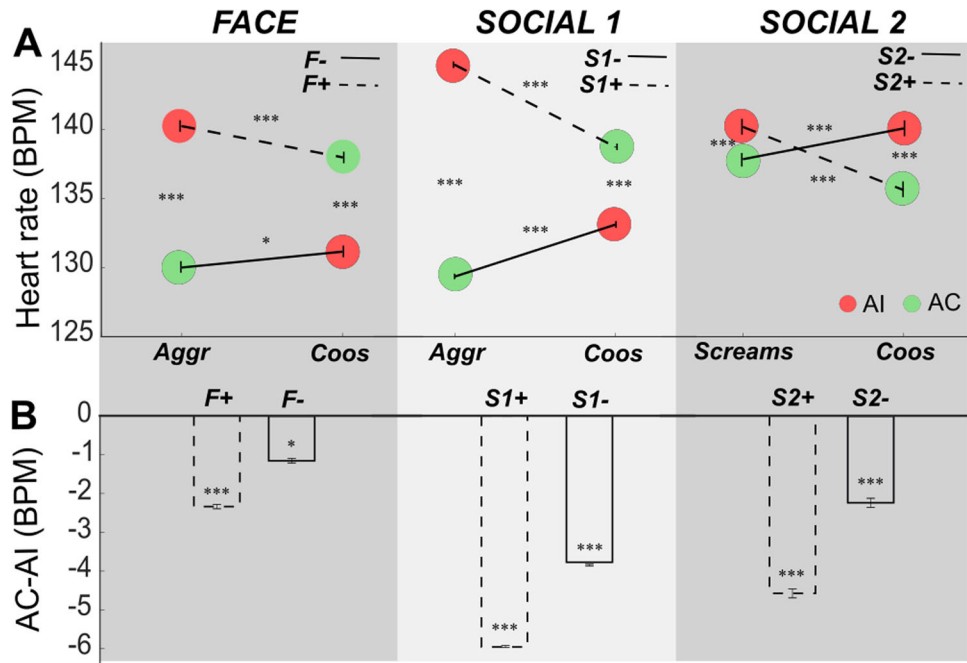

**Fig. 6 | Context-related heart rate (BMP) variations. A** Absolute heart rate (BMP. beats per minute) during the congruent (green) and incongruent (red) auditory blocks of each task. Dashed lines correspond to the positive affiliative context (F+, S1+ and S2+) as defined by the visual stimuli. whereas continuous lines refer to the negative aggressive (F− and S1−) or escape contexts (S2−). Contexts are defined by pairs involving the same vocalisation categories but different visual stimuli, as defined in Fig. 1b. There is a general context effect on heart rate (Friedman non-parametric test, $X_{2(253)} = 437.8$, $p = 6.7e\text{-}286$). There is a significant difference of HR for a same sound as a function of the context (Wilcoxon paired two-sided non-parametric test, all $n = 127$, aggressive calls between the F+ and F− contexts: $Z = 13.77$, $p = 3.6e\text{-}43$, Cohen's d: 16.3 and S1+ and S1−: $Z = 13.82$, $p = 1.8e\text{-}43$, Cohen's d: 91.6; Coos between the F+ and F−: $Z = 13.87$, $p = 9.1e\text{-}44$, Cohen's d: 47.05, S1+ and S1−: $Z = 13.78$, $p = 3.5e\text{-}43$, Cohen's d: 17.42 and S2+ and S2− contexts: $Z = 13.78$, $p = 3.6e\text{-}43$, Cohen's d: 5.987 and for screams between S2+ and S2− contexts: $Z = 13.77$, $p = 3.6e\text{-}43$, Cohen's d: 4.01). Each context pair shows significantly higher heart rates for incongruent auditory stimuli compared to congruent auditory stimuli (Friedman non-parametric test, Face: $X_{2(253)} = 271.442$, $p = 2.8e\text{-}82$; Social 1:, $X_{2(253)} = 295.34$, $p = 1.3e\text{-}87$; Social 2:, $X_{2(253)} = 174.66$, $p = 5.4e\text{-}78$). This is also true for each individual context (Wilcoxon paired two-sided non-parametric test. F+ : $Z = 13.98$, $p = 9.1e\text{-}49$, Cohen's d: 4.5, F−: $Z = 9.77$, $p = 0.012$, Cohen's d: 3.9, S1+: $Z = 13.76$, $p = 4.4e\text{-}49$, Cohen's d 19.7, S1−: $Z = 13.72$, $p = 4e\text{-}49$, Cohen's d 18.66, S2+: $Z = 13.82$, $p = 3.6e\text{-}49$, Cohen's d: 8.1, S2−: $Z = 13.77$, $p = 4.4e\text{-}49$, Cohen's d: 2.92). **B** Difference between AC and AI bloc (medians ± s.e). All significantly different from zero (Wilcoxon paired two-sided non-parametric test. F+ : $n = 127$, $Z = 13.98$, $p = 4.4e\text{-}5$, Cohen's d: 4.5, F−: $n = 127$, $Z = 9.77$, $p = 0.012$, Cohen's d: 3.9, S1+ : $n = 127$; $Z = 13.76$, $p = 2.4e\text{-}04$, Cohen's d 19.7, S1−: $n = 127$, $Z = 13.72$, $p = 2e\text{-}04$, Cohen's d: 18.66, S2+ : $n = 127$, $Z = 13.82$, $p = 4.3e\text{-}05$, Cohen's d: 8.1, S2−: $n = 127$, $Z = 13.77$, $p = 2.4e\text{-}04$, Cohen's d: 2.92. Note that for every item, Cohen's d coefficient is higher than 0.8. Each effect size is therefore considered as large. ***, $p < 0.001$; **, $p < 0.01$; *, $p < 0.05$; n.s., $p > 0.05$.

perceive[49,50]. The question here, therefore, is whether and how the LS and the STS combine visual and auditory social stimuli as a function of their semantic congruency. Multisensory integration is not straightforward to assess based on fMRI signals. A minimal criterion here would be to have significant %SC differences between the bimodal conditions and both of the unimodal conditions. Figure 9 shows the whole brain activation maps obtained for the two visual-auditory conditions, congruent (VAC, Fig. 9A) and incongruent (VAI, Fig. 9B) contrasted with fixation, as well as for the visual condition (vs. fixation) and the auditory condition (vs. fixation). These contrasts are presented for both face (Fig. 9, left panel) and the social contexts (Fig. 9, right panel). Figure 9C presents the contrast between the incongruent and congruent visuo-auditory conditions (VAI vs VAC).

Overall, in the face context, activations in the audio-visual conditions are not significantly different from the visual and auditory conditions alone (Fig. 9A, B, left panel). Likewise, no significant difference can be seen between the congruent and incongruent visuo-auditory conditions (Fig. 9C, left panel). Supplementary Fig. 15 compares %SC for the bimodal and unimodal conditions across all STS selected ROIs and all LS selected ROIs. Neither reach the minimal criteria set for multisensory integration. In the social context, activation in the audio-visual conditions show local significant differences relative to the visual and auditory conditions alone (Fig. 9A, B, right panel). When comparing the %SC for the bimodal and unimodal conditions across all STS selected ROIs and all LS selected ROIs, the STS ROIs reach the minimal criteria set for multisensory integration,

as their %SC is significantly different from each of the bimodal conditions and each of the unimodal conditions (Wilcoxon non-parametric test, AC vs VAC: $Z = 5.35$, $p < 0.01$; AC vs VAI: $Z = 4.06$, $p < 0.01$; V vs VAC: $Z = 2.64$, $p < 0.01$; V vs VAI: $Z = 2.48$, $p < 0.01$). Thus, multisensory integration appears to take place, specifically in the STS, and during the social context, possibly due to the higher ambiguity in interpreting social static scenes relative to faces (Supplementary Fig. 15). Importantly, and while most significant activations in the bimodal vs. unimodal auditory conditions are located within the audio-visual vs. fixation network, a bilateral activation located in the anterior medial part of the LS deserves attention. Indeed, this activation, encompassing part of the insula and of anterior SII/PV, is identified both in the congruent and incongruent auditory conditions and might be involved in the interpretation of semantic congruence between the visual and auditory stimuli. This possibility is addressed in the Discussion that follows.

## Discussion

Based on heart rate estimates and fMRI, our results show that rhesus monkeys systematically associate affiliative facial expressions or social scenes with corresponding affiliative vocalisations, aggressive facial expressions or social scenes with corresponding aggressive vocalisations, and escape visual scenes with scream vocalisations. In contrast, vocalisations that are incompatible with the visual information are fully suppressed, suggesting a top-down regulation over the processing of sensory input. In other words, rhesus monkeys correctly associate the

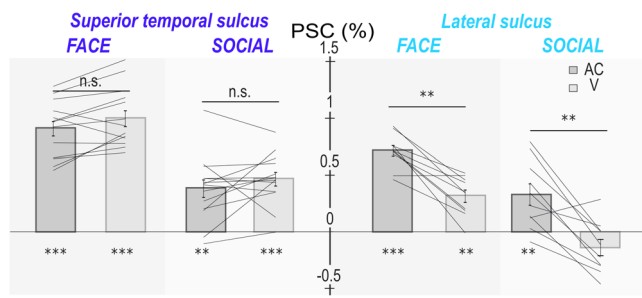

**Fig. 7 | Percentage of signal change (%SC) for selected left and right hemisphere ROIs in the lateral sulcus (light blue) and in the superior temporal sulci (dark blue). A** ROIs are 1.5 mm spheres located at local peak activations. Left and right hemisphere numbering associate mirror ROIs. ROI location in the each of the left and right STS and LS is described in the bottom flat maps. **B** %SC (median) are presented for each ROI (eight in right STS, six in left STS, four in left and six in right lateral sulcus) and each contrast of interest (V visual vs fixation, AC auditory congruent vs fixation, AI auditory incongruent vs fixation, VAC visuo-auditory congruent vs fixation, VAI visuo-auditory incongruent vs fixation).

**Fig. 8 | Percentage of signal change (PSC) across all lateral sulcus (light blue) and superior temporal sulci (dark blue) ROIs of both hemispheres, comparing the auditory and visual contexts (median ± s.e., single lines correspond to the PSC computed over single ROIs from the group analysis; $n = 14$ ROIs for STS and $n = 10$ ROIs for LS).** Statistical differences relative to fixation are between contexts and indicated as follows: ***, $p < 0.001$; **, $p < 0.01$; n.s., $p > 0.05$ (Wilcoxon two-sided non-parametric test). STS: FACE: AC: $n = 560$, $Z = 16.14$, $p = 4.1e$-57; V: $n = 560$, $Z = 19.35$, $p = 1.8e$-68; AC vs V: $Z = 1.68$, $p = 0.09$. SOCIAL: AC: $n = 1106$, $Z = 11.49$, $p = 0.0011$; V: $n = 1106$, $Z = 14.87$, $p = 1.5e$-49; AC vs V: $Z = 2.4$, $p = 0.051$. LS: FACE: AC: $n = 400$, $Z = 11.65$, $p = 2.4e$-31; V: $n = 400$, $Z = 4.84$, $p = 0.002$; AC vs V: $Z = 3.97$, $p = 0.01$. SOCIAL: AC: $n = 790$, $Z = 5.86$, $p = 0.002$; V: $n = 790$, $Z = -0.7$, $p = 0.45$; AC vs V: $Z = 4.7$, $p = 0.0013$.

meaning of a vocalisation with the meaning of a visual scene. This audio-visual, semantic binding with contextual information relies on a core functional network involving the superior temporal sulcus (STS) and the lateral sulcus (LS). LS regions of interest (ROIs) have a preference for auditory and audio-visual congruent stimuli while STS ROIs respond equally to auditory, visual and audio-visual congruent stimuli. Multisensory integration is only identified in the STS and only in the social condition in which visual information is expected to be more ambiguous than in the face condition. These observations are highly robust as they are reproduced over six sets of independent behavioural contexts, involving distinct associations of visual and auditory social information.

## Interpretation of social scenes and vocalisation by macaque monkeys

As is the case for human oral communication, monkey vocalisations are expected to be interpreted as a function of their emotional or contextual meaning. For example, a monkey scream indicates potential danger, is associated with fear and calls for escape and flight from the dangerous context. In contrast, coos are produced during positive social interactions and often elicit approach[8,9]. Here, we show that when two different types of vocalisations are presented together with a social visual stimulus, the heart rate of the monkeys is significantly lower when the vocalisation is congruent with the visual scene than

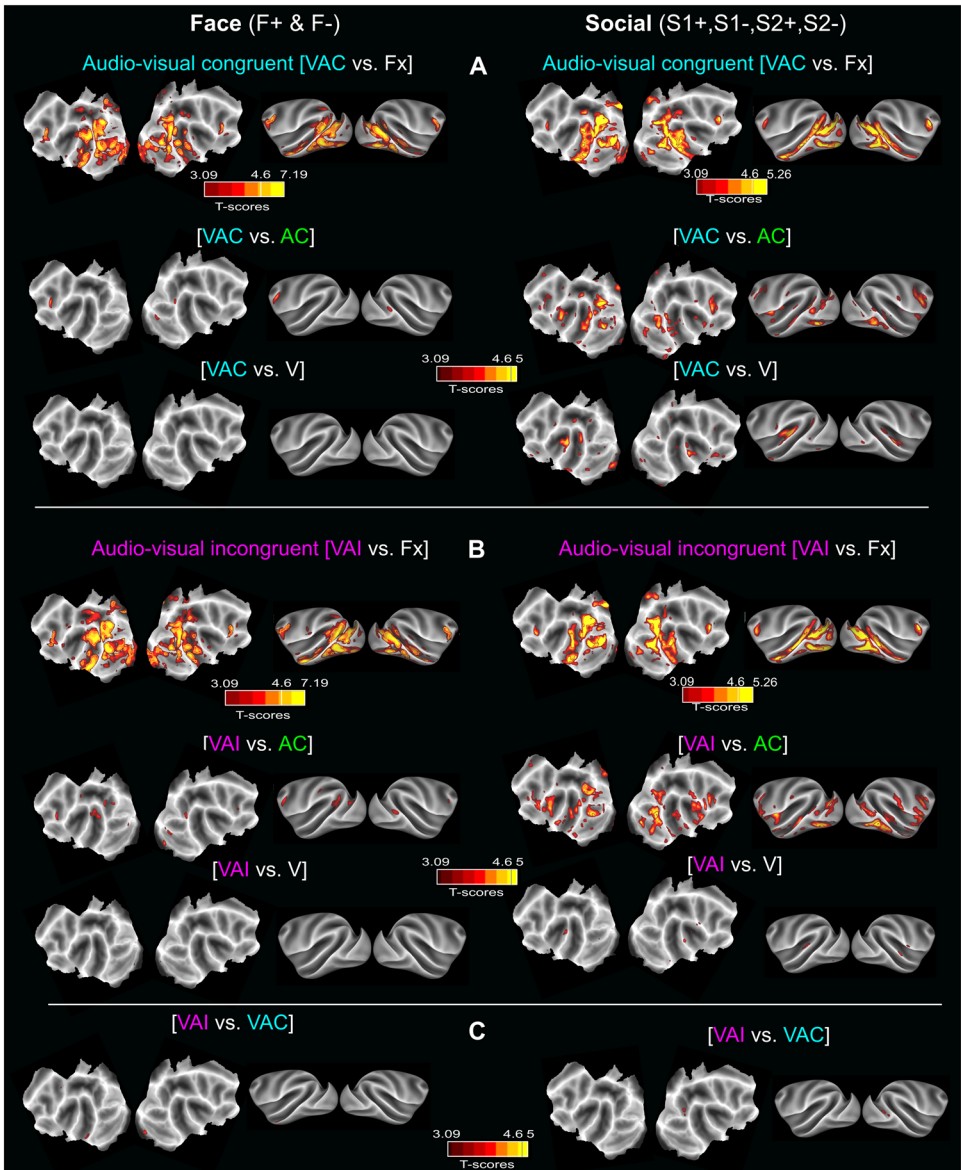

**Fig. 9 | Whole-brain activations for the Face (F+& F−) and Social contexts (S1+, S1−, S2+& S2−): bimodal versus unimodal contrasts. A** Whole-brain activation maps of the F+ (face affiliative) and F− (face aggressive) runs (left panel) and the S1+, S2+ (social affiliative 1 & 2), S1− (social aggressive) and S2− (social escape) runs (right panel) for the congruent audio-visual stimulation (blue). Contrasts from top to bottom: audio-visual vs. fixation, audio-visual vs. auditory congruent and audio-visual vs. visual. **B** Same as in (**A**) but for the incongruent audio-visual stimulation (pink). **C** Whole-brain activation maps for the audio-visual incongruent vs audio-visual congruent contrast. All else as in (**A**). Darker shades of red indicate level of significance at $p < 0.001$ uncorrected, $t$-score $> 3.09$. Lighter shades of yellow and brown outlines indicate level of significance at $p < 0.05$ FWE correction, $t$-score $> 4.6$. DF = [1, 5200] for Face and DF = [1,10344] for Social.

when the vocalisation is incongruent with the scene. Likewise, we show that the activity of the voice processing network is dramatically suppressed in response to the incongruent vocalisation. This pattern of activation provides neurobiological evidence that macaques infer meaning from both social auditory and visual information and are able to associate congruent information. In the network of interest, activations are not significantly different between the auditory, visual or audio-visual conditions. Most interestingly, aggressive calls are associated with both aggressive faces and aggressive social scenes, whereas coos are associated with both lipsmacks and inter-individual social grooming. We thus propose that these networks represent social meaning irrespective of sensory modality, thereby implying that social meaning is amodally represented. We hypothesize that such representations are ideal candidate precursors to the lexical categories that trigger, when activated, a coherent set of motor, emotional and social repertoires.

## Audio-visual association based on meaning and multisensory integration

The strict definition of multisensory integration involves the combination of sensory inputs from different modalities under the assumption of a common source[51,52]. In this context, it has been shown that multisensory integration speeds up reaction times and enhances perception[53–57], including when processing lip movement during speech[58–60]. Multisensory processes are also at play to predict the consequences of one modality onto another, i.e. in the temporal domain[61–64]. At the neuronal level, multisensory integration is defined as a process whereby the neuronal response to two sensory inputs is different from the sum of the neuronal responses to each on its own[48,65]. In the present study, the auditory and visual stimuli are associated based on their meaning (e.g., coos are associated with grooming) and possible contingency (e.g., screams are associated with escape scenes). Thus the audio-visual association described here goes

beyond the strict definition of two sensory inputs produced by a common source.

Additionally, by task design, two levels of audio-visual congruency can be defined. Level one is a first order congruency, defined within the audio-visual blocks, such that the auditory information can either be congruent (VAC) or incongruent (VAI) to the visual information. Level two is second order congruency, defined at the level of the run, such that given the visual information presented in a given run, the pure auditory blocks can either be defined as congruent (AC) or incongruent (AI) to the general visual context of this specific run, even if not simultaneously presented with the visual information. In order to probe whether first order congruency gives rise to multisensory integration, we applied the less stringent multisensory integration criteria used in fMRI studies, testing if audio-visual responses are statistically higher or lower than each of the uni-sensory conditions[66–70]. Face-voice integration has been described in the auditory cortex (CL, CM, in awake and anaesthetised monkeys; A1 only in awake monkeys) and the STS[40,71], and to a lesser extent in specific face-patches[36]. This latter study is worth noting as their experimental design matched, in important ways, our own, including audio-visual, visual only or auditory only stimuli. They used both monkey movies with a perfect match between visual and auditory stimulation in the audio-visual stimulus and created a computer-generated animated macaque avatar with the explicit intention of having synchronisation between the vocalisation and facial movements of the avatar. The study was thus explicitly testing multisensory integration under the hypothesis that the visual and auditory stimuli were associated with a common source. The audio-visual stimuli thus achieved a double congruence: they were temporally synchronised such that facial movements predicted vocalisations and as a consequence, they matched in semantic content. In the present study, our aim was to study the second type of congruence, i.e. semantic congruence. Our audio-visual stimuli were therefore not synchronised, but the two stimuli, when presented at the same time could be congruent (or incongruent) in semantic terms. The face-voice or scene-voice multisensory integration described by Khandhadia et al. is of a different nature to the one we report here. More specifically, in the present data, enhancement of the audio-visual response can only be seen in the contexts involving visual scenes. The parsimonious interpretation of these observations is that face-vocalisation binding was easier than scene-vocalisation binding and resulted in signal saturation, in agreement with the fact that neuronal multisensory integration is more pronounced for low saliency stimuli. The most significant difference between our study and that of Khandhadia et al. pertains to the second order congruency, an issue we discuss next.

Second order congruency is set by the visual information defining any given experimental run and results in major differences in how congruent and incongruent sounds are processed including in the absence of any visual stimulation. Congruent auditory information results in enhanced cortical activations relative to previous reports. Indeed, auditory activations have already been described in the STS[22,23,25,29,40,71,72]. However, and specific to our task design, the STS auditory activations described here in response to the congruent auditory stimuli are as strong as the visual responses (though with a trend to being slightly significantly weaker) and extend into the extrastriate visual cortex, thus suggesting cross-modal enhancement. In contrast, we show in this study an inhibition of irrelevant auditory information as a function of the context set by visual information. This process of filtering incongruent social auditory stimuli relative to social visual stimuli has already been shown at the behavioural level. Specifically, adults are shown to reliably filter out irrelevant social auditory information as a function of visual information while children below age 11 found this more challenging[73]. This was even more marked for children below age 7. This capacity of adults to filter irrelevant information is thought to arise from cross modal inhibition.

Such cross-modal inhibition has for example been described in the auditory cortex in response to visual and auditory stimuli presented simultaneously. Importantly, such cross-modal inhibition has been shown to switch on or off as a function of the context[74]. Accordingly, functional interactions between the visual and auditory networks can either result in an enhancement or in a suppression of cortical activity depending on the task and the presented stimuli[75]. The results we present here go beyond these early observations, as the inhibition of the irrelevant auditory stimulus does not take place at the time of presentation of the visual stimulus but when presented on its own, as the context is not set on a single trial basis but rather in well segregated behavioural runs. We hypothesize that our observations rely on a generalized form of cross-modal inhibition. This will have to be tested experimentally.

As discussed above, a specificity of our task design is that it creates, within each run, an implicit association between a set of social visual stimuli and their auditory match, possibly based on past learned sensory-motor associations and the development of internal models of what vocalisations are produced in a given visual context. This is very reminiscent of the recent description of auditory fMRI activations to learned sound sequences in the motor cortex of the macaque brain[76]. These auditory responses were only present in monkeys who had received an audio-motor training and were only present in response to the learned sound and were absent for other sounds. The authors propose that an internal model of auditory perception associating a given auditory set of stimuli with a given motor repertoire (and thus motor structure) was created by the training. We here argue that likewise, our current observations arise from the fact that macaques have, throughout their lifespan, associated specific macaque calls with specific social visual experiences, and that our specific task design allows to reveal this internal model. It is an open question as to how this mapping develops in young rhesus monkeys, and what experience is necessary.

It is worth noting that our results go against the predictive coding theory. This theory posits that the brain is constantly generating and updating an internal model of the environment. This model generates predictions of sensory input and compares these to actual sensory input[77,78]. Prediction errors are then used to update and revise the internal model[79]. In the context of predictive coding, when viewing an affiliative face, monkeys are expected to predict affiliative vocalisations. As a result, aggressive vocalisations in the context of affiliative faces are expected to generate prediction errors and hence higher activations than those observed for the affiliative vocalisations. This is not what our data show: when, viewing affiliative faces, there are enhanced responses to affiliative vocalisation and suppressed responses to aggressive vocalisations. This effect actually builds up as visual contextual information is reinforced through the run and is present in both the STS and the LS, i.e. at the early stages of auditory processing. Thus, these observations are inconsistent with the predictive coding experimental predictions. They suggest, instead, that the monkeys implement an active matching or association between the visual and the auditory social information, similar to a match to sample task, based on their life-long social experiences. In match to sample fMRI and EEG studies in humans[80] and electrophysiology studies in non-human primates[81,82], responses to the probe matching the sample is significantly higher than the response to a non-match probe, thus describing a match enhancement[83]. This is very similar to what we describe here, if considering the visual context as the probe and the auditory stimuli as the match and non-match probes. Further work is required to confirm this hypothesis.

An important question is how context is implemented into LS and STS. The STS is involved in multisensory integration and is shown to play a modulatory role on lateral sulcus functions during audio-visual stimulations[38,40,72]. However, the mechanisms subserving the observed selective cross modal inhibition of auditory processing based on the

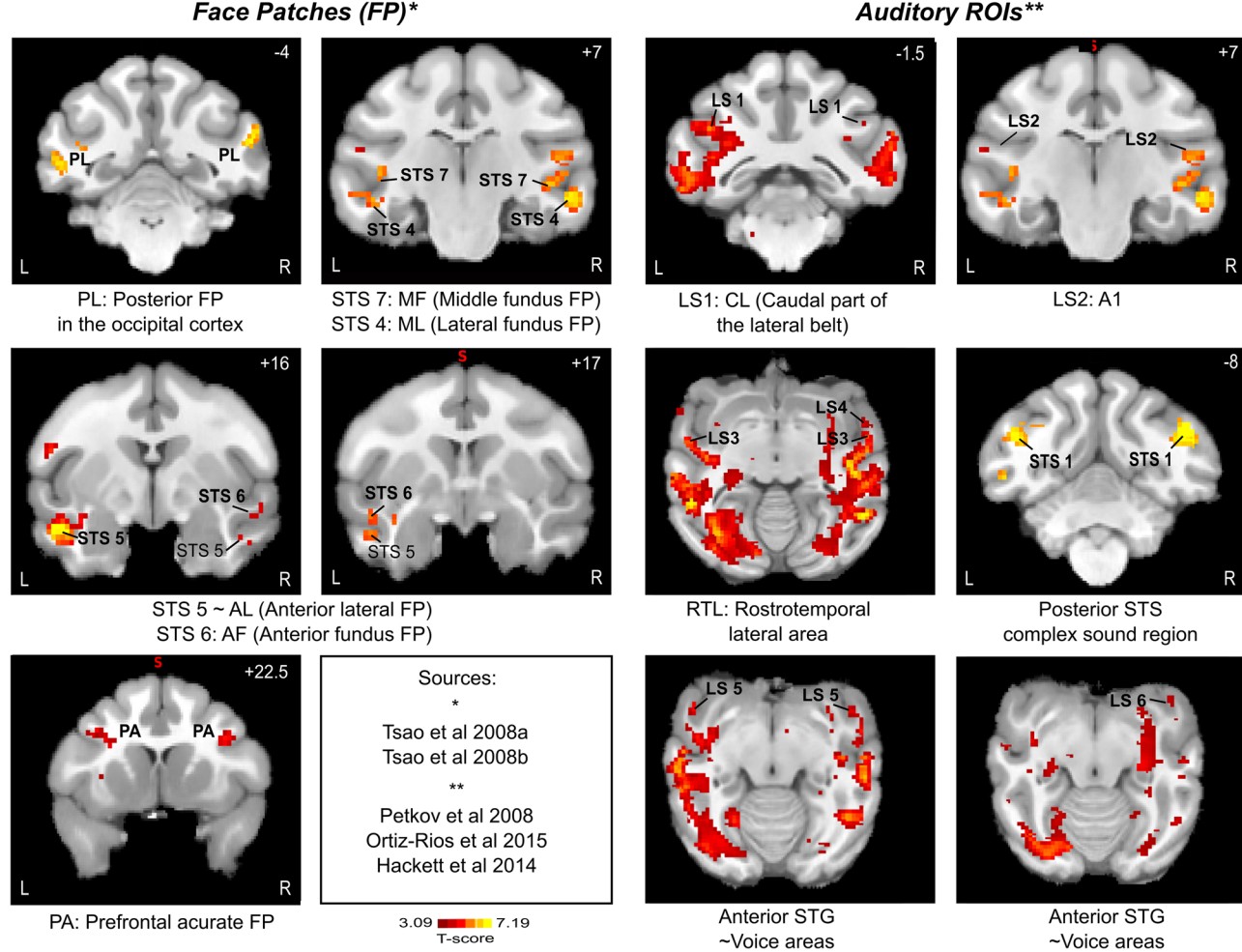

**Fig. 10 | Correspondence between task-related ROIs and face patches (left panels) and voice areas (right panels).** Color-scale runs start at *p* < 0.001 uncorrected levels. Task related ROIs are numbered as in Fig. 7. PA prefrontal acurate, AM anterior medial, AF anterior fundus, AL anterior lateral, MF middle fundus, ML middle lateral, PL a posterior face patch in the occipital cortex, CL Caudal part of the lateral belt, A1 primary auditory cortex, RTL Rostrotemporal lateral area, STS Superior Temporal Sulcus, STG Superior Temporal Gyrus. *: Sources for face patch localization. **: Sources for voice areas.

visual context are implemented not just during audio-visual blocks but throughout any given run. As a result, they are expected to originate from a higher order cortical region exerting a top-down control on both the LS and the STS. The prefrontal cortex is a choice region in this respect, as it connects to LS[84–86] and STS[87,88] and has been shown to play a crucial role in working memory and the top-down modulation of perception[89,90]. LS and STS are also connected to the cingulate cortex and orbitofrontal cortex[91,92]. These cortical regions that are involved in the processing of social interactions from visual cues and are thus in a position to provide feedback to the LS and STS based on the social dimension of the stimuli[13,93–96]. Lastly, LS and STS are also connected to the limbic system[91,92,97]. Accordingly, conspecific vocalisations activate a network recruiting, in addition to the voice patches, visual areas such as V4, MT, STS areas TE and TEO, as well as areas from the limbic and paralimbic system, including the hippocampus, the amygdala and the ventromedial prefrontal cortex (vmPFC)[18]. All of these regions are expected to contribute (most probably in coordination) to setting the context based on which auditory information is considered either as congruent or incongruent. These possibilities will have to be addressed experimentally.

**Audio-visual association based on social meaning possibly recruits the face and voice patches**

Face processing is highly specialized in the primate brain[20]. In the macaque brain, it recruits a specific system called the face patch

system, composed of interconnected areas, identified by both fMRI[14,15,17,21,24,27,28,35,98–100] and single cell recording[24,101,102]. This system recruits areas in the superior temporal sulcus, as well as in the prefrontal and orbito-frontal cortex. Specific limbic and parietal regions are also recruited together with this core system during, respectively, the emotional and attentional processing of faces[33]. The core face patches are divided into five STS areas (Anterior medial, AM; anterior fundus, AF; anterior lateral, AL; middle fundus, MF and middle lateral ML) and the PL (posterior lateral patch), a posterior face patch in the occipital cortex[35,103]. Based on a review of the literature, and anatomical landmark definitions, we associate the activation peaks identified in the present study with these five face patches (Fig. 10). Correspondence is unambiguous and the STS 4 ROIs matches ML, STS 7 matches MF, STS 5 matches AL and STS 6 matches AF. The occipital face patch PL is also identified in the general contrast maps as well as the frontal area defined in the literature as PA (prefrontal accurate)[44]. It is worth noting that in our experimental design, these face patches are activated both during the purely auditory congruent condition as well as during the visual conditions. Such activations are not reported by others during purely auditory conditions, indicating that this network is recruited during audio-visual association based on meaning. In the right hemisphere, two supplementary STS activations are reported, STS 2 and STS 3. They are located posterior to the putative ML face patch and ventral to the gaze following patch reported in the dorsal posterior infero-temporal cortex[104] and possibly coincide with an area

in the middle superior temporal cortex that has been recently described as modulated by the predictability of social interactions[105], though this would have to be tested explicitly.

The auditory processing circuit is proposed to be organized in two main networks, a ventral and a dorsal network (see for review[84,106,107]), such that the auditory ventral stream, also called the pattern or "what" stream, is activated by conspecific vocalisations whereas the dorsal stream, also called spatial or "where" stream, is involved in the spatial location of sounds[25,32]. Similarly, to face patches, voice processing by the "what" auditory stream, also involves a system of voice patches (for review, see[108]). In macaques, voice specific areas include the anterior superior temporal gyrus (aSTG), the orbitofrontal cortex (OFC) and a part of the STS close to the lateral sulcus[16,22,23,26,29,39]. This functional dissociation is observable as early as in the lateral belt such that its caudal part (CL) is selectively associated with sound location while the anterior part (AL) is more linked to sound identity such as vocalisations[106,109,110]. Again, based on a review of the literature, and anatomical landmark definitions, we associate the activation peaks identified in the present study with these voice patches (Fig. 10, This network is composed of prefrontal and temporal areas, but also, of visual striate and extrastriate visual areas (see Supplementary note attached to Supplementary Tables 1–3). Correspondence is unambiguous and the LS 1 ROI can be associated to CL (i.e. dorsal sound processing pathway) and LS2 to core primary auditory area A1. Within the ventral sound processing pathway, LS 4 ROI can be associated to area AL, LS 5 to rostro-temporal lateral area (RTL) and LS 6 to the rostro-temporal polar field (RTp). Last, LS 6 is compatible with the anterior most voice STG area described by Petkov and colleagues[26]. The voice patch system also involves the ventral dorsolateral prefrontal cortex or vlPFC[31], located in the inferior dimple at the boundary between area 45a and 46[111]. This cortical region has been proposed to play a key role in the cognitive control of vocalisations as well as in the interpretation of call meaning[112]. Microstimulations further indicate that this prefrontal voice patch is functionally connected with the putative macaque homologue of human's Broca area 44[113]. In the present study, the ventral prefrontal activation, while matching nicely with the PA face patch, only partially overlaps with the prefrontal voice patch, suggesting a possible functional specialization. Taken together, these results strongly suggest that the association between vocalisation meaning and social visual stimuli recruits the face and voice patch system.

Visual fMRI activations have already been described in the LS, in the primary auditory cortex and in the non-primary core (belt)[114]. This observation has been confirmed using single cell recording studies[115]. An important question to be addressed by single unit recording studies is whether the STS auditory activations correspond to neuromodulatory LFP modulations or to actual spiking activity. Quite interestingly, while we identify an audio-visual gradient between the LS and the STS, the LS showing higher activations for voice as compared to visual social stimuli, and the STS showing a preference for visual stimuli compared to auditory stimuli, no clear gradient of auditory or visual activations can be identified either within the STS or within the LS. This suggests that voice-social visual associations rely on the activity of the entire network, rather than on some of its subparts.

## Visual and auditory responses in the lateral sulcus and superior temporal sulcus

Expectedly, the LS activations in response to auditory stimuli are higher than its activations to visual stimuli (Fig. 8). This most probably arises from the fact that while the primary function of the LS is auditory processing, it receives (visual) input from the adjacent STS[38,116]. In contrast, based on the ROIs defined in the audio-visual face task, STS appears to be equally responsive to auditory and visual stimuli (Fig. 8, trend to significance), although AC − V/AC + V modulation indexes are significantly negative (Supplementary Fig. S5). When ROIs are defined

on the basis of a purely visual task, STS visual responses are significantly higher than STS auditory responses (Supplementary Fig. 14). Overall, this suggests the existence, within the STS, of specialized regions involved in the visuo-auditory association of social stimuli. While large areas of the STS become responsive to auditory stimuli during visuo-auditory association of social stimuli--perhaps due to a direct projection from the LS to the STS[88]--only some regions are activated to almost a similar level by both sensory modalities. These regions could contribute to the amodal representation of social stimuli.

To conclude, our experiments demonstrate, using indirect measures (heart rate and hemodynamic brain response), that macaque monkeys are able to associate social auditory and visual information based on their abstract meaning. This supports the idea that non-human primates display advanced social competences, amodally represented, that may have paved the way, evolutionary, for human social cognition, including its linguistic representations and expressions.

## Methods

### Subjects and surgical procedures

Two male rhesus monkeys (*Macaca mulatta*) participated in the study (T, 15 years, 10 kg and S, 12, 11 kg). The animals were implanted with a Peek MRI-compatible headset covered by dental acrylic. The anaesthesia for the surgery was induced by Zoletil (Tiletamine-Zolazepam, Virbac, 5 mg/kg) and maintained by isoflurane (Belamont, 1–2%). Post-surgery analgesia was ensured thanks to Temgesic (buprenorphine, 0.3 mg/ml, 0.01 mg/kg). During recovery, proper analgesic and antibiotic coverage was provided. The surgical procedures conformed to European and National Institutes of Health Guidelines for the Care and Use of Laboratory Animals.

### Experimental setup

During the scanning sessions, monkeys sat in a sphinx position in a plastic monkey chair[117] facing a translucent screen placed 60 cm from the eyes. Visual stimuli were retro-projected onto this translucent screen. Their head was restrained and the auditory stimuli were displayed by Sensimetrics MRI-compatible S14 insert earphones. The monkey chair was secured in the MRI with safety rubber stoppers to prevent any movement. Eye position (X, Y, right eye) was recorded thanks to a pupil-corneal reflection video-tracking system (EyeLink at 1000 Hz, SR-Research) interfaced with a program for stimulus delivery and experimental control (EventIDE®). Monkeys were rewarded for maintaining fixation into a $2 \times 2°$ tolerance window around the fixation point.

### General run design

On each run, monkeys were required to fixate a central cross on the screen (Fig. 1A). Runs followed a block design. Each run started with 10 s of fixation in the absence of sensory stimulation followed by three repetitions of a pseudo-randomized sequence containing six 16 s blocks: fixation (Fx), visual (Vi), auditory congruent (AC), auditory incongruent (AI), congruent audio-visual (VAC) and incongruent audio-visual (VAI) (Fig. 1A). The pseudo-randomization was implemented such that each block in each repetition was presented in a randomized order. Thus monkeys could not anticipate the sequence of stimuli. In addition, the initial blocks were either a visual block (Vi, VAC, VAI), or a fixation block followed by a visual block (Vi, VAC or VAI), such that context was set by visual information early on in each run. As a result, pure auditory blocks were always presented after a visual block and could thus be defined as congruent or incongruent to the visual information characterizing the block. Pseudo-randomization was also implemented such that, across all repetitions and all runs for a given context, each block was, on average, preceded by the same number of blocks from the other conditions. Quite crucially to the results

presented in this work, in 66% of the times, both AI and AC conditions were preceded by blocks involving visual stimulation (Vi, VAC and VAI). Last, each block (except the fixation block) consisted in an alternation of 500 ms stimuli (except for lipsmacks, 1 s dynamic stimuli succession) of the same semantic category (see Stimuli section below), in the visual, auditory or audio-visual modalities. In each block, 32 stimuli were presented randomly (16 for lipsmack). Each run ended by 10 s of fixation in the absence of any sensory stimulations.

## Face and social contexts

Six audio-visual contexts were presented to both monkeys, organized in runs as described above (Fig. 1A). Each context combined visual stimuli of identical social content with either semantically congruent or incongruent monkey vocalisations with the predominant visual stimuli (Fig. 1B). Runs always started by a block condition involving visual stimulations, thus setting the social context of the task and, as a result, defining auditory congruent and incongruent auditory stimuli. Given the structure of our task, two levels of congruency can be defined. A first order congruency is defined within the audio-visual blocks, such that the auditory information can either be congruent (VAC) or incongruent (VAI) to the visual information. The second order of congruence is defined at the level of the run, such that, given the visual information presented in a given run, the pure auditory blocks can either be defined as congruent (AC) or incongruent (AI) in this specific run, even if not simultaneously presented with the visual information. The face affiliative context (F+) combined lipsmacks with coos and aggressive calls. The face aggressive context (F−) combined aggressive faces with coos and aggressive calls. The first social affiliative context (S1+) combined grooming scenes with coos and aggressive calls. The second social affiliative context (S2+) combined grooming scenes with coos and screams. The social aggressive context (S1−) combined aggressive group or individual scenes with coos and aggressive calls. The social escape context (S2−) combined fleeing groups or individual scenes with coos and screams. Importantly, pairs of contexts (F+ &. F−; S1+ & S1−; S2+ & S2−) shared the same vocalisations, but opposite social visual content (i.e. opposite semantic content, defining either a positive or a negative context). All contexts were presented randomly and at least once during each scanning sessions.

## Stimuli

Vocalisations were recorded from semi-free-ranging rhesus monkeys during naturally occurring situations by Marc Hauser. Detailed acoustic and functional analyses of this repertoire has been published elsewhere (e.g.,[8,9]). Field recordings were then processed, restricting to selection of experimental stimuli to calls that were recorded from known individuals, in clearly identified situations, and that were free of competing noise from the environment. Exemplars from this stimulus set have already been used in several imaging studies[16,31,32,41,118]. All stimuli were normalized in intensity. The frequency ranges varied between the different types of stimuli as shown in Supplementary Fig. 16. For each of the three vocalisation categories, we used 10 unique exemplars coming from matched male and female individuals, thus controlling for possible effects due to gender, social hierarchy or individual specificity. Coos are vocalisations typically produced during affiliative social interactions, including grooming, approach, coordinated movement, and feeding. Aggressive calls are typically used by a dominant animal toward a subordinate, often as a precursor to an actual physical attack. Screams are produced by subordinates who are either being chased or attacked, or as they are witnessing others in the same condition. Face (lipsmacks and aggressive facial expression) and social scene (group grooming, aggressive individual alone or in group / escaping individual or group) stimuli were extracted from videos collected by the Ben Hamed lab, as well as by Marc Hauser on Cayo Santiago, Puerto Rico. Images were normalized for average intensity

and size. All stimuli were 4° x 4° in size. However, we decided to keep them in colour to get closer to natural stimuli even if it produced greater luminosity disparity between the different stimuli preventing us to use pupil diameter as a physiological marker. Only unambiguous facial expressions and social scenes were retained (Supplementary Fig. 16 and Fig. 1). A 10% blur was applied to all images, in the hope of triggering multisensory integration processes (but see result section). For each visual category, 10 stimuli were used.

## Scanning procedures

The in-vivo MRI scans were performed on a 3 T Magnetom Prisma system (Siemens Healthineers, Erlangen, Germany). For the anatomical MRI acquisitions, monkeys were first anesthetized with an intramuscular injection of ketamine (10 mg\kg). Then, the subjects were intubated and maintained under 1–2% of isoflurane. During the scan, animals were placed in a sphinx position in a Kopf MRI-compatible stereotaxic frame (Kopf Instruments, Tujunga, CA). Two L11 coils were placed on each side of the skull and a L7 coil was placed on the top of it. T1-weighted anatomical images were acquired for each subject using a magnetization-prepared rapid gradient-echo (MPRAGE) pulse sequence. Spatial resolution was set to 0.5 mm, with TR = 3000 ms, TE = 3.62 ms, Inversion Time (TI) = 1100 ms, flip angle = 8°, bandwidth=250 Hz/pixel, 144 slices. T2-weighted anatomical images were acquired per monkey, using a Sampling Perfection with Application optimized Contrasts using different flip angle Evolution (SPACE) pulse sequence. Spatial resolution was set to 0.5 mm, with TR = 3000 ms, TE = 366.0 ms, flip angle = 120°, bandwidth = 710 Hz/pixel, 144 slices. Functional MRI acquisitions were as follows. Before each scanning session, a contrast agent, composed of monocrystalline iron oxide nanoparticles, Molday ION™, was injected into the animal's saphenous vein (9–11 mg/kg) to increase the signal to noise ratio[117,119]. We acquired gradient-echoechoplanar images covering the whole brain (TR = 2000 ms; TE = 18 ms; 37 sagittal slices; resolution: 1.25 × 1.25 × 1.38 mm anisotropic voxels, flip angle = 90°, bandwidth = 1190 Hz/pixel) using an eight-channel phased-array receive coil; and a loop radial transmit-only surface coil (MRI Coil Laboratory, Laboratory for Neuro- and Psychophysiology, Katholieke Universiteit Leuven, Leuven, Belgium, see[120]. The coils were placed so as to maximise the signal on the temporal and prefrontal cortex. As a result, signal-to-noise was low in the occipital cortex (see Supplementary Fig. 4).

## Data description

In total, 76 runs were collected in 12 sessions for monkey T and 65 runs in 9 sessions for monkey S. Based on the monkey's fixation quality during each run (85% within the eye fixation tolerance window) we selected 60 runs from monkey T and 59 runs for monkey S in total, i.e. 10 runs per task, except for one task of monkey S.

## Data analysis

Data were pre-processed and analysed using AFNI (Cox, 1996), FSL (Jenkinson et al., 2012; Smith et al., 2013), SPM software (version SPM12, Wellcome Department of Cognitive Neurology, London, UK, https://www.fil.ion.ucl.ac.uk/spm/software/), JIP analysis toolkit (http://www.nitrc.org/projects/jip) and Workbench (https://www.humanconnectome.org/software/get-connectome-workbench). The T1-weighted and T2-weighted anatomical images were processed according to the HCP pipeline[121,122] and were normalized into the MY19 Atlas[123]. Functional volumes were corrected for head motion and slice time and skull-stripped. They were then linearly realigned on the T2-weighted anatomical image with flirt from FSL, the image distortions were corrected using nonlinear warping with JIP. A spatial smoothing was applied with a 3-mm FWHM Gaussian Kernel. A representative example of time courses is presented in Supplementary Fig. 17.

Fixed effect individual analyses were performed for each monkey, with a level of significance set at $p < 0.05$ corrected for multiple

comparisons (FWE, *t*-scores 4.6) and $p < 0.001$ (uncorrected level, t-scores 3.09). Head motion and eye movements were included as covariate of no interest. Because of the contrast agent injection, a specific MION hemodynamic response function (HRF)[117] was used instead of the BOLD HRF provided by SPM. The main effects were computed over both monkeys. In most analyses, face contexts and social contexts were independently pooled.

ROI analyses were performed as follows. ROIs were determined from the auditory congruent contrast (AC vs Fx) of face contexts with the exception of two ROIs of the right lateral sulcus (LS4 and LS6) that were defined from the same contrast of social contexts. ROIs were defined as 1.5 mm diameter spheres centred around the local peaks of activation. In total, eight ROIs were selected in the right STS, six from the left STS, four in the left LS and six in the right LS. Supplementary Fig. 13 shows the peak activations defining each selected ROI; so as to confirm the location of the peak activation on either of the inferior LS bank, the superior STS bank or the inferior STS bank. For each ROI, the activity profiles were extracted with the Marsbar SPM toolbox (marsbar.sourceforge.net) and the mean percent of signal change (±standard error of the mean across runs) was calculated for each condition relative to the fixation baseline. %SC were compared using non-parametric two-sided tests.

### Behaviour and heart rate
During each run of acquisition, videos of the faces of monkeys S and T were recorded in order to track heart rate variations (HRV) as a function of contexts and blocks[45]. We focus on heart rate variations between auditory congruent and incongruent stimuli. For each task, we extracted HRV during AC and AI blocs. As changes in cardiac rhythm are slow, analyses were performed over the second half (8 s of each block). This has been done for each run of each task, grouping both monkeys. Because the data were not normally distributed (Kolmogorov–Smirnov Test of Normality), we carried out Friedman tests and non-parametric post hoc tests.

### Reporting summary
Further information on research design is available in the Nature Research Reporting Summary linked to this article.

## Data availability
The data that support the findings of this study are available from the corresponding author upon reasonable request. Data are still being analysed for other purposes and cannot be made publically available at this time. A Source Data file provides the raw data used to create all of the figures of this paper except the whole brain fMRI contrast maps. Source data are provided with this paper.

## Code availability
The code that supports the findings of this study is available from the corresponding author upon reasonable request. The code is still being used for other purposes and cannot be made publically available at this time.

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

## Acknowledgements

S.B.H. were funded by the French National Research Agency (ANR)ANR-16-CE37-0009-01 grant and the LABEX CORTEX funding (ANR-11-LABX-0042) from the Université de Lyon, within the program Investissements d'Avenir (ANR-11-IDEX-0007) operated by the French National Research Agency (ANR). We thank Fidji Francioly and Laurence Boes for animal care, Julian Amengual and Justine Cléry for their rich scientific exchanges during data collection and analyses, Franck Lamberton and Danièle Ibarrola for their MRI methodological support and Holly Rayson for her help on visual stimuli collection. We thank Serge Pinède for technical assistance on the project.

## Author contributions

Conceptualization, S.B.H., M.F; Stimuli preparation, M.H., M.F, Q.G, M.G; Data Acquisition, M.F., M.G.; Methodology, M.F., S.C., Q.G., and S.B.H; Investigation, M.F. and S.B.H.; Writing – Original Draft, M.F. and S.B.H.; Writing – Review & Editing, S.B.H., M.F., M.H.; Funding Acquisition, S.B.H.; Supervision, S.B.H.

## Competing interests

The authors declare no competing interests.

## Ethics declaration

Animal experiments were authorized by the French Ministry for Higher Education and Research (project no. 2016120910476056 and 1588-2015090114042892) in accordance with the French transposition texts of Directive 2010/63/UE. This authorization was based on ethical evaluation by the French Committee on the Ethics of Experiments in Animals (C2EA) CELYNE registered at the national level as C2EA number 42.
