## [Peer Review File · Nature Communications]

Socially meaningful visual context either enhances or inhibits vocalization processing in the macaque brainREVIEWER COMMENTS

Reviewer #1 (Remarks to the Author):

Froesel and colleagues performed an fMRI study in two monkeys investigating audio-visual integration using socially meaningful stimuli. When monkeys are shown a combination of visual-auditory, and visual and auditory stimuli in isolation in different epochs, the meaning of the stimuli largely determines the fMRI responses in the superior temporal and lateral sulci. Specifically, when the auditory stimulus does not match the visual stimulus, activity is abolished, while matching auditory stimuli activate areas in the STS almost at the same level as the visual stimuli. This is indirect functional evidence showing that the monkeys can link the meaning of auditory and visual stimuli and that this association determines to a large degree the activity in the STS and LS. The fMRI results match with variations in heart rate during the different conditions. The authors also link these association of facial expressions with corresponding vocalizations to two distinct networks, based on a gPPI analysis.

This is a very exciting study, with highly surprising but seemingly robust results. I'm particularly impressed by the dramatic differences in activity evoked by the same auditory stimulus, depending on the context: i.e. congruent or incongruent relative to the emotional information carried by the visual stimulus. Also, the strength of the auditory responses in the superior temporal sulcus (when presented in the right context) are unpredicted. To the best of my knowledge, this is a highly novel finding in macaques that deserves to be published. Of course, this result raises a number of questions that should be addressed in follow-up studies: for example, what would happen if also congruent and incongruent visual stimuli were shown? What is the timeline of these associative-dependent activity changes. What is happening at neuronal level? Etc.

I'm less impressed by the gPPI analysis and I would suggest to drop this altogether. The 'one voxel' criterion (at $P < 0.005$ uncorrected) is impossible to defend, and as such these results weaken rather than strengthen the manuscript. The additional advantage to drop this part would be that the (too) lengthy discussion can be shortened considerably. In general, the text is sometime difficult to understand and there is room for grammatical improvements. Major:

1. It is difficult to understand why primary visual areas are not activated (in the standard visual versus fixation contrasts). In case this part of the brain was not scanned, the authors should indicate exactly what part was covered. Alternatively, provide temporal signal to noise maps, which may reveal differences in sensitivity across the cortex.
2. The heart rate results are in line with the fMRI results, which is interesting. However, this may also pose a problem as changes in heart rate might affect measured fMRI responses in a very complicated manner. For example, there are both positive and negative correlations with BOLD responses at short (6-12 sec) and longer (30-42 sec) time lags, respectively (e.g. Chang et al: Neuroimage. 2009 February 1; 44(3): 857–869.doi:10.1016/j.neuroimage.2008.09.029.). To rule out this unlikely but also more mundane explanation for their results, i.e. that the unexpected differences in fMRI activity are completely explained by differences in heart rate, the authors should add heart rate as an additional regressor (time-shifted) in their fMRI analysis.
3. There is value to show only group data. The reader must be convinced, however, that the data are not largely driven by one animal. Please address this issue (e.g. by showing conjunction maps of the two animals, or individual maps).

Despite the enthusiasm about this unexpected result, I believe that the presentation of the

data can be improved:

1. The dark green and dark red fonts on a black background should be avoided. These are difficult to read.
2. The flattened illustrations of the cortex are very different from what is usually shown. Therefore it is difficult 'to read them'. I would certainly add the operculum (which seems to be largely removed). I would also label all sulci.
3. The insets (indicating anterior / ventral etc..) in figures are not always correct.
4. Fig 6A: the orientation of the inflated brain and corresponding flatmaps are different. Please correct, as it hampers readability

Minor:

1. line 222: 'The STS appears equally responsive to visual and auditory stimuli.' But only statistics are provided for V vs Fix and A vs Fix. Please test directly V vs A to back up this statement.
2. As mentioned above, I would drop the PPI analysis. Please note that on Page 14, the authors wrongly refer to figure 8 instead of 9.
3. Line 405: PITd is located near the PMTS. The gaze following patch is located more dorsally (Marciniak et al. eLife 2014;3:e03222. DOI: 10.7554/eLife.03222)
4. Line 414: It is false to state that auditory responses have not yet been described in the STS. There are quite a few electrophysiology and fMRI papers in the monkey showing the opposite.

Reviewer #2 (Remarks to the Author):

In their manuscript "Neural bases of audio-visual integration of socially meaningful information in macaques", Froesel et al. use fMRI in macaque monkeys to assess the impact of audio-visual congruency versus incongruency in the social domain on brain activity and heart rate modulation. They report large changes of activity in the superior temporal sulcus (STS) and the lateral sulcus (LS) due to congruence and incongruence. The findings are entirely unexpected in sign, strength, and spatial extent. The authors then go on to determine functional connectivity of these two regions with other brain regions and find networks (with low significance) that are similar for STS and LS.

The main results, if true, are so astonishing, they would be some of the most remarkable findings in cognitive and systems neuroscience in the last thirty years. Figure 4 and others are suggesting, and Figure 6 shows directly the claim that areas throughout the STS show a response to a congruent auditory stimulus comparable to the visual stimulus in isolation, while an incongruent auditory stimulus hardly elicits any response at all – and a very similar pattern exists in the lateral sulcus. Considering that many areas in these analyses are classical visual or auditory areas, a result so expansive across these large regions and so strong is unexpected. Consider further that results on predictive coding would suggest the opposite pattern of results, a bigger response of the incongruent stimulus. So, there are three reasons to be really astonished by the results. To make as strong claims as the paper, maybe unwittingly, does require very thorough experimental design and controls. It is here where the paper falls severely short. Furthermore, the description of procedures and results is often unclear and unprecise, thus making it very difficult to fully understand what was done. I thus apologize up front, if I got something completely wrong.

My main concerns regard experimental design and lack of controls.

The authors suggest that the blocks within a run were varied following three pseudo-random sequences. (I cannot have more certainty, because descriptions of what the sequence is vary at different places of the manuscript. In Methods they state that there were "three repetitions of a pseudo-randomized sequence containing six possible 16 s blocks". It is

unclear why the blocks are called possible. The figure suggests that only the sequence of blocks was varied. "blocks randomization", a term used in Fig. 1, might also indicate randomization of image presentation within a block. Such lack of clarity in important descriptions are pervasive throughout the manuscript.)

The first problem with the design is that one would expect a fully counter-balanced design, especially when blocks follow each other. This however has not been done here. This is particularly problematic, when 16s blocks are used after administration of the contrast agent Molday ION which essentially doubles the time constant relative to the BOLD response.

Thus the response to each block massively "leaks over" into the response of the next one, and one has to make strong assumptions of linearity to be able to tell them apart.

The second problem with this procedure and lack of documentation of the three actual sequences is that it is not clear how congruence and incongruence were actually implemented. For an auditory stimulus presented in isolation to qualify as either congruent or incongruent, the visual stimuli must be presented before. One can only assume that the pseudorandom sequences were pseudo in the sense that this was guaranteed. However the example in Fig. 1 shows an incongruent audio-visual block leading.

Assuming that things were done this way, note then how radical a statement of congruence or incongruence might be: it might take half a minute for the incongruent auditory stimulus to be displayed after the visual condition, and one has to assume for the subject's brain to one prioritize the visual information as context for the auditory information and not vice versa, and for the brain to be able to differentiate between a congruent and an incongruent auditory condition after the passage of such a long period of time. One would expect a decline over time, which is not documented. But if it occurred, it would be hard to see how the response to congruent and incongruent auditory stimuli could come out as different as they do on average.

The authors find consistent results across multiple regions of interest (ROIs) in STS and LS. Leaving the problem of double-dipping in the selection of ROIs aside, there are huge differences in functional specialization within these regions. Take the face areas for example the authors are frequently referring to. These occupy small regions within the STS. Using the contrast agent the authors are using and face stimuli, one would expect much larger percent signal changes in these regions than the author report. It is possible that maybe coils were not positioned properly and thus the SNR was so much lower than expected. It would be essential for this paper to localize areas of functional specialization – for the visual and the auditory stimuli used – and define ROIs based on these. With these independently defined ROIs, it would then be possible to make a firm statement and say that face area x responds this much to faces in isolation and without context this much to vocalizations in isolation and without context, and then, in the main experiment we find the following pattern. It would then be possible to properly judge the new results in the context of established ones.

Currently, one sees large swathes of cortex activated, but one does not know which regions they correspond to, one does not know how reproducible results are. And it does not seem to matter to the authors who then lump them together to talk about the STS and LS, which are internally so heterogeneous.

I will not comment on connectivity results, which are statistically weak and not relevant for the main issues of the manuscript.

Adding to the difficulties understanding the manuscript is the lack of conceptual focus and a loose practice of citing the literature. Frequently the most relevant literature is not referenced or even wrong literature for the point in question.

Taken together, in its current form the manuscript does not allow for a thorough investigation of the results. This is very unfortunate should the results be correct, because they would have such high importance.

Reviewer #3 (Remarks to the Author):

This is an interesting study on the representation of species-specific communication calls in the macaque using functional MRI. The authors use a battery of calls, recorded in the wild by one of the co-authors (Marc Hauser, a well-known neuroethologist with expertise in communication calls and behavior of macaques). Each type of calls is uttered in specific situations or contexts, which can be affiliative (e.g. coos) or aggressive (e.g. growls). In their fMRI study, the authors play back the calls together with varying visual displays (faces or scenes) that are either congruent or incongruent with the meaning of the call. The initial set of scans (Figs. 1-4) yields very clear and impressive results, in that the combination of calls and incongruent visual displays leads to an almost complete suppression of the brain activation in auditory regions (lateral sulcus [LS] and superior temporal sulcus [STS]). Neurophysiologically, one would refer to these effects as crossmodal inhibition. Although crossmodal inhibition or suppression has been demonstrated in various other contexts, this is a novel set of circumstances and definitely worthy of reporting.

The authors then go beyond the immediate auditory-visual interactions and measure psychophysiological correlates in terms of heart rate changes (Fig. 5), using a device that was developed in their lab. This is valuable quantitative information at the behavioral level complementing purely behavioral observation.

In an extensive third part, the study tries to explain the effects measured with fMRI on the basis of connectivity with other brain regions. The region discussed first and foremost is prefrontal cortex, which is known to exert top-down modulation on lower-order sensory regions. It is also claimed to mediate emotional modulation (together with orbito-frontal cortex and parts of the limbic system). Second is parietal cortex in terms of attentional modulation. These two networks are then discussed in wide terms and in the context of "face and voice patches" (Fig. 10), dorsal and ventral streams, and social and emotional modulation. While this is a valuable and quite novel part of the study, I would strongly suggest that the results be discussed by comparison with classical neuroanatomical tracer studies. The work of Goldman-Rakic or of Joseph Price comes to mind. STS and its connections in particular have been studied quite extensively by Seltzer and Pandya 40 years ago and tends to be forgotten. I consider it a major and worthwhile task for today's imaging community to validate their results by relating them to those of classical anatomical tracer studies – across and within the same primate species.

This latter part of the Results and the Discussion threaten to become a bit of a hodgepodge of different ideas and seem to be driven by trendy topics, like face and voice patches or 'social neuroscience' in general. While vocal communication undoubtedly includes much social interaction, this should not be confused with the mechanistic aspects of visual-auditory interactions, all the way down to the single-unit level. These two aspects are distinct from one another and should be clearly kept apart and discussed as two distinct topics.

In summary, this is a heroic study that not only tries to measure visual-auditory interactions in vocal communication of nonhuman primates (which it does very nicely). By considering vocal communication as one form of social interaction, the study expands into other topics in social neuroscience, setting up top-down pathways that ultimately encompass the entire brain. It would be desirable to relate the results of functional connectivity from imaging to classical results of anatomical tracer studies. The functions of prefrontal and parietal cortex, seen here as crucial for top-down modulation, should be more clearly differentiated according to established neuroanatomical data and theory (e.g. Fuster). Otherwise the paper

is superb.

References: Although the authors try to do justice to each sub-field they discuss, they remain eclectic in terms of who gets mentioned. The vast literature on face patches may have too many references, while the literature on dual-stream (dorsal/ventral) pathways misses some key references (e.g. Rauschecker & Tian, 2000, and Rauschecker & Scott, 2009). The latter paper (together with Rauschecker, 2011, and the most recent Archakov et al., PNAS 2020) will also provide background on the important role of the dorsal stream in sensorimotor aspects of vocal communication and on internal models, something that is completely overlooked here.

Seltzer B, Pandya DN. Brain Res. 1978 Afferent cortical connections and architectonics of the superior temporal sulcus and surrounding cortex in the rhesus monkey.

We would like to thank the editor and the reviewers for their appreciation of our work and their constructive feedback on the manuscript. By taking into account this feedback, our revised manuscript has been considerably improved. Moreover, by clarifying several of the methodological details and providing additional analyses, we shore up the significance of our results. Importantly, none of the original findings are materially changed but rather, further supported. We first discuss some broad points raised by the reviewers and then provide a point by point response, divided into sections corresponding to each reviewer; the reviewer's comments are in black and our response is in blue.

1. Broad points.

a. As suggested by Reviewers #1 and #2, we have removed the connectivity analysis.

b. Reviewer 2 suggests additional data to be collected in order to precisely relate our reported activations to face and voice patches. We argue that, while this is interesting, it is beyond the scope of the present study which seeks to identify the cortical regions involved in the binding of visual and auditory information at large and the rules that govern this binding, independently of a priori functional assumptions.

We nonetheless do agree that relating our findings to the face- and voice-patch literature is one of the follow up experimental questions –amongst others- that will need to be addressed. This is why the last figure of the manuscript (in the discussion section) proposes a match between the localization of our peak activations and the face- and voice-patch described by others as a basis of discussion.

c. Based on the multiple comments of the reviewers, we have also changed the title to: “**Socially meaningful visual context either enhances or inhibits vocalization processing in the macaque brain**”.

2. Reviewer-specific points

Reviewer #1

Froesel and colleagues performed an fMRI study in two monkeys investigating audio-visual integration using socially meaningful stimuli. When monkeys are shown a combination of visual-auditory, and visual and auditory stimuli in isolation in different epochs, the meaning of the stimuli largely determines the fMRI responses in the superior temporal and lateral sulci. Specifically, when the auditory stimulus does not match the visual stimulus, activity is abolished, while matching auditory stimuli activate areas in the STS almost at the same level as the visual stimuli. This is indirect functional evidence showing that the monkeys can link the meaning of auditory and visual stimuli and that this association determines to a large degree the activity in the STS and LS. The fMRI results match with variations in heart rate during the different conditions. The authors also link these association of facial expressions with corresponding vocalizations to two distinct networks, based on a gPPI analysis.

This is a very exciting study, with highly surprising but seemingly robust results. I'm particularly impressed by the dramatic differences in activity evoked by the same auditory stimulus, depending on the context: i.e. congruent or incongruent relative to the emotional information carried by the visual stimulus. Also, the strength of the auditory responses in the superior temporal sulcus (when presented in the right context) are unpredicted. To the best of my knowledge, this is a highly novel finding in macaques that deserves to be published. Of course, this result raises a number of questions that should be addressed in follow-up studies: for example, what would happen if also congruent and incongruent visual stimuli were shown? What is the timeline of these associative-dependent activity changes? What is happening at neuronal level? Etc.

I'm less impressed by the gPPI analysis and I would suggest to drop this altogether. The 'one voxel' criterion (at $P < 0.005$ uncorrected) is impossible to defend, and as such these results weaken rather than strengthen the manuscript. The additional advantage to drop this part would be that the (too)

lengthy discussion can be shortened considerably. In general, the text is sometime difficult to understand and there is room for grammatical improvements.

We would like to thank reviewer 1 for his/her positive appreciation of our work. Along with this reviewer's suggestion, we have now removed the gPPI analysis as well as the related parts in the Abstract and Discussion.

1. It is difficult to understand why primary visual areas are not activated (in the standard visual versus fixation contrasts). In case this part of the brain was not scanned, the authors should indicate exactly what part was covered. Alternatively, provide temporal signal to noise maps, which may reveal differences in sensitivity across the cortex.

All the brain was scanned for this study using an eight-channel phased-array receive coil placed to optimize signal-to-noise ratio (SNR) over the temporal and prefrontal cortex. This has (expectedly) resulted in lower SNR in the occipital cortex, accounting for absence of significant activations in the occipital cortex. We now provide the temporal signal to noise maps (computed with the 3Dtstat -tsnr AFNI function) for the pooled face contexts and the pooled social contexts in Supplemental figure 1, as follows:

Supplemental figure 1. Temporal signal to noise maps for the pooled face contexts (left) and the pooled social contexts (right). Top: inflated left and right hemispheres. Bottom: flat maps of left and right hemispheres.

We refer to this supplemental figure 1 in the methods as follows: "The coils were placed so as to maximize the signal on the temporal and prefrontal cortex. As a result, signal-to-noise was low in the occipital cortex". We also make note of this point in the result section as follows: "Please note that receiving coils were placed so as to optimize temporal and prefrontal cortex signal-to-noise ratio (SNR). As a result, no activations can be seen in the occipital cortex (see temporal SNR maps in Supplemental figure 1)."

2. The heart rate results are in line with the fMRI results, which is interesting. However, this may also pose a problem as changes in heart rate might affect measured fMRI responses in a very complicated manner. For example, there are both positive and negative correlations with BOLD responses at short (6-12 sec) and longer (30-42 sec) time lags, respectively (e.g. Chang et al: Neuroimage. 2009 February 1; 44(3): 857-869.doi:10.1016/j.neuroimage.2008.09.029.). To rule out this unlikely but also more mundane explanation for their results, i.e. that the unexpected differences in fMRI activity are completely explained by differences in heart rate, the authors should add heart rate as an additional regressor (time-shifted) in their fMRI analysis.

This is a very relevant point. Supplemental figure 4 now compares the activations. The new results we present go beyond the previous analyses, as the inhibition of the irrelevant auditory stimulus remains unchanged when including heart rate as regressor of non interest (left) relative to the activations obtained when using only motion and eyes as regressors (initial analysis, right). Please note that

because alignment triggers between heart rate signals and MRI scans were missing for a few sessions, this supplemental analysis is run on fewer runs than those presented in the main paper (Face: 31 runs instead of 40; Social: 67 runs instead of 79).

This Supplemental figure 4 reads as follows:

Supplemental figure 4. Main contrast analyses of figures 2, 3 and 4 reproduced here including (left) or not (right) the heart rate as a regressor of non interest, for the Face (A) and Social (B) runs. Please note that because alignment triggers between heart rate signals and MRI scans are missing for a few sessions, this supplemental analysis is run on fewer runs than those presented in the main paper (Face: 31 runs instead of 40; Social: 67 runs instead of 79).

Supplemental figure 4 is now referred to in the revised manuscript as follows: “Importantly, heart rate changes appear to mirror hemodynamic signal modulation in the identified functional network.

Because changes in heart rate might affect measured fMRI responses (Chang et al., 2009), we re-ran the analyses presented in figures 2, 3 and 4 using heart rate as a regressor of non-interest in addition to head motion and eye position (Supplemental figure S4). Observed activations remained unchanged, thus indicating that the reported activations are not an artefact of changes in heart rate.”

3. There is value to show only group data. The reader must be convinced, however, that the data are not largely driven by one animal. Please address this issue (e.g. by showing conjunction maps of the two animals, or individual maps).

We completely understand this concern. Supplemental figure 2 now shows the main Vi, AC and AI contrasts for both monkeys (S and T). Both maps show the same general patterns as described in Figures 2, 3 and 4 on the group data.

Supplemental figure 2. Individual whole-brain activations for monkey T (left) and monkey S (right), for FACE contexts (F+ & F-, top) and FACE contexts (S1+, S2+ & S1-, S2-, bottom) for the visual (white, Vi vs. Fx), auditory congruent (green, AC vs. Fx) and auditory incongruent (red, AI vs. Fx) contrasts. Darker shades of red indicate level of significance at $p < 0.001$ uncorrected, t-score 3.09. Lighter shades of

yellow and brown outlines indicate level of significance at $p < 0.05$ FWE, t-score 4.6. ios: Inferior Occipital Sulcus; LS: Lateral Sulcus; STS: Superior Temporal Sulcus; CiS: Cingulate Sulcus; LuS: Lunate Sulcus; OrbS: Orbital Sulcus.

This supplemental Figure 2 is now referred to in the main text as follows:

“Figure 2: Whole-brain activation FACE contexts (F+ & F-): main contrasts. Whole-brain activation maps of the F+ (face affiliative) and F- (face aggressive) runs, cumulated over both monkeys, for the visual (white, Vi vs. Fx), auditory congruent (green, AC vs. Fx) and auditory incongruent (red, AI vs. Fx). Note that the AC and AI conditions contain exactly the same sound samples (coos and aggressive calls). Darker shades of red indicate level of significance at $p < 0.001$ uncorrected, t-score 3.09. Lighter shades of yellow and brown outlines indicate level of significance at $p < 0.05$ FWE, t-score 4.6. ios: Inferior Occipital Sulcus; LS: Lateral Sulcus; STS: Superior Temporal Sulcus; CiS: Cingulate Sulcus; LuS: Lunate Sulcus; OrbS: Orbital Sulcus. See supplemental figure S2 for individual monkey data. »

“Combining the F+ and F- face contexts (Figure 2, see supplemental figure S2 for individual monkey data), which includes faces expressing lipsmacks or aggressive threats, [...] In contrast, when we present the exact same aggressive calls and coos, the incongruent auditory versus fixation contrast leads to minimal activation, if any (Figure 2, see supplemental figure S2 for individual monkey data).”

“These observations are reproduced in a different set of contexts, in which the visual stimuli involve social scenes (grooming, aggression or escape) with either semantically congruent or incongruent vocalizations (Figure 4 for all social contexts on group data, Figure S3 for S+ and S- social contexts independently on group data and supplemental figure S2 for all social contexts on individual monkey data).”

4. The dark green and dark red fonts on a black background should be avoided. These are difficult to read.

We revised the figures containing dark green and red to light green and red.

5. The flattened illustrations of the cortex are very different from what is usually shown. Therefore it is difficult ‘to read them’. I would certainly add the operculum (which seems to be largely removed). I would also label all sulci.

We changed the figures to make it more similar to classical representation, by changing their orientation. We also reprocessed our anatomical images to recover the operculum. This was an oversight. We additionally labelled all sulci except for figure 8 due to the small size of the brains in this figure.

6. The insets (indicating anterior / ventral etc..) in figures are not always correct.

We have corrected these errors.

7. Fig 6A: the orientation of the inflated brain and corresponding flatmaps are different. Please correct, as it hampers readability

We changed the orientation of the flatmap and agree that it makes the figure more readily interpretable.

Minor:

1. line 222: 'The STS appears equally responsive to visual and auditory stimuli.' But only statistics are provided for V vs Fix and A vs Fix. Please test directly V vs A to back up this statement.

The text now reads as: "... while STS appears to be equally responsive to visual and auditory stimuli ($p > 0.05$)."

Figure 7 has also been changed to indicate non-significant differences as n.s.

2. As mentioned above, I would drop the PPI analysis. Please note that on Page 14, the authors wrongly refer to figure 8 instead of 9.

We followed the reviewer's suggestion and removed the PPI analysis from results and the discussion.

3. Line 405: PITd is located near the PMTS. The gaze following patch is located more dorsally (Marciniak et al. eLife 2014;3:e03222. DOI: 10.7554/eLife.0322)

Thank you for this expert feedback. The text has now been modified as follows: "In the right hemisphere, two supplementary STS activations are reported, STS 2 and STS 3. They are located posteriorly to the putative ML face patch and ventral to the gaze following patch reported in the dorsal posterior infero-temporal cortex (Marciniak et al., 2014) and possibly coincide with an area in the middle superior temporal cortex that has been recently described as modulated by the predictability of social interactions (Roumazeilles et al., 2021), though this would have to be tested explicitly."

4. Line 414: It is false to state that auditory responses have not yet been described in the STS. There are quite a few electrophysiology and fMRI papers in the monkey showing the opposite.

We acknowledge that we overstated this point in the original submission. The text has now been modified as follows: “Congruent auditory information results in enhanced cortical activations relative to previous reports. Indeed, auditory activations have already been described in the STS (Barraclough* et al., 2005; Ghazanfar et al., 2008; Joly, Pallier, et al., 2012; Joly, Ramus, et al., 2012; Ortiz-Rios et al., 2015; Perrodin et al., 2014; Poremba et al., 2003). However, and specific to our task design, the STS auditory activations described here in response to the congruent auditory stimuli are as strong as the visual responses and extend into the extra-striate visual cortex, thus suggesting cross-modal enhancement.”

Reviewer #2

In their manuscript “Neural bases of audio-visual integration of socially meaningful information in macaques”, Froesel et al. use fMRI in macaque monkeys to assess the impact of audio-visual congruency versus incongruency in the social domain on brain activity and heart rate modulation. They report large changes of activity in the superior temporal sulcus (STS) and the lateral sulcus (LS) due to congruence and incongruence. The findings are entirely unexpected in sign, strength, and spatial extent. The authors then go on to determine functional connectivity of these two regions with other brain regions and find networks (with low significance) that are similar for STS and LS. The main results, if true, are so astonishing, they would be some of the most remarkable findings in cognitive and systems neuroscience in the last thirty years. Figure 4 and others are suggesting, and Figure 6 shows directly the claim that areas throughout the STS show a response to a congruent auditory stimulus comparable to the visual stimulus in isolation, while an incongruent auditory stimulus hardly elicits any response at all – and a very similar pattern exists in the lateral sulcus. Considering that many areas in these analyses are classical visual or auditory areas, a result so expansive across these large regions and so strong is unexpected. Consider further that results on predictive coding would suggest the opposite pattern of results, a bigger response of the incongruent stimulus. So, there are three reasons to be really astonished by the results. To make as strong claims as the paper, maybe unwittingly, does requires very thorough experimental design and controls. It is here where the paper falls severely short.

We agree with reviewer #2 that our findings are quite unexpected, hence our effort to reproduce our observations in multiple tasks and independently in two animals (see below, point 5). In addition, we provide below an extensive response to the methodological questions raised by this reviewer. The point raised by reviewer #2 relative to predictive coding is a quite interesting and important point, and a research project in itself, which we believe would be crucial to address.

1. Furthermore, the description of procedures and results is often unclear and unprecise, thus making it very difficult to fully understand what was done. I thus apologize up front, if I got something completely wrong.

My main concerns regard experimental design and lack of controls. The authors suggest that the blocks within a run were varied following three pseudo-random sequences. (I cannot have more certainty, because descriptions of what the sequence is vary at different places of the manuscript. In Methods they state that there were “three repetitions of a pseudo-randomized sequence containing six possible 16 s blocks”. It is unclear why the blocks are called possible. The figure suggests that only the sequence of blocks was varied. “blocks randomization”, a term used in Fig. 1, might also indicate randomization of image presentation within a block. Such lack of clarity in important descriptions are pervasive throughout the manuscript.) The first problem with the design is that one would expect a fully counter-balanced design, especially when blocks follow each other. This however has not been done here.

We apologize for the lack of clarity perceived by reviewer #2 and for the general misunderstanding that resulted from this. We acknowledge that important information was not made explicit. We have now made several changes in the manuscript to address the doubts and questions of this reviewer. In summary, our experimental design precisely addresses the concerns of reviewer #2, thus fully supporting the results presented in the manuscript.

Methods: “General run design. On each run, monkeys were required to fixate a central cross on the screen (Figure 1A). Runs followed a block design. Each run started with 10 s of fixation in the absence of sensory stimulation followed by three repetitions of a pseudo-randomized sequence containing six 16 s blocks: fixation (Fx), visual (Vi), auditory congruent (AC), auditory incongruent (AI), congruent audio-visual (VAC) and incongruent audio-visual (VAI) (Figure 1A).

The pseudo-randomization was implemented such that each block in each repetition was presented in a randomized order. Thus monkeys could not anticipate the sequence of stimuli. In addition, the initial blocks were either a visual block (Vi, VAC, VAI), or a fixation block followed by a visual block (Vi, VAC or VAI), such that context was set by visual information early on in each run. As a result, pure auditory blocks were always presented after a visual block and could thus be defined as congruent or incongruent to the visual information characterizing the block. Pseudo-randomization was also implemented such that, across all repetitions and all runs for a given context, each block was, on average, preceded by the same number of blocks from the other conditions. Quite crucially to the results presented in this work, in 66% of the times, both AI and AC conditions were preceded by blocks involving visual stimulation (Vi, VAC and VAI). Last, each block (except the fixation block) consisted in an alternation of 500 ms stimuli (except for lipsmacks, 1s dynamic stimuli succession) of the same semantic category (see Stimuli section below), in the visual, auditory or audio-visual modalities. In each block, 32 stimuli were presented randomly (16 for lipsmack). Each run ended by 10 s of fixation in the absence of any sensory stimulations.

Face and social contexts

Six audio-visual contexts were presented to both monkeys, organized in runs as described above (Figure 1A). Each context combined visual stimuli of identical social content with either semantically congruent or incongruent monkey vocalizations with the predominant visual stimuli (Figure 1B). Runs always started by a block condition involving visual stimulation, thus setting the social context of the task and, as a result, defining auditory congruent and incongruent auditory stimuli. Given the structure of our task, two levels of congruency can be defined. A first order congruency is defined within the audio-visual blocks, such that the auditory information can either be congruent (VAC) or incongruent (VAI) to the visual information. The second order of congruence is defined at the level of the run, such that, given the visual information presented in a given run, the pure auditory blocks can either be defined as congruent (AC) or incongruent (AI) in this specific run, even if not simultaneously presented with the visual information. The face affiliative context (F+) combined lipsmacks with coos and aggressive calls. The face aggressive context (F-) combined aggressive faces with coos and aggressive calls. The first social affiliative context (S1+) combined grooming scenes with coos and aggressive calls. The second social affiliative context (S2+) combined grooming scenes with coos and screams. The social aggressive context (S1-) combined aggressive group or individual scenes with coos and aggressive calls. The social escape context (S2-) combined fleeing groups or individual scenes with coos and screams. Importantly, pairs of contexts (F+ & F-; S1+ & S1-; S2+ & S2-) shared the same vocalizations, but opposite social visual content (i.e. opposite semantic content, defining either a positive or a negative context). All contexts were presented randomly and at least once during each scanning session.”

Results: “In contrast, when we present the exact same aggressive calls and coos, the incongruent auditory versus fixation contrast leads to minimal activation, if any (Figure 2, see Supplemental figure S2 for individual monkey data). Again, this doesn’t depend on whether the incongruent sounds are aggressive calls (Figure 3D) or coos (Figure 3E). This pattern of activation therefore confirms that auditory activation does not depend on the nature of the vocalization. Rather, it depends on whether

the vocalizations are congruent or not to the semantic content of the visual stimuli. It is worth noting that, in 66% of the instances, both AI and AC conditions are preceded by blocks involving visual stimulation (Vi, VAC and VAI). Because this was the case for both AI and AC conditions, the absence of auditory activations in the AI vs. F contrast and the presence of temporal and occipital activations in the AC vs. F contrast cannot be interpreted as a trace of the activations resulting from the previous blocks, but rather, they have to be considered as a genuine phenomenon resulting from the structure of the task.”

In addition, we modified figure 1A so as to introduce examples of runs illustrating the block pseudo-randomization used.

Figure 1: A) Experimental design. Example of an aggressive face (F-) context. Each run was composed of three randomized repetitions of six different blocks of 16 seconds. The six blocks could be either visual stimuli only (Vi), auditory congruent stimuli only (AC), auditory incongruent stimuli only (AI), audio-visual congruent stimuli (VAC) or audio-visual incongruent stimuli (VAI), or fixation with no sensory stimulation (Fx). Block presentation was pseudo-randomized and counter-balanced so that, across all repetitions and all runs of given context, each block was, on average, preceded by the same number of blocks from the other conditions. Initial blocks were either a visual block (Vi, VAC, VAI), or a fixation block followed by a visual block (Vi, VAC or VAI), such that context was set by visual information early on in each run. Each sensory stimulation block contained a rapid succession of 500ms stimuli. Each run started and ended with 10 seconds of fixation. **B) Description of contexts.** Six different contexts were used. Each context combined visual stimuli of identical social content with either semantically congruent or incongruent monkey vocalizations. Pairs contexts shared the same auditory stimuli, but opposite social visual content (F+ vs. F-; S1+ vs. S1-; S2+ vs. S2-).

2. This is particularly problematic, when 16s blocks are used after administration of the contrast agent Molday ION which essentially doubles the time constant relative to the BOLD response. Thus the response to each block massively “leaks over” into the response of the next one, and one has to make strong assumptions of linearity to be able to tell them apart.

As specified in the material and methods, we used an HRF that is specific to the MION in our GLM analysis. This is a standard method used by non-human primate imaging labs that inject MION contrast agents to enhance SNR (Bodin et al., 2021; Clery et al., 2015; Guipponi, Odouard, et al., 2015; Leite et al., 2002; Moeller et al., 2008; Vanduffel et al., 2001). Using such a MION-specific HRF is designed to compensate for the slower dynamics of the MION response relative to the BOLD responses. Please note that given the pseudo-randomized counterbalanced structure of our design, the use of a suboptimal HRF would have induced noise in the results rather than such clear cut differences between the AC vs. Fx and AI vs. Fx contrasts as reported here.

More generally, the greater amplitude of MION response relative to BOLD has been consistently shown to outweigh the contrast loss caused by greater temporal smoothing due to its slower temporal dynamics (Leite et al., 2002). This has been shown for both blocked-designs and event-related designs. Indeed, Vanduffel et al., (2001) comparing, in awake macaque monkeys, the use of BOLD vs. MION in block-design experiments, in order to map the motion selective brain areas, show greater spatial localization and contrast increase in MION relative to BOLD. Such block-designs combined with MION have been extensively used in non-human primate imaging by us (Clery et al., 2015; J. Cléry et al., 2018; Guipponi, Cléry, et al., 2015) and others (Bell et al., 2009; Taubert et al., 2015; Tsao et al., 2003, 2008). A recent study by Pelekanos et al., (2020) generalizes this observation to event-related non-human primate fMRI experiments, revealing the absence of effect of speed or temporal dimensions of the experimental design.

Lastly, in our response to the previous point, we clarify that our design involves a pseudo-randomization and a counterbalancing of the conditions within the blocks, such that, 1) each condition is, on average, preceded by exactly the same number of presentations of the other conditions (thus leakage from the previous block has an identical statistical structure for all blocked conditions) and 2) AI and AC conditions are preceded in 66% of the cases by a block involving visual stimulation (thus leakage from the previous block if any would be identical between both AC and AI conditions, and activations would be closely matching the visual activations, which is definitively not the case—figures 2, 3 and 4).

Taken together, these results rule out the reviewer's suggestion concerning leakage, and supports the validity of our experimental design, analyses and results.

3. The second problem with this procedure and lack of documentation of the three actual sequences is that it is not clear how congruence and incongruence were actually implemented. For an auditory stimulus presented in isolation to qualify as either congruent or incongruent, the visual stimuli must be presented before. One can only assume that the pseudorandom sequences were pseudo in the sense that this was guaranteed.

Yes, indeed, this was guaranteed. We apologize for the lack of clarity perceived by reviewer #2 and for the general misunderstanding that resulted from this and we do acknowledge that important information was not made explicit. As explained in our response to reviewer #2, point 1, the initial blocks of each run were either a visual block (Vi, VAC, VAI), or a fixation block followed by a visual block (Vi, VAC or VAI), such that context was set by visual information early on in each run. As a result, pure auditory blocks were always presented after a visual block and could thus be defined as congruent or incongruent to the visual information characterizing any given run.

However, the example in Fig. 1 shows an incongruent audio-visual block leading.

Contrary to what reviewer #2 assumes, context is not uniquely set by Vi blocks but also by VAC and VAI blocks, as they all share the same visual stimuli and monkeys have prior knowledge that in a given

run, visual stimuli will always be of the same category. As a result, the initial VAI condition presented in Figure 1A does set the context of the run.

Changes to the main text are described in our response to reviewer #2, point 1.

Assuming that things were done this way, note then how radical a statement of congruence or incongruence might be: it might take half a minute for the incongruent auditory stimulus to be displayed after the visual condition, and one has to assume for the subject's brain to one prioritize the visual information as context for the auditory information and not vice versa, and for the brain to be able to differentiate between a congruent and an incongruent auditory condition after the passage of such a long period of time. One would expect a decline over time, which is not documented. But if it occurred, it would be hard to see how the response to congruent and incongruent auditory stimuli could come out as different as they do on average.

Given the structure of our task, two levels of congruency can be defined. A first order congruency is defined within the audio-visual blocks, such that the auditory information can either be congruent (VAC) or incongruent (VAI) to the visual information. The second order of congruence is defined at the level of the run, such that, given the visual information presented in a given run, the pure auditory blocks can either be defined as congruent (AC) or incongruent (AI) in this specific run, even if not simultaneously presented with the visual information.

Reviewer #2 assumes in his argumentation that monkeys are unaware of the second order congruency embedded in the task. The cardiac data presented in figure 5 speaks against this. Indeed, while average heart rates differ between tasks, a systematic statistically significant higher heart rate is observed on incongruent auditory blocks relative to congruent auditory blocks, irrespective of the actual identity of the auditory stimuli. This strongly supports the interpretation that monkeys prioritize visual information as context and not vice-versa. Taken together, these results speak against the possibility that the encoding of context would decline overtime, as by definition, the second order congruency defining the context applies to the entire run.

We hope our clarification of the experimental details satisfactorily addresses reviewer 2's methodological concerns.

4. The authors find consistent results across multiple regions of interest (ROIs) in STS and LS. Leaving the problem of double-dipping in the selection of ROIs aside, there are huge differences in functional specialization within these regions. Take the face areas for example the authors are frequently referring to. These occupy small regions within the STS. Using the contrast agent the authors are using and face stimuli, one would expect much larger percent signal changes in these regions than the author report. It is possible that maybe coils were not positioned properly and thus the SNR was so much lower than expected.

Reviewer #2 raises two points here.

The first pertains to the percent signal change that reviewer #2 finds weaker than expected. Supplemental figure S1 now shows the temporal SNR maps and clearly indicates that the coils were properly positioned to maximize SNR over the temporal, parietal and prefrontal cortex as explained in the methods section. Additionally, the reported %PSC in figure 6 for the Visual contrast in the face context ranges between 0.7 and 1.6%. The reported %PSC for the auditory congruent contrast ranges between 0.6 and 1.4%. This is in the range of the %PSC reported by others with similar stimuli: STS visual face stimuli (range 0.85-1.75 in the face patches of figure S8 in Tsao et al., 2008) and LS auditory voice stimuli (range ~0.6-1.6 in the lateral sulcus, figure 2, in Ortiz-Rios et al., 2015).

The second point pertains to double dipping. Figure 6 is not used to confirm the fact that the auditory incongruent condition is the less responsive condition as already illustrated in figures 2, 3 and 4. Rather, its aim is to compare the response patterns across selected ROIs and across the LS and the STS, using appropriate statistical tests. As a result, this does not fall in the category of double dipping.

5. It would be essential for this paper to localize areas of functional specialization – for the visual and the auditory stimuli used – and define ROIs based on these. With these independently defined ROIs, it would then be possible to make a firm statement and say that face area x responds this much to faces in isolation and without context this much to vocalizations in isolation and without context, and then, in the main experiment we find the following pattern. It would then be possible to properly judge the new results in the context of established ones. Currently, one sees large swathes of cortex activated, but one does not know which regions they correspond to, one does not know how reproducible results are. And it does not seem to matter to the authors who then lump them together to talk about the STS and LS, which are internally so heterogenous.

The main result of this paper is the observation that when a context is defined by social visual stimuli, incongruent social auditory stimuli are “extinguished” while congruent social auditory stimuli are not, and that this takes place at all stages of sensory processing and not just at higher levels.

While we do agree with reviewer #2 that it is an important scientific question to relate our findings with pre-existing scientific literature on face and voice processing (hence our figure 9 and our discussion section on this specific point), we strongly believe that identifying face patches and voice patches in these very same subjects will not change the impact of our finding, for the following reasons:

- 1- The main result of this paper is the observation that when a context is defined by social visual stimuli, incongruent social auditory stimuli are “extinguished” while congruent social auditory stimuli are not, and that this takes place at all stages of sensory processing and not just at higher levels. This finding raises crucial questions in terms of audio-visual processing in general and not just specifically in terms of vocal communication. This is the point that we highlight in the discussion and which reviewer #3 actually asks us to further emphasize.
- 2- Several studies have sought to identify and characterize voice patches using fMRI in monkeys. These studies, taken together, allow a better understanding of the voice patch system, although taken individually, these studies do not always report all identified voice patches. The same applies to face fields, although these are much better characterized both using fMRI and single cell recordings. Remarkably, we have been able to propose (Figure 9) correspondences between our auditory and visual activations and the main voice and face patches reported in the literature, suggesting that our specific experimental design recruits this network in a way that is not recruited by unimodal studies. Both reviewers #1 and #3, did not disagree with our matching result and reviewer #1 provided specific feedback on one matching result indicating that he/she has strong expertise on the matter.
- 3- Anticipating the interest that some readers might have in relating our activations to face and voice patches, we have produced a figure that specifically assigns our peak activations to previous findings, while addressing the putative role of face and voice patches in the discussion.
- 4- Last, as discussed in a recent collaborative paper from our group and major figures of non-human primate imaging (Russ et al., 2021), localizers are an important tool in non-human primate neuroimaging. However, exclusively relying on them might stand in the way of a new understanding of brain functions.

Given the comments above, we disagree that collecting more data to map the voice and face patches is necessary to understand the results we present. We nonetheless agree that this is an important area for further research.

Reviewer 2 also raises questions about data reproducibility, which we address here.

In the present manuscript, we used 6 different tasks. As illustrated in figure 1b, the exact same auditory stimuli are used in the pairs of tasks: (F+ and F-), (S1+ and S1-) and (S2+ and S2-). As a result, when analysing these tasks as a function of congruence, we are actually contrasting conditions with exactly the same auditory content, only the context set by visual information varying across conditions. In spite of this, we show:

1. Reproducibility across tasks: Our observations are reproduced both on the two face tasks and on the four social tasks. This degree of robustness is highlighted by reviewer #1.
2. Reproducibility across animals: We additionally produce a Supplemental figure S2 showing that our observations are not driven by one subject and are thus reproducible across both subjects.

This level of reproducibility is, we believe, fairly unusual in the field of primate neurophysiology.

Regarding lumping data together, we do not doubt that our ROIs are functionally heterogeneous, as addressed in the discussion section on the face and voice patches. What authorizes us to pool them in a unique analysis, is actually a statistical test that indicates that there is no ROI effect on the %PSC across LS ROIs (resp. STS ROIs), while there is a significant condition and a significant sulcus effect. This brings the interesting observation that while the LS responds preferentially to the auditory congruent condition and less so to the visual condition, STS is equally activated by both— an unexpected and functionally interesting result). While again we do agree that there would be immense value in understanding the specific contribution of each ROI to the task of interest, not only is this beyond the scope of the present work, but the statistical assessment speaks against this. We identify this as a limitation of fMRI, with other investigation methods being more appropriate to address this question.

Last, and quite importantly, we show in this study an inhibition of irrelevant auditory information as a function of the context set by visual information. This process of filtering incongruent social auditory stimuli relative to social visual stimuli has actually already been shown at the behavioural level. Specifically, human adults are shown to reliably filter out irrelevant social auditory information as a function of visual information while children below age 11years found this more challenging (Ross et al., 2021). This was even more marked for children below age 7. This capacity to filter irrelevant information observed in adults is thought to arise from cross-modal inhibition. Such cross-modal inhibition has for example been described in the auditory cortex in response to visual and auditory stimuli presented simultaneously. Importantly, such a cross-modal inhibition has been shown to switch on or off as a function of the context (Laurienti et al., 2002). Accordingly, functional interactions between the visual and auditory networks can either result in an enhancement or in a suppression of cortical activity depending on the task and the presented stimuli (Lewis et al., 2000). The results we present here go beyond these early observations, as the inhibition of the irrelevant auditory stimulus does not take place at the time of presentation of the visual stimulus but when presented on its own, as the context is not set on a single trial basis but rather in well segregated behavioural blocks. We hypothesize that our observations rely on a generalized form of cross-modal inhibition. This will have to be tested experimentally. This paragraph is now introduced as a new discussion point in the main paper.

Reviewer 3#:

This is an interesting study on the representation of species-specific communication calls in the macaque using functional MRI. The authors use a battery of calls, recorded in the wild by one of the co-authors (Marc Hauser, a well-known neuroethologist with expertise in communication calls and behavior of macaques). Each type of calls is uttered in specific situations or contexts, which can be affiliative (e.g. coos) or aggressive (e.g. growls). In their fMRI study, the authors play back the calls together with varying visual displays (faces or scenes) that are either congruent or incongruent with the meaning of the call. The initial set of scans (Figs. 1-4) yields very clear and impressive results, in that the combination of calls and incongruent visual displays leads to an almost complete suppression of the brain activation in auditory regions (lateral sulcus [LS] and superior temporal sulcus [STS]). Neurophysiologically, one would refer to these effects as crossmodal inhibition. Although crossmodal inhibition or suppression has been demonstrated in various other contexts, this is a novel set of circumstances and definitely worthy of reporting. The authors then go beyond the immediate auditory-visual interactions and measure psychophysiological correlates in terms of heart

rate changes (Fig. 5), using a device that was developed in their lab. This is valuable quantitative information at the behavioral level complementing purely functional observation.

In an extensive third part, the study tries to explain the effects measured with fMRI on the basis of connectivity with other brain regions. The region discussed first and foremost is prefrontal cortex, which is known to exert top-down modulation on lower-order sensory regions. It is also claimed to mediate emotional modulation (together with orbito-frontal cortex and parts of the limbic system). Second is parietal cortex in terms of attentional modulation. These two networks are then discussed in wide terms and in the context of "face and voice patches" (Fig. 10), dorsal and ventral streams, and social and emotional modulation.

We would like to thank reviewer 3 for his/her positive appreciation of our work.

1. While this is a valuable and quite novel part of the study, I would strongly suggest that the results be discussed by comparison with classical neuroanatomical tracer studies. The work of Goldman-Rakic or of Joseph Price comes to mind. STS and its connections in particular have been studied quite extensively by Seltzer and Pandya 40 years ago and tends to be forgotten. I consider it a major and worthwhile task for today's imaging community to validate their results by relating them to those of classical anatomical tracer studies – across and within the same primate species.

While we completely take the reviewer's recommendation that today's neuroimaging community should validate their results by relating them to those of classical anatomical tracer studies, the last part of the results measuring psychophysiological correlates and the corresponding discussion have been removed from the new version of the manuscript at the request of reviewers' #1 and #2 as well as the editor.

We however did integrate the rich classical neuroanatomical tracer studies into the discussion as follows: "An important question is how context is implemented into LS and STS. The STS is involved in multisensory integration and is shown to play a modulatory role on lateral sulcus functions during audio-visual stimulations (Barraclough* et al., 2005; Ghazanfar et al., 2005; Perrodin et al., 2014). However, the mechanisms subserving the observed selective cross modal inhibition of auditory processing based on the visual context are implemented not just during audio-visual blocks but throughout any given run. As a result, they are expected to originate from a higher order cortical region exerting a top-down control on both the LS and the STS. The prefrontal cortex is a choice region in this respect, as it connects to LS (Rauschecker & Tian, 2000; Romanski et al., 1999; Saleem et al., 2014) and STS (Seltzer & Pandya, 1989, 1994) and has been shown to play a crucial role in working memory and the top-down modulation of perception (Fuster, 2002; Goldman-Rakic, 1996). LS and STS are also connected to the cingulate cortex and orbitofrontal cortex (Kondo et al., 2003; Saleem et al., 2008). These cortical regions that are involved in the processing of social interactions from visual cues and are thus in a position to provide feedback to the LS and STS based on the social dimension of the stimuli (Cléry et al., 2021; Roberts, 2006; Rudebeck et al., 2006; Rushworth et al., 2007; Sliwa & Freiwald, 2017). Lastly, LS and STS are also connected to the limbic system (Amaral & Price, 1984; Kondo et al., 2003; Saleem et al., 2008). Accordingly, conspecific vocalisations activate a network recruiting, in addition to the voice patches, visual areas such as V4, MT, STS areas TE and TEO, as well as areas from the limbic and paralimbic system, including the hippocampus, the amygdala and the ventromedial prefrontal cortex (vmPFC) (Gil-da-Costa et al., 2004). All of these regions are expected to contribute (most probably in coordination) to setting the context based on what auditory information is considered either as congruent or incongruent. This will have to be addressed experimentally."

This latter part of the Results and the Discussion threaten to become a bit of a hodgepodge of different ideas and seem to be driven by trendy topics, like face and voice patches or 'social neuroscience' in general. While vocal communication undoubtedly includes much social interaction, this should not be confused with the mechanistic aspects of visual-auditory interactions, all the way down to the single-unit level. These two aspects are distinct from one another and should be clearly kept apart and discussed as two distinct topics.

We thank reviewer #3 for this insightful comment. We have significantly extended our discussion on multisensory integration and audio-visual association and reordered the discussion such that this section is addressed before the discussion on face and voice patches. This section now reads as follows: **“Audio-visual association based on meaning and multisensory integration**

The strict definition of multisensory integration involves the combination of sensory inputs from different modalities under the assumption of a common source (Lee & Noppeney, 2014; Stein et al., 2014). In this context, it has been shown that multisensory integration speeds up reaction times and enhances perception (Grant & Seitz, 2000; Lehmann & Murray, 2005; Murray et al., 2005; Raab, 1962; Welch et al., 1986), including when processing lip movement during speech (Navarra & Soto-Faraco, 2007; Shahin & Miller, 2009; Van Wassenhove et al., 2005). Multisensory processes are also at play to predict the consequences of one modality onto another, i.e. in the temporal domain (Cléry et al., 2015, 2017; Cléry et al., 2020; Guipponi et al., 2015). At the neuronal level, multisensory integration is defined as a process whereby the neuronal response to two sensory inputs is different from the sum of the neuronal responses to each on its own (Avillac et al., 2007; Stein et al., 2009). In the present study, the auditory and visual stimuli are associated based on their meaning (e.g., coos are associated with grooming) and possible contingency (e.g., screams are associated with escape scenes). Thus the audio-visual association described here goes beyond the strict definition of two sensory inputs produced by a common source.

Additionally, by task design, two levels of audio-visual congruency can be defined: 1) a first order congruency, defined within the audio-visual blocks, such that the auditory information can either be congruent (VAC) or incongruent (VAI) to the visual information; 2) a second order congruency, defined at the level of the run, such that, given the visual information presented in a given run, the pure auditory blocks can either be defined as congruent (AC) or incongruent (AI) to the general visual context of this specific run, even if not simultaneously presented with the visual information. In order to probe whether first order congruency gives rise to multisensory integration, we apply the less stringent multisensory integration criteria used in fMRI studies, testing if audio-visual responses are statistically higher (or lower) than each of the uni-sensory conditions (Beauchamp, 2005; Gentile et al., 2010; Pollick et al., 2011; Tyll et al., 2013; Werner & Noppeney, 2010). Although face-voice integration has been described in the auditory cortex (CL, CM, in awake and anesthetized monkeys; A1 only in awake monkeys) and the STS (Ghazanfar et al., 2008; Perrodin et al., 2015), and to a lesser extent in specific face-patches (Khandhadia et al., 2021), here, enhancement of the audio-visual response can only be seen in the contexts involving visual scenes. The parsimonious interpretation of these observations is that face-vocalization binding was easier than scene-vocalization binding. This resulted in increased integrative processes, specifically in this latter condition, in agreement with the fact that neuronal multisensory integration is more pronounced for low saliency stimuli.

Second order congruency is set by the visual information defining a given experimental run and results in major differences in how congruent and incongruent sounds are processed including in the absence of any visual stimulation. Congruent auditory information results in enhanced cortical activations relative to previous reports. Indeed, auditory activations have already been described in the STS (Barraclough* et al., 2005; Ghazanfar et al., 2008; Joly, Pallier, et al., 2012; Joly, Ramus, et al., 2012; Ortiz-Rios et al., 2015; Perrodin et al., 2014; Poremba et al., 2003). However, and specific to our task design, the STS auditory activations described here in response to the congruent auditory stimuli are as strong as the visual responses and extend into the extra-striate visual cortex, thus suggesting cross-modal enhancement. In contrast, we show in this study an inhibition of irrelevant auditory information as a function of the context set by visual information. This process of filtering incongruent social auditory stimuli relative to social visual stimuli has actually already been shown at the behavioural level. Specifically, adults are shown to reliably filter out irrelevant social auditory information as a function of visual information while children below age 11 found this more challenging (Ross et al., 2021). This was even more marked for children below age 7. This capacity to filter irrelevant information observed in adults is thought to arise from cross modal inhibition. Such cross modal inhibition has for

example been described in the auditory cortex in response to visual and auditory stimuli presented simultaneously. Importantly, such a cross-modal inhibition has been shown to switch on or off as a function of the context (Laurienti et al., 2002). Accordingly, functional interactions between the visual and auditory networks can either result in an enhancement or in a suppression of cortical activity depending on the task and the presented stimuli (Lewis et al., 2000). The results we present here go beyond these early observations, as the inhibition of the irrelevant auditory stimulus not only takes place at the time of presentation of the visual stimulus but also when presented on its own, as the context is not set on a single trial basis but rather in well segregated behavioural runs. We hypothesize that our observations rely on a generalized form of cross-modal inhibition. This will have to be tested experimentally.

As discussed above, a specificity of our task design is that it creates, within each run, an implicit association between a set of social visual stimuli and their auditory match, possibly based on past learned sensori-motor associations and the development of internal models of what vocalizations are produced in a given visual context. This is very reminiscent of the recent description of auditory fMRI activations to learned sound sequences in the motor cortex of the macaque brain (Archakov et al., 2020). These auditory responses were only present in monkeys who had received an audio-motor training and were only present in response to the learned sound and were absent for other sounds. The authors propose that an internal model of auditory perception associating a given auditory set of stimuli with a given motor repertoire (and thus motor structure) was created by the training. We here argue that likewise, our current observations arise from the fact that macaques have, throughout their lifespan, associated specific macaque calls with specific social visual experiences, and that our specific task design allows to reveal this internal model.

An important question is how context is implemented into LS and STS. The STS is involved in multisensory integration and is shown to play a modulatory role on lateral sulcus functions during audio-visual stimulations (Barraclough* et al., 2005; Ghazanfar et al., 2005; Perrodin et al., 2014). However, the mechanisms subserving the observed selective cross modal inhibition of auditory processing based on the visual context are implemented not just during audio-visual blocks but all throughout any given run. As a result, they are expected to originate from a higher order cortical region exerting a top-down control on both the LS and the STS. The prefrontal cortex is a choice region in this respect, as it connects to LS (Rauschecker & Tian, 2000; Romanski et al., 1999; Saleem et al., 2014) and STS (Seltzer & Pandya, 1989, 1994) and has been shown to play a crucial role in working memory and the top-down modulation of perception (Fuster, 2002; Goldman-Rakic, 1996). LS and STS are also connected to the cingulate cortex and orbitofrontal cortex (Kondo et al., 2003; Saleem et al., 2008). These cortical regions that are involved in the processing of social interactions from visual cues and are thus in a position to provide feedbacks to the LS and STS on the social dimension of the stimuli (Cléry et al., 2021; Roberts, 2006; Rudebeck et al., 2006; Rushworth et al., 2007; Sliwa & Freiwald, 2017). Lastly, LS and STS are also connected to the limbic system (Amaral & Price, 1984; Kondo et al., 2003; Saleem et al., 2008). Accordingly, conspecific vocalisations activate a network recruiting, in addition to the voice patches, visual areas such as V4, MT, STS areas TE and TEO, as well as areas from the limbic and paralimbic system, including the hippocampus, the amygdala and the ventromedial prefrontal cortex (vmPFC) (Gil-da-Costa et al., 2004). All of these regions are expected to contribute (most probably in coordination) to setting the context based on which auditory information is considered either as congruent or incongruent. This will have to be addressed experimentally.”

We also changed the abstract as follows: “Social interactions rely on the interpretation of semantic and emotional information, often from multiple sensory modalities. In primates, both audition and vision serve the interpretation of communicative signals. The neural mechanisms subserving the perception of social auditory information based on a general context set by visual information are unknown. Based on heart rate estimates and functional neuroimaging, results indicate that macaque monkeys show an enhanced response to vocalizations that are congruently associated with a given visual context, while suppressing the response to vocalizations that are incongruent, including during

unimodal presentations. Affiliative vocalizations are enhanced during affiliative contexts, aggressive vocalizations are enhanced during aggressive contexts, and screams are enhanced during escape contexts. Affiliative vocalizations are inhibited in aggressive and escape contexts as well as aggressive calls and screams in affiliative contexts. Overall, this demonstrates that auditory processing is highly modulated by top-down visual contextual information. This cross-modal modulation does not require sensory simultaneity between the auditory and the visual information.”

3. In summary, this is a heroic study that not only tries to measure visual-auditory interactions in vocal communication of nonhuman primates (which it does very nicely). By considering vocal communication as one form of social interaction, the study expands into other topics in social neuroscience, setting up top-down pathways that ultimately encompass the entire brain. It would be desirable to relate the results of functional connectivity from imaging to classical results of anatomical tracer studies. The functions of prefrontal and parietal cortex, seen here as crucial for top-down modulation, should be more clearly differentiated according to established neuroanatomical data and theory (e.g. Fuster). Otherwise the paper is superb.

Although the discussion pertaining to the gPPI analysis has been removed in our revised manuscript, we nonetheless discuss putative top-down modulators of the cortical areas highlighted in the study, including the frontal and parietal cortex as follows (already cited in response to point 1 of reviewer #3): “An important question is how context is implemented into LS and STS. The STS is involved in multisensory integration and is shown to play a modulatory role on lateral sulcus functions during audio-visual stimulations (Barraclough* et al., 2005; Ghazanfar et al., 2005; Perrodin et al., 2014). However, the mechanisms subserving the observed selective cross modal inhibition of auditory processing based on the visual context are implemented not just during audio-visual blocks but all throughout any given run. As a result, they are expected to originate from a higher order cortical region exerting a top-down control on both the LS and the STS. The prefrontal cortex is a choice region in this respect, as it connects to LS (Rauschecker & Tian, 2000; Romanski et al., 1999; Saleem et al., 2014) and STS (Seltzer & Pandya, 1989, 1994) and has been shown to play a crucial role in working memory and the top-down modulation of perception (Fuster, 2002; Goldman-Rakic, 1996). LS and STS are also connected to the cingulate cortex and orbitofrontal cortex (Kondo et al., 2003; Saleem et al., 2008). These cortical regions that are involved in the processing of social interactions from visual cues and are thus in a position to provide feedbacks to the LS and STS on the social dimension of the stimuli (Cléry et al., 2021; Roberts, 2006; Rudebeck et al., 2006; Rushworth et al., 2007; Sliwa & Freiwald, 2017). Lastly, LS and STS are also connected to the limbic system (Amaral & Price, 1984; Kondo et al., 2003; Saleem et al., 2008). Accordingly, conspecific vocalisations activate a network recruiting, in addition to the voice patches, visual areas such as V4, MT, STS areas TE and TEO, as well as areas from the limbic and paralimbic system, including the hippocampus, the amygdala and the ventromedial prefrontal cortex (vmPFC) (Gil-da-Costa et al., 2004). All of these regions are expected to contribute (most probably in coordination) to setting the context based on which auditory information is considered either as congruent or incongruent. This will have to be addressed experimentally.”

References: Although the authors try to do justice to each sub-field they discuss, they remain eclectic in terms of who gets mentioned. The vast literature on face patches may have too many references, while the literature on dual-stream (dorsal/ventral) pathways misses some key references (e.g. Rauschecker & Tian, 2000, and Rauschecker & Scott, 2009).

This comment is well taken and addressed as follows: “The auditory processing circuit is proposed to be organized in two main networks, a ventral and a dorsal network (see for review Kuśmierk & Rauschecker, 2014; Rauschecker & Scott, 2009; Rauschecker & Tian, 2000), such that the auditory ventral stream, also called the pattern or “what” stream, is activated by conspecific vocalizations whereas the dorsal stream, also called spatial or “where” stream, is involved in the spatial location of sounds (Ortiz-Rios et al., 2015; Russ et al., 2008). Similarly, to face patches, voice processing by the “what” auditory stream, also involves a system of voice patches (for review, see Belin, 2017).”.

The latter paper (together with Rauschecker, 2011, and the most recent Archakov et al., PNAS 2020) will also provide background on the important role of the dorsal stream in sensorimotor aspects of vocal communication and on internal models, something that is completely overlooked here.

We would like to thank reviewer 3 for pointing out this highly relevant study which we discuss as follows: “As discussed above, a specific feature of our task design is that it creates, within each run, an implicit association between a set of social visual stimuli and their auditory match, possibly based on past learned sensori-motor associations and the development of internal models of what vocalizations are produced in a given visual context. This is reminiscent of the recent description of auditory fMRI activations to learned sound sequences in the motor cortex of the macaque brain (Archakov et al., 2020). These auditory responses were only present in monkeys who had received an audio-motor training and were only present in response to the learned sound and were absent for other sounds. The authors propose that an internal model of auditory perception associating a given auditory set of stimuli with a given motor repertoire (and thus motor structure) was created by the training. In parallel, we argue that the results presented here arise from the fact that macaques have, throughout their lifespan, associated specific macaque calls with specific social visual experiences, and that our specific task design allows us to reveal this internal model.”

Reference to the response to the reviewers

Amaral, D., & Price, J. L. (1984). Amygdalo-cortical projections in the monkey (*Macaca fascicularis*).

The Journal of comparative neurology. <https://doi.org/10.1002/CNE.902300402>

Archakov, D., DeWitt, I., Kuśmierk, P., Ortiz-Rios, M., Cameron, D., Cui, D., Morin, E. L., VanMeter, J.

W., Sams, M., Jääskeläinen, I. P., & Rauschecker, J. P. (2020). Auditory representation of learned sound sequences in motor regions of the macaque brain. *Proceedings of the National Academy of Sciences*, *117*(26), 15242- 15252. <https://doi.org/10.1073/pnas.1915610117>

Avillac, M., Ben Hamed, S. , & Duhamel, J.-R. (2007). Multisensory Integration in the Ventral

Intraparietal Area of the Macaque Monkey. *Journal of Neuroscience*, *27*(8), 1922- 1932. <https://doi.org/10.1523/JNEUROSCI.2646-06.2007>

Barraclough*, N. E., Xiao*, D., Baker, C. I., Oram, M. W., & Perrett, D. I. (2005). Integration of Visual

and Auditory Information by Superior Temporal Sulcus Neurons Responsive to the Sight of Actions. *Journal of Cognitive Neuroscience*, *17*(3), 377- 391.

<https://doi.org/10.1162/0898929053279586>

Beauchamp, M. S. (2005). See me, hear me, touch me : Multisensory integration in lateral occipital-temporal cortex. *Current Opinion in Neurobiology*, *15*(2), 145- 153.

<https://doi.org/10.1016/j.conb.2005.03.011>

- Bell, A. H., Hadj-Bouziane, F., Frihauf, J. B., Tootell, R. B. H., & Ungerleider, L. G. (2009). Object Representations in the Temporal Cortex of Monkeys and Humans as Revealed by Functional Magnetic Resonance Imaging. *Journal of Neurophysiology*, *101*(2), 688- 700.
<https://doi.org/10.1152/jn.90657.2008>
- Bodin, C., Trapeau, R., Nazarian, B., Sein, J., Degiovanni, X., Baurberg, J., Rapha, E., Renaud, L., Giordano, B. L., & Belin, P. (2021). Functionally homologous representation of vocalizations in the auditory cortex of humans and macaques. *Current Biology*.
<https://doi.org/10.1016/j.cub.2021.08.043>
- Chang, C., Cunningham, J. P., & Glover, G. H. (2009). Influence of heart rate on the BOLD signal : The cardiac response function. *NeuroImage*, *44*(3), 857- 869.
<https://doi.org/10.1016/j.neuroimage.2008.09.029>
- Cléry, J. C., Hori, Y., Schaeffer, D. J., Menon, R. S., & Everling, S. (2021). Neural network of social interaction observation in marmosets. *eLife*, *10*, e65012.
<https://doi.org/10.7554/eLife.65012>
- Cléry, J. C., Schaeffer, D. J., Hori, Y., Gilbert, K. M., Hayrynen, L. K., Gati, J. S., Menon, R. S., & Everling, S. (2020). Looming and receding visual networks in awake marmosets investigated with fMRI. *NeuroImage*, *215*, 116815. <https://doi.org/10.1016/j.neuroimage.2020.116815>
- Cléry, J., Guipponi, O., Odoard, S., Pinède, S., Wardak, C., & Hamed, S. B. (2017). The Prediction of Impact of a Looming Stimulus onto the Body Is Subserved by Multisensory Integration Mechanisms. *Journal of Neuroscience*, *37*(44), 10656- 10670.
<https://doi.org/10.1523/JNEUROSCI.0610-17.2017>
- Clery, J., Guipponi, O., Odoard, S., Wardak, C., & Ben Hamed, S. (2015). Impact Prediction by Looming Visual Stimuli Enhances Tactile Detection. *Journal of Neuroscience*, *35*(10), 4179- 4189. <https://doi.org/10.1523/JNEUROSCI.3031-14.2015>

- Cléry, J., Guipponi, O., Odouard, S., Wardak, C., & Ben Hamed, S. (2018). Cortical networks for encoding near and far space in the non-human primate. *NeuroImage*, *176*, 164- 178.
<https://doi.org/10.1016/j.neuroimage.2018.04.036>
- Cléry, J., Guipponi, O., Odouard, S., Wardak, C., & Hamed, S. B. (2015). Impact Prediction by Looming Visual Stimuli Enhances Tactile Detection. *Journal of Neuroscience*, *35*(10), 4179-4189.
<https://doi.org/10.1523/JNEUROSCI.3031-14.2015>
- Fuster, J. M. (2002). Physiology of executive functions : The perception-action cycle. In *Principles of frontal lobe function* (p. 96-108). Oxford University Press.
<https://doi.org/10.1093/acprof:oso/9780195134971.003.0006>
- Gentile, G., Petkova, V. I., & Ehrsson, H. H. (2010). Integration of Visual and Tactile Signals From the Hand in the Human Brain : An fMRI Study. *Journal of Neurophysiology*, *105*(2), 910-922.
<https://doi.org/10.1152/jn.00840.2010>
- Ghazanfar, A. A., Chandrasekaran, C., & Logothetis, N. K. (2008). Interactions between the Superior Temporal Sulcus and Auditory Cortex Mediate Dynamic Face/Voice Integration in Rhesus Monkeys. *Journal of Neuroscience*, *28*(17), 4457-4469.
<https://doi.org/10.1523/JNEUROSCI.0541-08.2008>
- Ghazanfar, A. A., Maier, J. X., Hoffman, K. L., & Logothetis, N. K. (2005). Multisensory Integration of Dynamic Faces and Voices in Rhesus Monkey Auditory Cortex. *Journal of Neuroscience*, *25*(20), 5004-5012. <https://doi.org/10.1523/JNEUROSCI.0799-05.2005>
- Gil-da-Costa, R., Braun, A., Lopes, M., Hauser, M. D., Carson, R. E., Herscovitch, P., & Martin, A. (2004). Toward an evolutionary perspective on conceptual representation : Species-specific calls activate visual and affective processing systems in the macaque. *Proceedings of the National Academy of Sciences*, *101*(50), 17516-17521.
<https://doi.org/10.1073/pnas.0408077101>

- Goldman-Rakic, P. S. (1996). Regional and cellular fractionation of working memory. *Proceedings of the National Academy of Sciences*, 93(24), 13473-13480.
<https://doi.org/10.1073/pnas.93.24.13473>
- Grant, K. W., & Seitz, P. F. (2000). The use of visible speech cues for improving auditory detection of spoken sentences. *The Journal of the Acoustical Society of America*, 108(3 Pt 1), 1197-1208.
<https://doi.org/10.1121/1.1288668>
- Guipponi, O., Cléry, J., Odouard, S., Wardak, C., & Ben Hamed, S. (2015). Whole brain mapping of visual and tactile convergence in the macaque monkey. *NeuroImage*, 117, 93-102.
<https://doi.org/10.1016/j.neuroimage.2015.05.022>
- Guipponi, O., Odouard, S., Pinède, S., Wardak, C., & Ben Hamed, S. (2015). fMRI Cortical Correlates of Spontaneous Eye Blinks in the Nonhuman Primate. *Cerebral Cortex*, 25(9), 2333-2345.
<https://doi.org/10.1093/cercor/bhu038>
- Joly, O., Pallier, C., Ramus, F., Pressnitzer, D., Vanduffel, W., & Orban, G. A. (2012). Processing of vocalizations in humans and monkeys : A comparative fMRI study. *NeuroImage*, 62(3), 1376-1389. <https://doi.org/10.1016/j.neuroimage.2012.05.070>
- Joly, O., Ramus, F., Pressnitzer, D., Vanduffel, W., & Orban, G. A. (2012). Interhemispheric Differences in Auditory Processing Revealed by fMRI in Awake Rhesus Monkeys. *Cerebral Cortex*, 22(4), 838-853. <https://doi.org/10.1093/cercor/bhr150>
- Khandhadia, A. P., Murphy, A. P., Romanski, L. M., Bizley, J. K., & Leopold, D. A. (2021). Audiovisual integration in macaque face patch neurons. *Current Biology*.
<https://doi.org/10.1016/j.cub.2021.01.102>
- Kondo, H., Saleem, K. S., & Price, J. L. (2003). Differential connections of the temporal pole with the orbital and medial prefrontal networks in macaque monkeys. *The Journal of Comparative Neurology*, 465(4), 499-523. <https://doi.org/10.1002/cne.10842>

- Kuśmierk, P., & Rauschecker, J. P. (2014). Selectivity for space and time in early areas of the auditory dorsal stream in the rhesus monkey. *Journal of Neurophysiology*, *111*(8), 1671-1685.
<https://doi.org/10.1152/jn.00436.2013>
- Laurienti, P. J., Burdette, J. H., Wallace, M. T., Yen, Y.-F., Field, A. S., & Stein, B. E. (2002). Deactivation of Sensory-Specific Cortex by Cross-Modal Stimuli. *Journal of Cognitive Neuroscience*, *14*(3), 420-429. <https://doi.org/10.1162/089892902317361930>
- Lee, H., & Noppeney, U. (2014). Temporal prediction errors in visual and auditory cortices. *Current Biology*, *24*(8), R309-R310. <https://doi.org/10.1016/j.cub.2014.02.007>
- Lehmann, S., & Murray, M. M. (2005). The role of multisensory memories in unisensory object discrimination. *Cognitive Brain Research*, *24*(2), 326-334.
<https://doi.org/10.1016/j.cogbrainres.2005.02.005>
- Leite, F. P., Tsao, D., Vanduffel, W., Fize, D., Sasaki, Y., Wald, L. L., Dale, A. M., Kwong, K. K., Orban, G. A., Rosen, B. R., Tootell, R. B. H., & Mandeville, J. B. (2002). Repeated fMRI using iron oxide contrast agent in awake, behaving macaques at 3 Tesla. *NeuroImage*, *16*(2), 283-294.
<https://doi.org/10.1006/nimg.2002.1110>
- Lewis, J. W., Beauchamp, M. S., & DeYoe, E. A. (2000). A Comparison of Visual and Auditory Motion Processing in Human Cerebral Cortex. *Cerebral Cortex*, *10*(9), 873-888.
<https://doi.org/10.1093/cercor/10.9.873>
- Marciniak, K., Atabaki, A., Dicke, P. W., & Thier, P. (2014). Disparate substrates for head gaze following and face perception in the monkey superior temporal sulcus. *eLife*, *3*, e03222.
<https://doi.org/10.7554/eLife.03222>
- Moeller, S., Freiwald, W. A., & Tsao, D. Y. (2008). Patches with Links : A Unified System for Processing Faces in the Macaque Temporal Lobe. *Science*, *320*(5881), 1355-1359.
<https://doi.org/10.1126/science.1157436>
- Murray, M. M., Molholm, S., Michel, C. M., Heslenfeld, D. J., Ritter, W., Javitt, D. C., Schroeder, C. E., & Foxe, J. J. (2005). Grabbing Your Ear : Rapid Auditory–Somatosensory Multisensory

- Interactions in Low-level Sensory Cortices Are Not Constrained by Stimulus Alignment. *Cerebral Cortex*, 15(7), 963-974. <https://doi.org/10.1093/cercor/bhh197>
- Navarra, J., & Soto-Faraco, S. (2007). Hearing lips in a second language : Visual articulatory information enables the perception of second language sounds. *Psychological Research*, 71(1), 4-12. <https://doi.org/10.1007/s00426-005-0031-5>
- Ortiz-Rios, M., Kuśmierk, P., DeWitt, I., Archakov, D., Azevedo, F. A. C., Sams, M., Jääskeläinen, I. P., Keliris, G. A., & Rauschecker, J. P. (2015). Functional MRI of the vocalization-processing network in the macaque brain. *Frontiers in Neuroscience*, 9. <https://doi.org/10.3389/fnins.2015.00113>
- Pelekanos, V., Mok, R. M., Joly, O., Ainsworth, M., Kyriazis, D., Kelly, M. G., Bell, A. H., & Kriegeskorte, N. (2020). Rapid event-related, BOLD fMRI, non-human primates (NHP) : Choose two out of three. *Scientific Reports*, 10(1), 7485. <https://doi.org/10.1038/s41598-020-64376-8>
- Perrodin, C., Kayser, C., Logothetis, N. K., & Petkov, C. I. (2014). Auditory and visual modulation of temporal lobe neurons in voice-sensitive and association cortices. *The Journal of Neuroscience: The Official Journal of the Society for Neuroscience*, 34(7), 2524-2537. <https://doi.org/10.1523/JNEUROSCI.2805-13.2014>
- Perrodin, C., Kayser, C., Logothetis, N. K., & Petkov, C. I. (2015). Natural asynchronies in audiovisual communication signals regulate neuronal multisensory interactions in voice-sensitive cortex. *Proceedings of the National Academy of Sciences*, 112(1), 273-278. <https://doi.org/10.1073/pnas.1412817112>
- Pollick, F., Love, S., & Latinus, M. (2011). Cerebral Correlates and Statistical Criteria of Cross-Modal Face and Voice Integration. *Seeing and Perceiving*, 24(4), 351-367. <https://doi.org/10.1163/187847511X584452>
- Poremba, A., Saunders, R. C., Crane, A. M., Cook, M., Sokoloff, L., & Mishkin, M. (2003). Functional Mapping of the Primate Auditory System. *Science*, 299(5606), 568-572. <https://doi.org/10.1126/science.1078900>

- Raab, D. H. (1962). Division of Psychology : Statistical Facilitation of Simple Reaction Times*. *Transactions of the New York Academy of Sciences*, 24(5 Series II), 574-590.
<https://doi.org/10.1111/j.2164-0947.1962.tb01433.x>
- Rauschecker, J. P., & Scott, S. K. (2009). Maps and streams in the auditory cortex : Nonhuman primates illuminate human speech processing. *Nature Neuroscience*, 12(6), 718-724.
<https://doi.org/10.1038/nn.2331>
- Rauschecker, J. P., & Tian, B. (2000). Mechanisms and streams for processing of “what” and “where” in auditory cortex. *Proceedings of the National Academy of Sciences*, 97(22), 11800-11806.
<https://doi.org/10.1073/pnas.97.22.11800>
- Roberts, A. C. (2006). Primate orbitofrontal cortex and adaptive behaviour. *Trends in Cognitive Sciences*, 10(2), 83-90. <https://doi.org/10.1016/j.tics.2005.12.002>
- Romanski, L. M., Tian, B., Fritz, J., Mishkin, M., Goldman-Rakic, P. S., & Rauschecker, J. P. (1999). Dual streams of auditory afferents target multiple domains in the primate prefrontal cortex. *Nature Neuroscience*, 2(12), 1131-1136. <https://doi.org/10.1038/16056>
- Ross, P., Atkins, B., Allison, L., Simpson, H., Duffell, C., Williams, M., & Ermolina, O. (2021). Children cannot ignore what they hear : Incongruent emotional information leads to an auditory dominance in children. *Journal of Experimental Child Psychology*, 204, 105068.
<https://doi.org/10.1016/j.jecp.2020.105068>
- Roumazeilles, L., Schurz, M., Lojkiewicz, M., Verhagen, L., Schüffelgen, U., Marche, K., Mahmoodi, A., Emberton, A., Simpson, K., Joly, O., Khamassi, M., Rushworth, M. F., Mars, R. B., & Sallet, J. (2021). *Social prediction modulates activity of macaque superior temporal cortex* (p. 2021.01.22.427803). <https://doi.org/10.1101/2021.01.22.427803>
- Rudebeck, P. H., Buckley, M. J., Walton, M. E., & Rushworth, M. F. S. (2006). A Role for the Macaque Anterior Cingulate Gyrus in Social Valuation. *Science*, 313(5791), 1310-1312.
<https://doi.org/10.1126/science.1128197>

- Rushworth, M. F. S., Behrens, T. E. J., Rudebeck, P. H., & Walton, M. E. (2007). Contrasting roles for cingulate and orbitofrontal cortex in decisions and social behaviour. *Trends in Cognitive Sciences*, 11(4), 168-176. <https://doi.org/10.1016/j.tics.2007.01.004>
- Russ, B. E., Ackelson, A. L., Baker, A. E., & Cohen, Y. E. (2008). Coding of Auditory-Stimulus Identity in the Auditory Non-Spatial Processing Stream. *Journal of Neurophysiology*, 99(1), 87-95. <https://doi.org/10.1152/jn.01069.2007>
- Saleem, K. S., Kondo, H., & Price, J. L. (2008). Complementary circuits connecting the orbital and medial prefrontal networks with the temporal, insular, and opercular cortex in the macaque monkey. *The Journal of Comparative Neurology*, 506(4), 659-693. <https://doi.org/10.1002/cne.21577>
- Saleem, K. S., Miller, B., & Price, J. L. (2014). Subdivisions and connectional networks of the lateral prefrontal cortex in the macaque monkey. *The Journal of Comparative Neurology*, 522(7), 1641-1690. <https://doi.org/10.1002/cne.23498>
- Seltzer, B., & Pandya, D. N. (1989). Frontal lobe connections of the superior temporal sulcus in the rhesus monkey. *Journal of Comparative Neurology*, 281(1), 97-113. <https://doi.org/10.1002/cne.902810108>
- Seltzer, B., & Pandya, D. N. (1994). Parietal, temporal, and occipital projections to cortex of the superior temporal sulcus in the rhesus monkey : A retrograde tracer study. *Journal of Comparative Neurology*, 343(3), 445-463. <https://doi.org/10.1002/cne.903430308>
- Shahin, A. J., & Miller, L. M. (2009). Multisensory integration enhances phonemic restoration. *The Journal of the Acoustical Society of America*, 125(3), 1744-1750. <https://doi.org/10.1121/1.3075576>
- Sliwa, J., & Freiwald, W. A. (2017). A dedicated network for social interaction processing in the primate brain. *Science*, 356(6339), 745-749. <https://doi.org/10.1126/science.aam6383>

- Stein, B. E., Stanford, T. R., Ramachandran, R., Perrault, T. J., & Rowland, B. A. (2009). Challenges in quantifying multisensory integration : Alternative criteria, models, and inverse effectiveness. *Experimental Brain Research*, *198*(2), 113. <https://doi.org/10.1007/s00221-009-1880-8>
- Stein, B. E., Stanford, T. R., & Rowland, B. A. (2014). Development of multisensory integration from the perspective of the individual neuron. *Nature Reviews Neuroscience*, *15*(8), 520-535. <https://doi.org/10.1038/nrn3742>
- Taubert, J., Van Belle, G., Vanduffel, W., Rossion, B., & Vogels, R. (2015). The effect of face inversion for neurons inside and outside fMRI-defined face-selective cortical regions. *Journal of Neurophysiology*, *113*(5), 1644-1655. <https://doi.org/10.1152/jn.00700.2014>
- Tsao, D. Y., Freiwald, W. A., Knutsen, T. A., Mandeville, J. B., & Tootell, R. B. H. (2003). Faces and objects in macaque cerebral cortex. *Nature Neuroscience*, *6*(9), 989. <https://doi.org/10.1038/nn1111>
- Tsao, D. Y., Moeller, S., & Freiwald, W. A. (2008). Comparing face patch systems in macaques and humans. *Proceedings of the National Academy of Sciences*, *105*(49), 19514-19519. <https://doi.org/10.1073/pnas.0809662105>
- Tyll, S., Bonath, B., Schoenfeld, M. A., Heinze, H.-J., Ohl, F. W., & Noesselt, T. (2013). Neural basis of multisensory looming signals. *NeuroImage*, *65*, 13-22. <https://doi.org/10.1016/j.neuroimage.2012.09.056>
- Vanduffel, W., Fize, D., Mandeville, J. B., Nelissen, K., Hecke, P. V., Rosen, B. R., Tootell, R. B. H., & Orban, G. A. (2001). Visual Motion Processing Investigated Using Contrast Agent-Enhanced fMRI in Awake Behaving Monkeys. *Neuron*, *32*(4), 565-577. [https://doi.org/10.1016/S0896-6273\(01\)00502-5](https://doi.org/10.1016/S0896-6273(01)00502-5)
- Van Wassenhove, V., Grant, K. W., & Poeppel, D. (2005). Visual speech speeds up the neural processing of auditory speech. *Proceedings of the National Academy of Sciences*, *102*(4), 1181-1186. <https://doi.org/10.1073/pnas.0408949102>

Welch, R. B., DuttonHurt, L. D., & Warren, D. H. (1986). Contributions of audition and vision to temporal rate perception. *Perception & Psychophysics*, 39(4), 294-300.

<https://doi.org/10.3758/BF03204939>

Werner, S., & Noppeney, U. (2010). Superadditive Responses in Superior Temporal Sulcus Predict Audiovisual Benefits in Object Categorization. *Cerebral Cortex*, 20(8), 1829-1842.

<https://doi.org/10.1093/cercor/bhp248>

REVIEWER COMMENTS

Reviewer #1 (Remarks to the Author):

In general, the authors performed an excellent job in addressing the comments of all reviewers. Although the quest for independent ROI definitions by the second reviewer is commendable, I am convinced that it will not change the large picture at all. In my mind, this is less critical. However, triggered by the second reviewer's comment (actually I thought about it as well right after I submitted my first review), the authors should discuss at length the implications of these thought-provoking results in relation to predictive coding. At first sight, these data are plain evidence against predictive coding using multi-sensory stimuli. Given the current importance of predictive coding, the authors should address this point.

Reviewer #2 (Remarks to the Author):

I thank the authors for their replies to my previous points of concerns. That said, I should also acknowledge that I have not found all of them convincing. I also do not feel that the manuscript quality is substantially improved, as major weaknesses remain. I will state my main points of concern below. Before doing so, I want to point out that the manuscript will need a thorough revision to improve clarity. There are lots of minor mistakes throughout, missing information, e.g. on figures, that make it an unnecessarily difficult read. For example, I am still not a hundred percent sure whether each run presents one consistent context – the Methods section seems to suggest so – but this should really be clear from the very beginning.

My first main concern is that statements about effect strengths are being made (e.g. line 111 following), which would require statements of beta values or percent signal change, all the while their values are not given, but rather (typically incomplete) statistical measures are reported in support, and figure after figure shows t-score maps and not effect strengths. The authors should check and provide the correct information to back up their claims. Similarly, there are claims of lack of significant differences between conditions made without formal tests (e.g. lines 128/129). These basic errors should be avoided throughout the manuscript. I would suggest, for good measure, to show a sample time course for a select ROI, probably deconvolved for the Molday ION HRF, early on, which should give the readers a feel for actual effect strengths. And Figure 6, or a variant of it, e.g. again for one sample ROI, should be presented earlier in the manuscript. Effects should be quantified. E.g. a response modulation index of the AC condition compared the Visual condition relative to baseline (activity during fixation condition) and for the AC and the AI condition.

There are several facts about the AC versus AI response differences that are astounding. First, there is time. I am assuming, as the Methods section appears to suggest, that context stayed constant throughout a run, but changed across runs. So, one might assume that the context effect grows stronger as more and more blocks with visual information are presented. Is this the case? Then there is the time from the last visual presentation (reminder of the context) – it can easily take 32 seconds from the end of a visual block to the beginning of an AI block. So, one would assume that effects might get weaker as time passes from the last presentation of visual context information. Is that the case? And is this also the case for the heart rate data.

Second, it is surprising just how consistent these effects are across the STS and across the lateral sulcus (Fig. 6). Given the huge functional variation across either of these sulci, it is very surprising that independent of this, there are AC responses of the same magnitude as visual ones, and AI responses close to zero. Activity levels during the fixation block should

be added to Fig. 6 as a baseline condition. It is difficult for me to understand why the authors would not map face areas, which is very straightforward to do, and, ideally, voice areas, which is also not difficult to do, and then run ROI analyses for these. This should be a minimum to get the manuscript beyond the description of basically one main experiment with some variation. To the very least it would allow the authors to make a strong claim about the consistency of effects across space independent of other functional specializations.

Yet, while this is the case, the response of the AC condition varies a lot with the context being a face or social (third point). In both cases the response to the AC condition happens to be very similar to that of the context setting visual stimulus, but as a result it varies greatly across these context conditions. It is mind-boggling how the social context should, over the course of half a minute so strongly depend on whether it was set by faces or by more general social stimuli.

The spatial patterns presented in Figs. 2-4 are very broad, yet locally patchy. It is difficult to assess how solid each of these activations is and thus how much trust to put into these broad patterns. The authors should register each of these activations with a common macaque brain atlas, and identify which brain areas are modulated and report effects in a table.

Relatedly, Supplemental Figure 1 shows huge variation of SNR strength even or especially within the parts of the brain for which coil positioning has been optimized. How do they explain this? How does the pattern of results they find correlate with this map? (The authors state that because of the low SNR in occipital cortex, they did not see any activations there. How can we know that this is true? How low was the SNR? Was it really too low to be able to see anything at all?)

The description of heart rate variations is confusing. The dominant thing one sees in the figure (which should be improved with complete descriptions of labels) is an interaction effect, while the manuscripts starts talking about a main effect that is much smaller. It certainly goes to show there is a behavioral signature. It would be important to know its timecourse (see comment above) in comparison to that of the pattern of brain activation. While it is reassuring that the regression of the heart rate did not alter the results, it would still be prudent to consider potential effects on hemodynamic coupling.

This brings me to my last point, the engagement with the past literature. There is single unit data from various brain regions on face voice integration, including consistency effects. The authors should engage with this past literature more closely and more quantitatively. What were the biggest effects reported in this literature and how do they compare to the similarly quantified (see comment above) effects reported here? One recent paper on the matter is Khandhadia et al., which the authors cite, but rather mention than really evaluate.

Khandhadia et al. study two specific face areas and find evidence for face voice interactions in one, but not the other. They also have quantifications of effect strength. There should be a deeper consideration of just how surprising the current results are in light of the past literature and maybe a critical reflection on whether the current results reflect an underlying neural truth or other factors as well. Similarly, I do think, despite the authors' response, that a comparison to the predictive coding account would be important for the discussion, again highlighting that the current results are the opposite of that account and discussing why this might be.

Lastly, the abstract in the current manuscript is highly descriptive of detail and fails to highlight conceptual relevance.

Reviewer #3 (Remarks to the Author):

The authors have responded comprehensively and adequately to my critique and comments. I feel the manuscript is further improved by these changes and by the authors' response to the other reviewers. From my point of view, I don't see any reason to deny this important and novel work immediate publication. The term 'heroic' that I've used in my original review applies even more to the present version.

We would like to thank the reviewers for their appreciation of our work and their constructive feedback on the manuscript. By taking into account this feedback, our revised manuscript has been considerably improved. Moreover, by clarifying several of the methodological details and providing additional analyses, we shore up the significance of our results. Importantly, none of the original findings are materially changed but rather, are further supported. Below is a point by point response to the questions raised by the reviewers.

Reviewer #1 (Remarks to the Author):

1. In general, the authors performed an excellent job in addressing the comments of all reviewers. Although the quest for independent ROI definitions by the second reviewer is commendable, I am convinced that it will not change the large picture at all. In my mind, this is less critical.

We now use a visual task consisting of runs composed of pseudorandom alternations of blocks of faces with different facial expressions (aggressive, neutral, lip-smacking, scared) and fixation. As expected by Reviewer 1, this allows us to generalize our observations at locations independent from the peak activations identified in the main audio-visual task. Please see our response to reviewer #2, point #10, for a more extensive description of these observations.

2. However, triggered by the second reviewer's comment (actually I thought about it as well right after I submitted my first review), the authors should discuss at length the implications of these thought-provoking results in relation to predictive coding. At first sight, these data are plain evidence against predictive coding using multi-sensory stimuli. Given the current importance of predictive coding, the authors should address this point.

We have now included an independent discussion paragraph on predictive coding that reads as follows:

"It is worth noting that our results go against the predictive coding theory. This theory posits that the brain is constantly generating and updating an internal model of the environment. This model then generates predictions of sensory input and compares these to actual sensory input (Friston, 2010; Rao & Ballard, 1999). Prediction errors are then used to update and revise the internal model (Millidge et al., 2022). In the context of predictive coding, when viewing an affiliative face, monkeys are expected to predict affiliative vocalisations. As a result, aggressive vocalisations in the context of affiliative faces are expected to generate prediction errors and hence higher activations than those observed for the affiliative vocalisations. This is not what our data show: when, viewing affiliative faces, there are enhanced responses to affiliative vocalisation and suppressed responses to aggressive vocalisations. This effect actually builds up as visual contextual information is reinforced through the run and is present in both the STS and the LS, i.e. at the early stages of auditory processing. Thus, these observations are inconsistent with the predictive coding experimental predictions. They suggest that the monkeys implement an active matching or association between the visual and the auditory social information, similar to a match to sample task, based on their life long social experiences. In match to sample fMRI and EEG studies in humans (Druzgal & D'Esposito, 2001) and electrophysiology studies in non-human primates (Miller & Desimone, 1994; Suzuki et al., 1997), responses to the probe matching the sample is significantly higher than the response to a non-match probe, thus describing a match enhancement (Suzuki & Eichenbaum, 2000). This is very similar to what we describe here, if considering the visual context as the probe and the auditory stimuli as the match and non-match probes. Further work is required to confirm this hypothesis."

Reviewer #2 (Remarks to the Author):

I thank the authors for their replies to my previous points of concerns. That said, I should also acknowledge that I have not found all of them convincing. I also do not feel that the manuscript quality is substantially improved, as major weaknesses remain. I will state my main points of concern below.

1. Before doing so, I want to point out that the manuscript will need a thorough revision to improve clarity. There are lots of minor mistakes throughout, missing information, e.g. on figures, that make it an unnecessarily difficult read. For example, I am still not a hundred percent sure whether each run presents one consistent context – the Methods section seems to suggest so – but this should really be clear from the very beginning.

We have now thoroughly reread our manuscript, tracking possible minor errors and missing information, including on the figures, and figure legends.

In addition, it has now also been made explicit, at the very beginning of the result section, that indeed, each run presents a unique and consistent context as follow: “We designed six variants of a unique task in which we systematically manipulated the general semantics or meaning of the context as specified by visual information and presented as independent runs in the sessions. Each context, and so each independent run, combined visual stimuli of identical social content with either semantically congruent or incongruent monkey vocalisations presented together with the visual stimuli or not. [...] Importantly, paired contexts shared the same auditory stimuli, but opposite social visual content (Figure 1B), thus opposite semantic content and meaning. All contexts were presented randomly in independent runs and at least once during each scanning session. We report group fMRI and group heart-rate analyses. All reported statistics are based on non-parametric tests.”

2. My first main concern is that statements about effect strengths are being made (e.g. line 111 following), which would require statements of beta values or percent signal change, all the while their values are not given, but rather (typically incomplete) statistical measures are reported in support, and figure after figure showed t-score maps and not effect strengths. The authors should check and provide the correct information to back up their claims. Similarly, there are claims of lack of significant differences between conditions made without formal tests (e.g. lines 128/129). These basic errors should be avoided throughout the manuscript.

Reviewer #2 is correct in that effect sizes are only presented later on in the manuscript in Figure 6 and in that statistical reports focused on our major observations. This has been corrected throughout the revised manuscript and new supplementary figures have been added to back up the t-score maps (see this point, below). In addition, in response to further comments from reviewer #2 below, we have added modulation index distributions (e.g. point 5) as well as tables documenting %SC and associated statistics across conditions (e.g. point 11).

In particular, line 11 now reads as: “Combining the F+ and F- face contexts (Figure 2, see Supplemental figure S1A for individual monkey maps and supplemental figure S2 and main figure 6 for effect sizes in representative ROIs), which includes...” Figure S2 is discussed in point 4 below.

Lines 128/129 now reads as: “These activations are significantly higher than those observed for the incongruent vocalisation, whether the congruent auditory stimuli are coos (Figure 3B, figure S2 and figure S6 for the effect strengths of the t-score maps) or aggressive calls (Figure 3C), although congruent coos led to significantly higher activations than congruent aggressive calls (figure S6) ...” Figure S6 is described next. Figure S2 is discussed in point 4 below. Figure S6 is discussed below.

Supplemental figure S6 now quantifies the effect size associated with the t-score maps presented in figures 2 and 3, and separately shows the %SC for the Coos and Aggressive calls in each of the positive and negative visual face contexts, for the STS and the LS ROIs defined in figure 6.

Supplemental figure S6 and its legend are duplicated below:

Supplemental figure S6. Percentage of signal change (%SC) across all superior temporal sulcus and the lateral sulcus ROIs of both hemispheres (median + se, see ROI definition in figure 6), comparing auditory congruent and incongruent conditions vs fixation for the face context F+ and F- separately, and coos and aggressive calls independently. Note that auditory congruent (AC) stimuli for the positive face context (F+) are coos and incongruent auditory stimuli (AI) are aggressive calls (Aggr). For the negative face context (F-), this is reversed as coos are incongruent with respect to the context and aggressive calls are congruent. The specific types of coos and aggressive calls used in both F+ and F- contexts are identical. Statistical differences relative to fixation or across conditions in the F+ and F- face contexts are indicated as follows: ***, $p < 0.001$; **, $p < 0.01$; *, $p < 0.05$; n.s., $p > 0.05$ (Wilcoxon non-parametric test: STS: F+: Coos: $Z = 12.5$, $p < 0.001$; Aggr: $Z = 3.35$, $p = 0.019$; Coos vs Aggr: $Z = 7.67$, $p < 0.001$ F-: Aggr: $Z = 10.01$, $p < 0.001$; Coos: $Z = 3.34$, $p = 0.019$; Coos vs Aggr: $Z = 5.19$, $p < 0.001$. Coos_F+ vs Coos_F-: $Z = 7.63$, $p < 0.001$; Aggr_F+ vs Aggr_F-: $Z = 5.29$, $p < 0.001$; Coos_F+ vs Aggr_F-: $Z = 3.13$, $p = 0.002$; Coos_F- vs Aggr_F+: $Z = 0.1$, $p = 0.92$. LS: F+: Coos: $Z = 8.69$, $p < 0.001$; Aggr: $Z = 3.69$, $p < 0.001$; Coos vs Aggr: $Z = 4.98$, $p < 0.001$; F-: Aggr: $Z = 7.77$, $p < 0.001$; Coos: $Z = 4.43$, $p < 0.001$; Coos vs Aggr: $Z = 2.86$, $p = 0.016$; Coos_F+ vs Coos_F-: $Z = 4.45$, $p < 0.001$; Aggr_F+ vs Aggr_F-: $Z = 2.64$, $p = 0.008$; Coos_F+ vs Aggr_F-: $Z = 2.65$, $p = 0.007$; Coos_F- vs Aggr_F+: $Z = 0.65$, $p = 0.51$). This figure quantifies the effect strengths of the t-score maps presented in Figures 2 and 3.

Likewise, Supplemental figure S10 now quantifies the effect size associated with the t-score maps presented in figure 4, and separately shows the %SC for the Coos and Negative calls in each of the positive and negative visual social context, for the STS and the LS ROIs defined in figure 6.

This figure is referred to in the main text as follows: "... with either semantically congruent or incongruent vocalisations (Figure 4 for all social contexts on group data, see Supplemental figure S1B for individual monkey data, [...], and figures S9 and S10 for effect strengths in representative ROIs of the t-score map, [...])." Figure S9 is discussed in point 4 below.

Supplemental figure S10 and its legend are duplicated below:

Supplemental figure S10. Percentage of signal change (%SC) across all superior temporal sulcus and the lateral sulcus ROIs of both hemispheres (median + se, see ROI definition in figure 6), comparing auditory congruent and incongruent conditions vs fixation for the social context S+ and S- separately, and coos and negative calls (aggressive calls and screams) independently. Note that auditory congruent (AC) stimuli for the positive social context (S+) are coos and incongruent auditory stimuli (AI) are aggressive calls and screams (Neg). For the negative social context (S-), this is reversed as coos are incongruent with respect to the context and aggressive calls and screams are congruent. The specific types of coos and aggressive calls and screams used in both S+ and S- contexts are identical. Statistical differences relative to fixation or across conditions in the S+ and S- face contexts are indicated as follows: ***, $p < 0.001$; **, $p < 0.01$; *, $p < 0.05$; n.s., $p > 0.05$ (Wilcoxon non-parametric test: STS: S+: Coo: $Z = 5.30$, $p < 0.001$; Neg: $Z = 0.33$, $p = 0.74$; Coos vs Neg: $Z = 3.08$, $p = 0.002$ S-: Neg: $Z = 12.76$, $p < 0.001$; Coo: $Z = 3.58$, $p < 0.001$; Coos vs Neg: $Z = 5.72$, $p < 0.001$. Coos_S+ vs Coos_S-: $Z = 0.99$, $p = 0.31$; Neg_S+ vs Neg_S-: $Z = 7.57$, $p < 0.001$; Coos_S+ vs Neg_S-: $Z = 4.23$, $p < 0.001$; Coos_S- vs Neg_S+: $Z = 0.24$, $p = 0.02$. LS: S+: Coo: $Z = 2.04$, $p = 0.03$; Neg: $Z = 0.26$, $p = 0.79$; Coos vs Neg: $Z = 0.65$, $p = 0.5$. S-: Neg: $Z = 6.89$, $p < 0.001$; Coo: $Z = 6.62$, $p < 0.001$; Coos vs Neg: $Z = 3.86$, $p < 0.001$; Coos_S+ vs Coos_S-: $Z = 2.82$, $p = 0.014$; Neg_S+ vs Neg_S-: $Z = 4.24$, $p < 0.001$; Coos_S+ vs Neg_S-: $Z = 3.71$, $p < 0.001$; Coos_S- vs Neg_S+: $Z = 3.71$, $p < 0.001$). This figure quantifies the effect strengths of the t-score maps presented in Figure 4.

3. I would suggest, for good measure, to show a sample time course for a select ROI, probably deconvolved for the Molday ION HRF, early on, which should give the readers a feel for actual effect strengths.

Following on reviewer's #2 suggestion, we now present in supplementary figure S18 a representative response time course, deconvolved for the Molday ION HRF, for the different stimulation conditions. The data are derived from 4 runs with the same block structure (one per F+ and F- face context and per monkey). Figure S18 and its legend are reproduced below.

Figure S18 is cited in the methods section, Data analysis, as follows: "A spatial smoothing was applied with a 3-mm FWHM Gaussian Kernel. A representative example of time courses is presented in supplemental figure S18."

Supplemental figure S18. Median time courses in percent signal change extracted from the STS ROIs in a representative Face run. Time courses were averaged across monkeys, hemispheres and ROIs for runs having the same bloc organization. Block conditions (16s) are indicated by different colors: VAI in purple, VAC in cyan, fixation in white, AI in red, AC in green and Vi in gray. Specifically, time courses were extracted thanks to MarsBaR (Brett et al., 2002), for each STS ROIs and for each of the selected Face context runs. We averaged the signal across these ROIs and across runs with the same block sequence. This includes 4 runs (one per F+ and F- face context and per monkey). We converted the signal in percent signal change ($(\text{signal} - \text{signal's median} / \text{signal's median}) \times 100$). We then inverted the signal to account for the fact that the MION produces a decrease in signal following neuronal activation.

4. And Figure 6, or a variant of it, e.g. again for one sample ROI, should be presented earlier in the manuscript.

We have now added supplemental figures S2 and S9 respectively describing the effect sizes of the statistical maps presented in figures 2 and 4. These are reproduced below. In addition, as per reviewer #2's point 3, distributions of modulation indexes are now also presented (see point 5, below).

These supplemental figures are referred to in the main text as follows: "Combining the F+ and F- face contexts (Figure 2, see Supplemental figure S1A for individual monkey maps and figure S2 and main figure 6 for effect strengths in representative ROIs), ..." and "Importantly, this context also leads to the same robust bilateral activations as the visual contrast: the extra-striate cortex, along the superior temporal sulcus (STS) ($p < 0.05$ FWE), as well as in the prefrontal and intraparietal cortex ($p < 0.0001$ uncorrected). Percent signal change at local peak activations in the lateral sulcus and superior temporal sulcus are presented in supplemental figure S2 and supplemental figure S5 (left) represents the distribution of AC-AI/AC+AI and AC-V/AC+V modulation indexes across ROIs, thus describing the effect strength." and "These activations are significantly higher than those observed for the incongruent vocalisations, whether the congruent auditory stimuli are coos (Figure 3B and figure S6 and figure S3 for the effect strengths of the t-score maps) ..." and "... with either semantically congruent or incongruent vocalisations (Figure 4 for all social contexts on group data, see Supplemental figure S1B for individual monkey data, [...], figure S9 and S10 for effect strengths in representative ROIs of the t-score map, [...])."

Supplemental figure S2: Percentage of signal change (%SC) for selected left and right hemisphere ROIs in the lateral sulcus and in the superior temporal sulci. ROIs are 1.5mm spheres located at local peak activations of these two sulci. ROI location in each of the left and right STS and LS is described in the bottom flat maps of figure 6. %SC (mean +/- se) are presented for each ROI (8 in right STS, 6 in left STS, 4 in left and 6 in right lateral sulcus) for the face context, for the contrast Visual vs Fixation (Vi), Auditory congruent vs Fixation (AC) and Auditory incongruent vs Fixation (AI).

Supplemental figure S9: Percentage of signal change (%SC) for selected left and right hemisphere ROIs in the lateral sulcus and in the superior temporal sulci. ROIs are 1.5mm spheres located at local peak activations of these two sulci. ROI location in each of the left and right STS and LS is described in the bottom flat maps of figure 6. %SC (mean +/- se) are presented for each ROI (8 in right STS, 6 in left STS, 4 in left and 6 in right lateral sulcus) for the social context, for the contrast Visual vs fixation (Vi), Auditory congruent vs fixation (AC) and auditory incongruent vs fixation (AI).

5. Effects should be quantified. E.g. a response modulation index of the AC condition compared to the Visual condition relative to baseline (activity during fixation condition) and for the AC and the AI condition.

Effects are now quantified in supplementary figures S6 (point 2 above), S10 (point 2 above), S3, S7 and S11 and new supplementary tables T1, T2, T3 (point 11 below).

In addition, we now introduce the following supplemental figure S5, representing the modulation index of the AC condition compared to the Vi condition relative to fixation baseline (AC-V/AC+V) and the modulation index of the AC condition compared to the AI condition relative to fixation baseline (AC-AI/AC+AI), for both the LS and STS, and for each of the face and social runs. These modulation indexes represent a quantification of the effects displayed in supplemental figures S6, S10, S2, S9, and main figure 6. Figure S5 and its legend are reproduced below.

Figure S5 is cited in the main text as follows: "Percent signal change at local peak activations in the lateral sulcus and superior temporal sulcus are presented in supplemental figure S2. Supplemental

figure S5A (left) represents the distribution of AC-AI/AC+AI (figure S5A) and AC-V/AC+V (figure S5B) modulation indexes across ROIs, thus describing the effect strength.” And: “To further quantify the effect strength of congruency on auditory processing, we computed an AC-AI/AC+AI modulation index (figure S5A) for both face and social contexts. In both lateral and superior temporal sulci and both types of contexts, this index reveals a significantly higher activation for auditory congruent vocalisation than auditory incongruent stimuli.”

Supplemental figure S5. Distribution of modulation index of percentage signal change (%SC) for the AC condition relative to fixation baseline compared to the AI condition relative to fixation baseline (A, (AC-AI/AC+AI)) and for the AC condition relative to fixation baseline compared to the Visual condition relative to fixation baseline (B, (AC-V/AC+V)), for the STS and LS, and each of the face and social runs, computed on individual ROIs across all runs. The thin line represents the zero index (no modulation of the hemodynamic response). The Thick vertical line represents the median of the modulation index distribution. Statistical differences relative to zero are indicated as follows: ***, $p < 0.001$; **, $p < 0.01$; n.s., $p > 0.05$ (Wilcoxon non-parametric test, A) AC/AI index: FACE: STS: $Z = 4.78$, $p < 0.001$ LS: $Z = 3.99$, $p < 0.001$; SOCIAL: STS: $Z = 4.79$, $p < 0.001$ LS: $Z = 2.89$, $p < 0.001$. B) AC/V index: FACE: STS: $Z = 2.38$, $p < 0.01$, STS: $Z = 3.19$, $p < 0.001$; SOCIAL: STS: $Z = 2.73$, $p < 0.001$ LS: 4.10, $p < 0.001$).

6. There are several facts about the AC versus AI response differences that are astounding. First, there is time. I am assuming, as the Methods section appears to suggest, that context stayed constant throughout a run, but changed across runs. So, one might assume that the context effect grows stronger as more and more blocks with visual information are presented. Is this the case?

This is a very relevant question. We now provide a supplemental figure S12a that shows that modulation index of AC relative to AI progressively grows stronger from the first occurrence in the run to the last. This effect is present in both the STS and the LS, though it is more marked in the STS. This strongly indicates, that as predicted by reviewer 2, context effect grows stronger and stronger within the run. Figure S12a and its legend are reproduced below together with our response to point 6.

Figure S12a is cited in the main text as follows: "It is worth noting that, in 66% of the instances, both AI and AC conditions are preceded by blocks involving visual stimulation (Vi, VAC and VAI). Because this was the case for both AI and AC conditions, the absence of auditory activations in the AI vs. Fx contrast and the presence of temporal and occipital activations in the AC vs. Fx contrast cannot be interpreted as a trace of the activations resulting from the previous blocks. Instead, this pattern of responses should be considered as a process that results from the structure of the task. Indeed, the AC-AI/AC+AI modulation index progressively grows stronger within any given run, as visual stimulation reinforces context-related information. This supports the idea that the observed enhancement of AC relative to AI is context-dependent (supplemental figure S12A). In addition, this modulation index is not significantly different whether the auditory stimuli were presented right after a block containing visual information or separated in time from it (see Supplemental figure S12B)."

7. Then there is the time from the last visual presentation (reminder of the context) – it can easily take 32 seconds from the end of a visual block to the beginning of an AI block. So, one would assume that effects might get weaker as time passes from the last presentation of visual context information. Is that the case? And is this also the case for the heart rate data.

This is again a very relevant question. We now provide a supplemental figure S12b showing that the modulation index of AC relative to AI is not significantly different whether the auditory stimuli were presented right after a block containing visual information or at a distance from it. This is the case in both the STS and the LS. This strongly indicates that the decreased responsiveness to the AI stimulus results from the context set by the run and not by the immediately preceding visual information. Figure S12b and its legend are reproduced below.

Figure S12b is cited in the main text as follows: "It is worth noting that, in 66% of the instances, both AI and AC conditions are preceded by blocks involving visual stimulation (Vi, VAC and VAI). Because this was the case for both AI and AC conditions, the absence of auditory activations in the AI vs. Fx contrast and the presence of temporal and occipital activations in the AC vs. Fx contrast cannot be interpreted as a trace of the activations resulting from the previous blocks. Instead, this pattern of responses should be considered as a process that results from the structure of the task. Indeed, the AC-AI/AC+AI modulation index progressively grows stronger within any given run, as visual stimulation reinforces context-related information. This supports the idea that the observed enhancement of AC relative to AI is context-dependent (supplemental figure S12A). In addition, this modulation index is not significantly different whether the auditory stimuli were presented right after a block containing visual information or separated in time from it (see Supplemental figure S12B)."

Evolution of AC-AI/AC+AI modulation index as a function of

Supplemental figure S12. Distribution of modulation index of percentage signal change (%SC) for the AC condition relative to fixation baseline compared to the AI condition relative to fixation baseline (AC-AI/AC+AI), as a function of repetition order in the run (A) or as a function of the distance from the last visual block (B), for each of the STS and LS, and each of the face and social runs, computed on individual ROIs across all runs. In (A), 1: first occurrence of AC or AI, 2: second occurrence, 3: third occurrence. In (B), 1: AC or AI just following a block with visual stimuli presentations, 2: AC or AI presented two blocks away from a block with visual stimuli presentations. Statistical differences relative to baseline or across conditions are indicated as follows: ***, $p < 0.001$; **, $p < 0.01$; *, $p < 0.05$; n.s., $p > 0.05$ (Wilcoxon non-parametric test: A) STS 1: $Z = 3.21$, $p < 0.001$; 2: $Z = 3.41$, $p < 0.001$; 3: $Z = 4.78$, $p < 0.001$; 1-2: $Z = 1.58$, $p = 0.11$; 1-3: $Z = 4.16$, $p < 0.01$; 2-3: $Z = 1.81$, $p = 0.06$. LS: 1: $Z = 1.57$, $p = 0.11$; 2: $Z = 2.38$, $p = 0.02$; 3: $Z = 4.38$, $p < 0.01$; 1-2: $Z = 1.77$, $p = 0.07$; 1-3: $Z = 2.3$, $p = 0.02$; 2-3: $Z = 0.86$, $p = 0.38$. B) STS: 1: $Z = 3.26$, $p < 0.001$; 2: $Z = 3.62$, $p < 0.001$; 1-2: $Z = 1.58$, $p = 0.19$; LS: 1: $Z = 3.18$, $p < 0.001$; 2: $Z = 3.28$, $p < 0.001$; 1-2: $Z = 0.05$, $p = 0.8$).

8. Second, it is surprising just how consistent these effects are across the STS and across the lateral sulcus (Fig. 6). Given the huge functional variation across either of these sulci, it is very surprising that independent of this, there are AC responses of the same magnitude as visual ones, and AI responses close to zero. Activity levels during the fixation block should be added to Fig. 6 as a baseline condition.

Figure 6 represents % signal change relative to baseline. As a result, it already answers the request of reviewer 2. This is now mentioned in the figure legend as "(B) %SC (median) are presented for each ROI (8 in right STS, 6 in left STS, 4 in left and 6 in right lateral sulcus) and each contrast of interest (V: visual vs fixation, AC: auditory congruent vs fixation, AI: auditory incongruent vs fixation, VAC: visuo-auditory congruent vs fixation, VAI: visuo-auditory incongruent vs fixation)." We have also added the figure labels.

9. It is difficult for me to understand why the authors would not map face areas, which is very straightforward to do, and, ideally, voice areas, which is also not difficult to do, and then run ROI analyses for these. This should be a minimum to get the manuscript beyond the description of basically one main experiment with some variation. To the very least it would allow the authors to make a strong claim about the consistency of effects across space independent of other functional specializations.

In response to Reviewer #2's comment, we have collected new data, in the same animals, from a visual task consisting of runs composed of pseudorandom alternations of blocks of faces with different facial expressions (aggressive, neutral, lip-smacking, scared) and fixation. This allows us to identify the face-related network in these monkeys and to generalize our observations at cortical locations independent from the peak activations identified in the main audio-visual task. This is now discussed in a supplementary figure S15 and associated note as follows:

Supplemental figure S15. (A) Whole-brain activation maps of a pure visual task, pooled across both monkeys, for lipsmack + aggressive blocks versus fixation contrast. Darker shades of red indicate level

of significance at $p < 0.001$ uncorrected, t-score 3.09. Lighter shades of yellow indicate level of significance at $p < 0.05$ FWE, t-score 4.6. (B) ROIs are 1.5mm red spheres located at local peak activations of the pure visual task, in the STS. Left and right hemisphere numbering associate mirror ROIs. (C) ROIs are 1.5mm blue spheres located at local peak activations of the audio-visual task described in the main text. Left and right hemisphere numbering associate mirror ROIs, reproducing main Figure 6. (D) STS location of the ROIs from the pure visual task (red) overlaid on those identified from the audio-visual task (blue). (E) Percentage signal change (%SC, median + se) in the audio-visual face context and each condition of interest vs fixation (V: visual, AC: auditory congruent, AI: auditory incongruent, VAC: visuo-auditory congruent, VAI: visuo-auditory incongruent) are presented for each ROI defined from the pure visual task. No ROI effect $p > 0.05$, condition effect **, $p < 0.01$, no interaction effect $p > 0.05$ (Friedman non-parametric test). This reproduces the results presented in main Figure 6, with ROIs defined in the visual task. (F) Percentage of signal change (%SC) across all ROIs (extracted from the pure visual task) of superior temporal sulcus in both hemispheres (median + se), comparing the different conditions of interest vs fixation in the audio-visual face context (V: visual, AC: auditory congruent, AI: auditory incongruent, VAC: visuo-auditory congruent, VAI: visuo-auditory incongruent). Black: all ROIs; red: only ROIs that do not overlap with the ROIs defined from the main audio-visual task. Statistical differences relative to fixation or between conditions in the face context and indicated as follows: ***, $p < 0.001$; **, $p < 0.01$; *, $p < 0.05$, n.s., $p > 0.05$ (Wilcoxon non-parametric test). (G) Same as E, but for the audio-visual social task. This reproduces the results presented in main Figure 6, with ROIs defined in an independent visual task. (H) Same as F, but for the audio-visual social task.

“Supplemental note associated with supplementary figure S15. We use a visual task consisting of runs composed of pseudorandom alternations of blocks of faces with different facial expressions (aggressive, neutral, lip-smacking, fear) and fixation. We use this task to define ROIs that are independent from the audio-visual tasks presented in the main paper and we reproduce our main observation that in these new ROIs, the %SC in the AI condition (relative to fixation), is significantly lower than in all the other conditions. This thus allows to generalize our observations at locations independent from the peak activations identified in the main audio-visual task. Specifically, we use the aggressive+lipsmack vs. fixation contrast, thus using the same visual stimuli as in the audio-visual face task, to identify face responsive activations (Figure S15a). We identified the activation local maxima observed in the STS with this contrast (Figure S15b). Figure S15d shows the location of these peaks (red) relative to those defined in the audio-visual task (blue, Figure S15c). While some of these ROIs closely overlap with those identified in the audio-visual task, others don't, suggesting a specialization for social audio-visual processing in these latter ROIs. In both the audio-visual face (Figure S15e, Figure S15f) and social task (Figure S15g, Figure S15h), AI %SC relative to fixation is, over all ROIs, significantly lower than %SC in all other conditions, including when only considering the ROIs from the visual task that are non-overlapping with the ROIs from the audio-visual task (Figure S15g, Figure S15h, red). Overall, this rules out any possible concern about repeatedly sampling the same data set and indicates that the results reported in the main manuscript are not idiosyncratic to the ROI definition.

Design of the pure Visual runs

The design of the visual runs was similar to that of the audio-visual run design, organized in blocks, except for the fact that all blocks were pure visual blocks and varied as a function of facial emotions. The six possible 16 s blocks were: fixation (Fx), lipsmack, fearful monkey faces, aggressive monkey faces, neutral monkey faces and scrambled monkey faces. As for the audio-visual runs, each block consisted in an alternation of 500 ms stimuli (except for lip smacks, 1s dynamic stimuli succession) of the same emotional category, as was the case in the main task.”

Supplemental Note N1 is now mentioned in the main text as follows: “No interhemispheric difference could be noted (LS: FACE $F_{(1,40)} = 0.136$; $p = 0.714$; SOCIAL: $F_{(1,78)} = 0.727$; $p = 0.396$ and STS: FACE $F_{(1,40)} = 0.014$; $p = 0.906$; SOCIAL: $F_{(1,78)} = 0.544$; $p = 0.463$). Note that these observations are preserved when ROIs are defined in an independent set of data identifying face-related activation local maxima from a purely visual task (see supplementary figure S15 and its associated note).”

10. Yet, while this is the case, the response of the AC condition varies a lot with the context being a face or social (third point). In both cases the response to the AC condition happens to be very similar to that of the context setting visual stimulus, but as a result it varies greatly across these context conditions. It is mind-boggling how the social context should, over the course of half a minute so strongly depend on whether it was set by faces or by more general social stimuli.

This is an interesting comment from reviewer #2. We are now discussing this point in a new discussion section as follows:

“Visual and auditory responses in the lateral sulcus and superior temporal sulcus

Expectedly, the LS activations in response to auditory stimuli are higher than its activations to visual stimuli (Figure 7). This most probably arises from the fact that while the primary function of the LS is auditory processing, it receives (visual) input from the adjacent STS (Calvert, 2001; Ghazanfar et al., 2005). In contrast, based on the ROIs defined in the audio-visual face task, STS appears to be equally responsive to auditory and visual stimuli (Figure 7, trend to significance), although AC-V/AC+V modulation indexes are significantly negative (figure S5). When ROIs are defined on the basis of a purely visual task, STS visual responses are significantly higher than STS auditory responses (Figure S15). Overall, this suggests the existence, within the STS, of specialized regions involved in the visuo-auditory association of social stimuli. While large areas of the STS become responsive to auditory stimuli during visuo-auditory association of social stimuli perhaps due to a direct projection from the LS to the STS (Seltzer & Pandya, 1994) -- only some regions are activated to almost a similar level by both sensory modalities. These regions could contribute to the amodal representation of social stimuli.

The specific point about auditory congruent activations being different between the face and social context is now discussed in the result section as follows: “Last, V and AC activations were significantly weaker in the social context relative to the face context (AC: STS: $Z = 7.17$, $p < 0.001$; LS: $Z = 4.9$, $p < 0.001$; V: STS: $Z = 6.54$, $p < 0.001$; LS: 4.32 $p < 0.001$). This is most probably due to the fact that both visual (faces vs. social scenes) and auditory stimuli (coos + aggressive calls vs. coos + aggressive calls + screams) were different between the two contexts. This could have resulted in low level sensory differences in stimulus processing due to differences in spatial and auditory frequency content. Alternatively, these differences might have generated a different engagement from the monkeys in the task for faces and scenes. Yet, another possibility is that the non-human primate brain does not process in exactly the same way the association of social auditory stimuli with facial expressions and with scenes. This will have to be further explored. Overall, therefore, LS appears preferentially sensitive to auditory stimuli whereas the STS appears more responsive to visual than auditory stimuli. In supplemental figure S5B, we show the modulation index of AC versus Vi for both sulci and type of context.”

11. The spatial patterns presented in Figs. 2-4 are very broad, yet locally patchy. It is difficult to assess how solid each of these activations is and thus how much trust to put into these broad patterns. The authors should register each of these activations with a common macaque brain atlas, and identify which brain areas are modulated and report effects in a table.

We have now added three supplemental figures S3, S7 and S11, showing the activation patterns presented in figures 2, 3 and 4 respectively, co-registered with the CIVM macaque atlas (<https://scalablebrainatlas.incf.org/macaque/CBCetal15>, Calabrese et al., 2015). In addition, three supplemental tables (T1, T2 and T3) are produced reporting the size of the effects for the V, AC and AI conditions, for both the face and social contexts.

This material is reproduced below and is now mentioned in the main text as follows: “Supplemental figure S3 represents these activation patterns overlaid with the CIVM non-human primate atlas parcellation; corresponding percentage signal change (%SC) for each area described in supplemental table T1 for the visual, auditory congruent and auditory incongruent vs. fixation contrasts.” And: “Supplemental figure S7 represents the activation patterns of figure 3 overlaid with the CIVM non-human primate atlas parcellation and corresponding percentage signal change (%SC) for each area are described in supplemental table T2 for the visual, auditory congruent and auditory incongruent vs. fixation contrasts.” And: “Supplemental figure S11 represents these activation patterns overlaid with the CIVM non-human primate atlas parcellation; corresponding percentage signal change (%SC) for each area is described in supplemental table T3 for the visual, auditory congruent and auditory incongruent vs. fixation contrasts.”

Supplemental figure S3. All as in Figure 2, with the CIVM atlas (<https://scalablebrainatlas.incf.org/macaque/CBCetal15>, Calabrese et al., 2015) overlaid onto the cortical activations. %SC (median +/-s.e.) in each anatomical ROI described in supplementary table T1.

Supplemental figure S7. All as in Figure 3, with the CIVM atlas (<https://scalablebrainatlas.incf.org/macaque/CBCetal15>, Calabrese et al., 2015) overlaid onto the cortical activations. PSC (median +/-s.e.) in each anatomical ROI described in supplementary table T2.

Supplemental figure S11. All as in Figure 4, with the CIVM atlas (<https://scalablebrainatlas.incf.org/macaque/CBCetal15>, Calabrese et al., 2015) overlaid onto the cortical activations. PSC (median +/-s.e.) in each anatomical ROI described in supplementary table T3.

	Hemisphere	Face context					
		Vi vs Fx		AC vs Fx		AI vs Fx	
		PSC	se	PSC	se	PSC	se
V1	R	0,82	0,25	0,56	0,21	0,49	0,31
	L	1,14	0,23	0,93	0,22	0,15	0,32
V2	R	0,92	0,17	0,95	0,14	0,15	0,18
	L	0,89	0,14	0,77	0,12	-0,03	0,15
V3d	R	0,56	0,09	0,62	0,10	0,06	0,10
	L	0,95	0,13	0,68	0,12	0,08	0,12
V3v	R	0,50	0,30	0,54	0,27	-0,05	0,35
	L	0,46	0,20	0,57	0,17	0,09	0,22
V3A	R	0,71	0,13	0,50	0,10	0,10	0,13
	L	0,80	0,13	0,47	0,12	-0,23	0,13
area 2	R	0,14	0,30	-0,02	0,32	-0,16	0,35
	L	0,20	0,27	0,01	0,30	-0,30	0,33
area 3b	R	0,25	0,11	0,28	0,13	-0,01	0,12
	L	0,19	0,11	0,26	0,12	0,02	0,11
Depth IntraParietal area (DIP)	R	0,30	0,07	0,20	0,07	0,10	0,08
	L	0,14	0,09	0,22	0,08	0,00	0,09
PEa	R	0,23	0,13	0,20	0,10	-0,07	0,12
	L	0,20	0,15	0,32	0,13	-0,12	0,15
POa external part	R	-0,10	0,21	-0,09	0,21	0,10	0,23
	L	0,03	0,22	0,34	0,25	-0,12	0,26
POa internal part	R	0,22	0,09	0,41	0,08	0,09	0,09
	L	0,22	0,10	0,39	0,08	0,01	0,09
V4d	R	1,15	0,28	0,71	0,24	0,09	0,29
	L	1,08	0,26	0,52	0,24	-0,13	0,28
V4v	R	0,73	0,32	0,11	0,32	-0,08	0,38
	L	0,51	0,25	0,38	0,23	0,06	0,29
V4t	R	1,15	0,28	0,71	0,24	0,09	0,29
	L	1,08	0,26	0,52	0,24	-0,13	0,28
Parietal area PG	R	0,09	0,75	-0,20	0,80	-0,40	0,91
	L	0,07	0,72	0,20	0,82	0,11	0,92
Area PG associated region of the STS (PGa)	R	0,72	0,21	0,27	0,21	-0,09	0,21
	L	1,00	0,18	0,55	0,18	0,18	0,18
lateral auditory koniocortex (AKL)	R	0,29	0,23	0,81	0,27	0,42	0,27
	L	0,46	0,33	0,68	0,36	0,33	0,37
medial auditory koniocortex (AKM)	R	0,27	0,17	0,51	0,19	0,30	0,19
	L	0,58	0,35	0,70	0,40	0,13	0,41
fundus of the Superior Temporal Sulcus (f STS)	R	1,02	0,16	0,68	0,17	0,30	0,19
	L	1,04	0,22	0,83	0,26	0,12	0,28
Medial Superior Temporal area (MST)	R	0,43	0,16	0,24	0,15	-0,04	0,19
	L	0,38	0,12	0,33	0,11	-0,06	0,13
Medial Temporal area (v5) (MT)	R	1,09	0,18	0,66	0,18	0,12	0,18
	L	0,75	0,12	0,59	0,15	-0,06	0,13
Retrosular area (Rel)	R	0,33	0,21	0,21	0,22	-0,15	0,21
	L	0,44	0,13	0,32	0,11	0,18	0,11
Paraauditory area caudal part (PaAC)	R	0,21	0,09	0,66	0,11	0,03	0,10
	L	0,40	0,21	0,66	0,22	0,27	0,22
Parainsular cortex lateral part (PaIL)	R	-0,07	0,25	0,46	0,27	0,28	0,28
	L	0,27	0,19	0,50	0,22	0,29	0,18
Prokoniocortex medial part (ProKM)	R	0,47	0,36	0,37	0,39	0,25	0,45
	L	0,57	0,31	0,46	0,37	0,23	0,38
Temporal area TE occipital part (TEO)	R	1,09	0,39	0,64	0,40	0,02	0,46
	L	1,07	0,38	0,62	0,38	0,06	0,40
Temporal area TE occipital medial part (TEOm)	R	0,85	0,39	0,36	0,41	0,15	0,47
	L	0,94	0,30	0,33	0,58	0,00	0,65
TEa	R	1,10	0,15	0,52	0,15	0,11	0,14
	L	1,07	0,14	0,71	0,15	0,14	0,12
Intraparietal sulcus associated area in the STS (IPa)	R	0,77	0,24	0,19	0,24	0,00	0,24
	L	1,16	0,17	0,64	0,15	0,29	0,15
Temporal parietooccipital associated area in STS (TPO)	R	0,78	0,21	0,80	0,20	0,41	0,21
	L	0,86	0,23	0,68	0,25	0,12	0,24
TPO caudal part (TPOC)	R	1,01	0,22	0,71	0,22	0,11	0,23
	L	0,78	0,17	0,69	0,21	0,03	0,17
area 45A	R	0,77	0,12	0,70	0,13	0,00	0,12
	L	0,86	0,10	0,47	0,11	0,21	0,09
area 45B	R	0,72	0,14	0,60	0,11	0,20	0,14
	L	0,68	0,14	0,31	0,12	-0,11	0,13
area 9/46	R	0,70	0,11	0,58	0,12	0,27	0,13
	L	1,26	0,29	0,53	0,28	0,29	0,31
area 47 (old_12) orbital part	R	0,27	0,10	0,28	0,10	0,09	0,10
	L	0,22	0,25	0,10	0,28	-0,07	0,31

Supplementary Table T1: Percentage of signal change (PSC) across all selected atlas ROIs of both hemispheres (median + se), comparing visual, auditory congruent and auditory incongruent conditions vs fixation for the face context (F+ and F- combined). This is a companion table to figure 2 and S3. ROIs are selected based on whether they contain significantly activated voxels as per the analysis presented in figures 2, 3 and/or 4 and are extracted from the CIVM atlas. They are outlined in Supplementary Figure S3. In bold, significant activations relative to fixation baseline, $p < 0.001$ ($p < 0.05$, adjusted for multiple comparisons). Selected ROIs are: V1, V2, V3d, V3v, V3A, area 2, area 3b, Depth IntraParietal area (DIP), Pea, POa external part, POa internal part, V4d, V4v, V4t, Parietal area PG, Area PG associated region of the STS (PGa), lateral auditory koniocortex (AKL), medial auditory koniocortex (AKM), fundus of the Superior Temporal Sulcus (f STS), Medial Superior Temporal area (MST), Medial Temporal area (v5) (MT), Retroinsular area (Rel), Paraauditory area caudal part (PaAC), Parainsular cortex lateral part (PaL), Prokoniocortex medial part (ProKM), Temporal area TE occipital part (TEO), Temporal area TE occipital medial part (TEOm), TEa, Intraparietal sulcus associated area in the STS (IPa), Temporal parietooccipital associated area in STS (TPO), TPO caudal part (TPOC), area 45A, area 45B, area 9/46, area 47 (old_12 orbital part).

Supplementary Table T2: Percentage of signal change (PSC) across all selected atlas ROIs of both hemispheres (median + se), comparing visual, auditory congruent and incongruent conditions vs fixation for F+ and F- contexts separately. This is a companion table to figure 3 and S7. all as in Table T1.

	Hemisphere	Social context					
		Vi vs Fx		AC vs Fx		AI vs Fx	
		PSC	se	PSC	se	PSC	se
V1	R	0,30	0,21	0,39	0,21	0,04	0,23
	L	0,43	0,19	0,35	0,19	0,32	0,19
V2	R	0,57	0,17	0,55	0,16	0,20	0,16
	L	0,61	0,10	0,52	0,10	0,09	0,10
V3d	R	0,53	0,15	0,46	0,17	0,02	0,13
	L	0,68	0,14	0,36	0,09	0,18	0,18
V3v	R	0,56	0,20	0,18	0,22	0,07	0,15
	L	0,71	0,28	0,23	0,18	0,00	0,20
V3A	R	0,53	0,07	0,45	0,08	0,20	0,07
	L	0,68	0,09	0,37	0,10	0,19	0,09
area 2	R	0,10	0,38	-0,14	0,32	0,07	0,26
	L	0,05	0,33	-0,11	0,26	0,05	0,23
area 3b	R	0,16	0,13	-0,01	0,13	-0,02	0,11
	L	0,11	0,17	-0,10	0,16	0,17	0,13
Depth IntraParietal area (DIP)	R	0,08	0,08	0,15	0,08	-0,01	0,08
	L	0,23	0,06	0,12	0,06	0,15	0,05
PEa	R	0,46	0,19	0,16	0,14	-0,06	0,12
	L	0,49	0,31	0,05	0,23	0,11	0,21
POa external part	R	0,18	0,30	0,04	0,22	0,01	0,19
	L	0,29	0,44	-0,12	0,37	0,09	0,32
POa internal part	R	0,36	0,06	0,37	0,07	0,08	0,07
	L	0,35	0,07	0,18	0,07	0,13	0,07
V4d	R	0,73	0,12	0,27	0,12	0,28	0,10
	L	0,93	0,11	0,76	0,11	0,10	0,09
V4v	R	0,64	0,46	-0,15	0,36	-0,08	0,31
	L	0,57	0,46	-0,03	0,36	-0,06	0,32
V4t	R	0,73	0,12	0,27	0,12	0,28	0,10
	L	0,93	0,11	0,76	0,11	0,10	0,09
Parietal area PG	R	-0,01	0,82	-0,15	0,65	-0,21	0,55
	L	0,21	0,88	-0,32	0,70	-0,14	0,60
Area PG associated region of the STS (PGa)	R	0,19	0,08	0,19	0,09	0,10	0,08
	L	0,40	0,06	0,23	0,06	0,08	0,06
lateral auditory koniocortex (AKL)	R	-0,06	0,15	0,22	0,12	0,08	0,15
	L	-0,08	0,16	0,08	0,16	0,28	0,12
medial auditory koniocortex (AKM)	R	-0,03	0,19	-0,03	0,17	0,07	0,14
	L	-0,13	0,17	0,14	0,18	0,17	0,14
fundus of the Superior Temporal Sulcus (f STS)	R	0,51	0,13	0,38	0,11	0,16	0,09
	L	0,57	0,16	0,29	0,16	0,14	0,13
Medial Superior Temporal area (MST)	R	0,27	0,17	0,00	0,13	0,01	0,11
	L	0,53	0,14	0,26	0,11	0,10	0,12
Medial Temporal area (v5) (MT)	R	0,43	0,14	0,33	0,10	0,07	0,10
	L	0,44	0,10	0,32	0,09	0,06	0,08
Retrosular area (Rel)	R	0,24	0,09	0,08	0,10	-0,03	0,09
	L	0,29	0,10	0,16	0,09	0,19	0,07
Paraauditory area caudal part (PaAC)	R	0,03	0,11	0,49	0,10	0,05	0,09
	L	-0,20	0,21	0,23	0,20	0,20	0,15
Parainsular cortex lateral part (PaLL)	R	-0,21	0,18	0,12	0,18	0,03	0,15
	L	0,00	0,12	0,34	0,13	0,24	0,11
Prokoniocortex medial part (ProKM)	R	0,12	0,15	0,08	0,18	0,00	0,13
	L	0,07	0,11	0,12	0,13	0,10	0,09
Temporal area TE occipital part (TEO)	R	0,93	0,33	0,10	0,29	-0,11	0,25
	L	1,05	0,20	0,61	0,18	0,18	0,16
Temporal area TE occipital medial part (TEOm)	R	0,34	0,15	0,17	0,15	0,07	0,11
	L	0,38	0,20	0,30	0,16	0,07	0,14
TEa	R	0,51	0,12	0,29	0,12	0,11	0,11
	L	0,59	0,15	0,34	0,14	0,11	0,12
Intraparietal sulcus associated area in the STS (IPa)	R	0,35	0,12	0,22	0,12	0,03	0,12
	L	0,56	0,13	0,42	0,11	0,04	0,10
Temporal parietooccipital associated area in STS (TPO)	R	0,18	0,10	0,26	0,12	0,15	0,09
	L	0,43	0,13	0,33	0,12	0,16	0,10
TPO caudal part (TPOC)	R	0,58	0,20	0,17	0,17	-0,01	0,14
	L	0,71	0,20	0,36	0,18	0,12	0,16
area 45A	R	0,52	0,19	0,18	0,14	0,09	0,13
	L	0,42	0,19	0,27	0,15	0,21	0,14
area 45B	R	0,57	0,10	0,18	0,09	0,11	0,09
	L	0,48	0,13	0,21	0,11	0,08	0,10
area 9/46	R	0,67	0,32	0,48	0,26	-0,01	0,23
	L	0,74	0,42	0,12	0,35	0,16	0,31
area 47 (old_12) orbital part	R	0,05	0,08	-0,02	0,08	0,11	0,07
	L	0,25	0,20	-0,09	0,20	0,02	0,15

Supplementary Table T3: Percentage of signal change (PSC) across all selected atlas ROIs of both hemispheres (median + se), comparing visual, auditory congruent and incongruent conditions vs fixation for social context (S1+, S1-, S2+, S2- combined). This is a companion table to figure 4 and S11. all as in Table T1.

12. Relatedly, Supplemental Figure 1 shows huge variation of SNR strength even or especially within the parts of the brain for which coil positioning has been optimized. How do they explain this? How does the pattern of results they find correlate with this map? (The authors state that because of the low SNR in occipital cortex, they did not see any activations there. How can we know that this is true? How low was the SNR? Was it really too low to be able to see anything at all?)

The reviewer's comment calls for a clarification of supplemental figure S1 (now figure S4). We have now added a supplementary note that reads as follows and figure S1 has been changed accordingly. These are now called in the main text as follows: "Please note that receiving coils were placed so as to optimize temporal and prefrontal cortex signal-to-noise ratio (SNR). As a result, no activations can be seen in the occipital cortex (see temporal SNR maps in Supplemental figure S4 and precise mean and std signal evaluation in occipital cortex and STS; Please note that in spite of these lower SNR in the occipital cortex, %SC based on an atlas defined ROIs are occasionally significant for the Visual vs. Fixation contrast, and (less so) for the Auditory congruent vs. Fixation contrast, in V1, V2, V3 and V4: see Tables T1, T2 and T3)."

New figure S4 (previously S1) is now reproduced below as well as its legend:

Supplemental figure S4. Temporal signal to noise maps for the pooled face contexts (left) and the pooled social contexts (right). Top: inflated left and right hemispheres. Bottom: flat maps of left and right hemispheres. SNR is calculated by dividing the mean signal by its standard deviation. Low SNR in the occipital cortex is accounted for by both low signal and high signal variability (Face context: left: mean signal: 1847, STD: 230; right: mean signal: 1457, STD: 209; Social context: left: mean signal: 1684, STD: 192; right: mean signal: 1317, STD: 255). This is to be compared to areas of high SNR such as the STS, which is characterized by high signal mean and low signal variability (Face context: left: mean signal: 3003, STD: 80; right: mean signal: 2596, STD: 57; Social context: left: mean signal: 3208, STD: 80; right: mean signal: 2400, STD: 84). Please note that in spite of these lower SNR in the occipital cortex, %SC based on an atlas defined ROIs are occasionally significant for the Visual vs. fixation contrast, and (less so) for the Auditory congruent vs. fixation contrast, in V1, V2, V3 and V4 (see Tables T1, T2 and T3).

13. The description of heart rate variations is confusing. The dominant thing one sees in the figure (which should be improved with complete descriptions of labels) is an interaction effect, while the manuscripts starts talking about a main effect that is much smaller. It certainly goes to show there is a behavioural signature. It would be important to know its timecourse (see comment above) in

comparison to that of the pattern of brain activation. While it is reassuring that the regression of the heart rate did not alter the results, it would still be prudent to consider potential effects on hemodynamic coupling.

Labels in main figure 5 have been improved and a more exhaustive description of the observed statistical effects are now included in the figure legend as well as in the main text. All items are now reproduced below:

Main text: “Heart rate variations depend on semantic congruence with visual context.

In this study, monkeys were required to fixate the centre of the screen while the different auditory and visual stimuli were presented. As a result, it was not possible to analyse whether gaze is spontaneously affected by the different stimulus categories. It was, however, possible to analyse heart-rate variation using a video-based method developed by our team (Froesel et al., 2020). Figure 5 focuses on heart rate variation in response to the auditory sound categories in the different contexts. Heart rate responses, described in figure 5 of Froesel et al. 2020, are typically slow to build up (several seconds). As a result, quantifications of heart rate information were carried out in the second half of the block (last 8 seconds).

We observe a main context effect on heart rate measures (Figure 5A, Friedman non-parametric test, $X_{2(253)} = 437.8$, $p < 0.001$), such that overall heart rate (HR) varies in response to a specific sound, as a function of the type of run being used. Differences in HR are observed between face runs and the two types of social runs, most probably due to the identity of the visual and auditory stimuli, and how they are processed by the monkeys. While this pattern is interesting, we focus here on the observed differences in HR between the positive and negative contexts of runs involving identical stimuli. For the paired contexts (F+/F- and S1+/S1-) both types of sounds (i.e. coos and aggressive calls) are associated with higher heart rate in the positive contexts than in the negative contexts (Wilcoxon paired non-parametric test, aggressive calls between the F+ and F- contexts: $Z = 13.77$, $p < 0.001$, Cohen's d: 16.3 and S1+ and S1-: $Z = 13.82$, $p < 0.001$, Cohen's d: 91.6; Coos between the F+ and F-: $Z = 13.87$, $p < 0.001$, Cohen's d: 47.05, S1+ and S1-: $Z = 13.78$, $p < 0.001$ Cohen's d: 17.42). Screams are also associated with higher heart rate in the positive context than in the negative context (S2+/S2-: Wilcoxon paired non-parametric test, $Z = 13.77$, $p < 0.001$ Cohen's d: 4.01). A reverse effect is observable for coos in the negative context containing screams (S2+/S2-), i.e. heart rate is higher in the negative context than in the positive context (Wilcoxon paired non-parametric test, $Z = 13.78$, $p < 0.001$, Cohen's d: 5.987). Although heart rate measures vary from one context to the other, in all contexts, congruent auditory (Figure 5A, green) is systematically associated with lower heart rates than incongruent auditory (Figure 5A, red, Friedman non-parametric test, Face: $X_{2(253)} = 271.442$, $p < 0.001$; Social 1: , $X_{2(253)} = 295.34$, $p < 0.001$; Social 2: , $X_{2(253)} = 174.66$, $p < 0.001$, Wilcoxon paired non-parametric test. F+: $Z = 13.98$, $p < 0.001$, Cohen's d: 4.5, F-: $Z = 9.77$, $p = 0.012$, Cohen's d: 3.9, S1+: $Z = 13.76$, $p < 0.001$, Cohen's d: 19.7, S1-: $Z = 13.72$, $p < 0.001$, Cohen's d: 18.66, S2+: $Z = 13.82$, $p < 0.001$, Cohen's d: 8.1, S2-: $Z = 13.77$, $p < 0.001$, Cohen's d: 2.92). This effect is more pronounced for the social contexts (S1+/S1- and S2+/S2-) than for the face contexts (Figure 5B, F+/F-, Wilcoxon, $F = 17.45$, $p < 0.001$, Cohen's d: 1.81). This suggests an intrinsic difference between the processing of faces and social scenes. This effect is also more pronounced for contexts involving affiliative visual stimuli (F+, S1+ and S2+) than for contexts involving aggressive or escape visual stimuli (Figure 5b. F-, S1- and S2-, Wilcoxon non-parametric test, $F = 13.20$, $p < 0.001$, Cohen's d: 1.73). This latter interaction possibly reflects an additive effect between the semantics and emotional valence of the stimuli. Indeed, affiliative auditory stimuli are reported to decrease heart rate relative to aggressive or alarm stimuli (Kreibig. 2010). As a result, emotionally positive stimuli would enhance the semantic congruence effect, while emotionally negative stimuli would suppress the semantic congruence effect. Overall, these observations indicate that semantic congruence is perceptually salient, at least implicitly.

Figure 5: Context-related heart rate (BMP) variations. A) Absolute heart rate (BPM, beats per minute) during the congruent (green) and incongruent (red) auditory blocks of each task. Dashed lines correspond to the positive affiliative context (F+, S1+ and S2+) as defined by the visual stimuli, whereas continuous lines refer to the negative aggressive (F- and S1-) or escape contexts (S2-). Contexts are defined by pairs involving the same vocalisation categories but different visual stimuli, as defined in Figure 1b. There is a general context effect on heart rate (Friedman non-parametric test, $X_{2(253)} = 437.8, p < 0.001$). There is a significant difference of HR for a same sound as a function of the context (Wilcoxon paired non-parametric test, aggressive calls between the F+ and F- contexts: $Z = 13.77, p < 0.001$, Cohen's $d = 16.3$ and S1+ and S1-: $Z = 13.82, p < 0.001$, Cohen's $d = 91.6$; Coos between the F+ and F-: $Z = 13.87, p < 0.001$, Cohen's $d = 47.05$, S1+ and S1-: $Z = 13.78, p < 0.001$, Cohen's $d = 17.42$ and S2+ and S2- contexts: $Z = 13.78, p < 0.001$, Cohen's $d = 5.987$ and for screams between S2+ and S2- contexts: $Z = 13.77, p < 0.001$, Cohen's $d = 4.01$). Each context pair shows significantly higher heart rates for incongruent auditory stimuli compared to congruent auditory stimuli (Friedman non-parametric test, Face: $X_{2(253)} = 271.442, p < 0.001$; Social 1: $X_{2(253)} = 295.34, p < 0.001$; Social 2: $X_{2(253)} = 174.66, p < 0.001$). This is also true for each individual context (Wilcoxon paired non-parametric test. F+: $Z = 13.98, p < 0.001$, Cohen's $d = 4.5$, F-: $Z = 9.77, p = 0.012$, Cohen's $d = 3.9$, S1+: $Z = 13.76, p < 0.001$, Cohen's $d = 19.7$, S1-: $Z = 13.72, p < 0.001$, Cohen's $d = 18.66$, S2+: $Z = 13.82, p < 0.001$, Cohen's $d = 8.1$, S2-: $Z = 13.77, p < 0.001$, Cohen's $d = 2.92$). B) Difference between AC and AI bloc medians. All significantly different from zero (Wilcoxon paired non-parametric test. $p < 0.001$ for all contexts except F-: $p < 0.05$, cf A) part). Note that for every item, Cohen's d coefficient is higher than 0.8. Each effect size is therefore considered as large.

Figure 5 of Froesel et al., 2020, referenced in the main text at the beginning of the section on Heart Rate, is reproduced here for the sake of reviewer 2:

Figure 5. EVM HR estimate modulation (mean \pm s.e.) by (A) monkey screams (Wilcoxon test comparing pre-stimulus [-400 -100 ms] and post-stimulus [100 400] epochs, $p=0.03$) and (B) monkey aggressive faces ($p=0.001$).

In addition to using Heart rate as a regressor of non-interest (old figure S4, now figure S13), we now explicitly tested the coupling between heart rate and brain activations, using heart rate as a regressor of interest in the GLM. No activations could be observed. This is now added in the main text as follows : “Importantly, the temporal dynamics of heart rate changes appear to mirror hemodynamic signal modulation in the identified functional network. Because changes in heart rate might affect measured fMRI responses (Chang et al., 2009), we re-ran the analyses presented in figures 2, 3 and 4 using heart rate as a regressor of non-interest in addition to head motion and eye position (Supplemental figure S13). Observed activations remained unchanged, thus indicating that the reported activations are not an artefact of changes in heart rate. In order to further estimate the degree of coupling between heart rate and brain activations, we run a GLM using heart rate as a regressor of interest. No activations could be observed including at uncorrected levels.”

14. This brings me to my last point, the engagement with the past literature. There is single unit data from various brain regions on face voice integration, including consistency effects. The authors should engage with this past literature more closely and more quantitatively. What were the biggest effects reported in this literature and how do they compare to the similarly quantified (see comment above) effects reported here? One recent paper on the matter is Khandhadia et al., which the authors cite, but rather mention than really evaluate. Khandhadia et al. study two specific face areas and find evidence for face voice interactions in one, but not the other. They also have quantifications of effect strength. There should be a deeper consideration of just how surprising the current results are in light of the past literature and maybe a critical reflection on whether the current results reflect an underlying neural truth or other factors as well.

This specific comment is now implemented in the discussion as follows: “Face-voice integration has been described in the auditory cortex (CL, CM, in awake and anaesthetised monkeys; A1 only in awake monkeys) and the STS (Ghazanfar et al., 2008; Perrodin et al., 2014), and to a lesser extent in specific

face-patches (Khandhadia et al., 2021). This latter study is worth noting as their experimental design matched, in important ways, our own, including audio-visual, visual only or auditory only stimuli. They used both monkey movies with a perfect match between visual and auditory stimulation in the audio-visual stimulus and created a computer-generated animated macaque avatar with the explicit intention of having synchronisation between the vocalisation and facial movements of the avatar. The study was thus explicitly testing multisensory integration under the hypothesis that the visual and auditory stimuli were associated with a common source. The audio-visual stimuli thus achieved a double congruence: they were temporally synchronised such that facial movements predicted vocalisations and as a consequence, they matched in semantic content. In the present study, our aim was to study the second type of congruence, i.e. semantic congruence. Our audio-visual stimuli were therefore not synchronised, but the two stimuli, when presented at the same time could be congruent (or incongruent) in semantic terms. The face-voice or scene-voice multisensory integration described by Khandhadia et al. is of a different nature to the one we report here. More specifically, **in the present data**, enhancement of the audio-visual response can only be seen in the contexts involving visual scenes. The parsimonious interpretation of these observations is that face-vocalisation binding was easier than scene-vocalisation binding and resulted in signal saturation, in agreement with the fact that neuronal multisensory integration is more pronounced for low saliency stimuli. The most significant difference between our study and that of Khandhadia et al. **pertains to the second order congruency, an issue we discuss next.**

15. Similarly, I do think, despite the authors' response, that a comparison to the predictive coding account would be important for the discussion, again highlighting that the current results are the opposite of that account and discussing why this might be.

We have now included an independent discussion paragraph on predictive coding that reads as follows:

"It is worth noting that our results go against the predictive coding theory. This theory posits that the brain is constantly generating and updating an internal model of the environment. This model then generates predictions of sensory input and compares these to actual sensory input (Friston, 2010; Rao & Ballard, 1999). Prediction errors are then used to update and revise the internal model (Millidge et al., 2022). In the context of predictive coding, when viewing an affiliative face, monkeys are expected to predict affiliative vocalisations. As a result, aggressive vocalisations in the context of affiliative faces are expected to generate prediction errors and hence higher activations than those observed for the affiliative vocalisations. This is not what our data show: when, viewing affiliative faces, there are enhanced responses to affiliative vocalisation and suppressed responses to aggressive vocalisations. This effect actually builds up as visual contextual information is reinforced through the run and is present in both the STS and the LS, i.e. at the early stages of auditory processing. Thus, these observations are inconsistent with the predictive coding experimental predictions. They suggest that the monkeys implement an active matching or association between the visual and the auditory social information, similar to a match to sample task, based on their life long social experiences. In match to sample fMRI and EEG studies in humans (Druzgal & D'Esposito, 2001) and electrophysiology studies in non-human primates (Miller & Desimone, 1994; Suzuki et al., 1997), responses to the probe matching the sample is significantly higher than the response to a non-match probe, thus describing a match enhancement (Suzuki & Eichenbaum, 2000). This is very similar to what we describe here, if considering the visual context as the probe and the auditory stimuli as the match and non-match probes. Further work is required to confirm this hypothesis."

16. Lastly, the abstract in the current manuscript is highly descriptive of detail and fails to highlight conceptual relevance.

Social interactions rely on the interpretation of semantic and emotional information, often from multiple sensory modalities. Nonhuman primates send and receive auditory and visual communicative signals. However, the neural mechanisms underlying the association of visual and auditory information based on their common social meaning are unknown. Using heart rate estimates and functional neuroimaging, we show that in the lateral and superior temporal sulcus of the macaque monkey, neural responses are enhanced in response to species-specific vocalisations paired with a matching visual context, or when vocalisations follow, in time, visual information, but inhibited when vocalisation are incongruent with the visual context. For example, responses to affiliative vocalisations are enhanced when paired with affiliative contexts but inhibited when paired with aggressive or escape contexts. Overall, we propose that the identified neural network represents social meaning irrespective of sensory modality. This network may represent a precursor to our amodal linguistic representations..”

Reviewer #3 (Remarks to the Author):

The authors have responded comprehensively and adequately to my critique and comments. I feel the manuscript is further improved by these changes and by the authors' response to the other reviewers. From my point of view, I don't see any reason to deny this important and novel work immediate publication. The term 'heroic' that I've used in my original review applies even more to the present version.

We again would like to thank reviewer 3 for his/her appreciation of our work.

References to Response to reviewers

- Brett, M., Anton, J.-L., Valabregue, R., & Poline, J.-B. (s. d.). *Region of interest analysis using an SPM toolbox*. 1.
- Calabrese, E., Badea, A., Coe, C. L., Lubach, G. R., Shi, Y., Styner, M. A., & Johnson, G. A. (2015). A diffusion tensor MRI atlas of the postmortem rhesus macaque brain. *NeuroImage*, *117*, 408-416. <https://doi.org/10.1016/j.neuroimage.2015.05.072>
- Calvert, G. A. (2001). Crossmodal Processing in the Human Brain: Insights from Functional Neuroimaging Studies. *Cerebral Cortex*, *11*(12), 1110-1123. <https://doi.org/10.1093/cercor/11.12.1110>
- Druzgal, T. J., & D'Esposito, M. (2001). A neural network reflecting decisions about human faces. *Neuron*, *32*(5), 947-955. [https://doi.org/10.1016/s0896-6273\(01\)00519-0](https://doi.org/10.1016/s0896-6273(01)00519-0)
- Friston, K. (2010). The free-energy principle : A unified brain theory? *Nature Reviews Neuroscience*, *11*(2), 127-138. <https://doi.org/10.1038/nrn2787>
- Froesel, M., Goudard, Q., Hauser, M., Gacoin, M., & Ben Hamed, S. (2020). Automated video-based heart rate tracking for the anesthetized and behaving monkey. *Scientific Reports*, *10*(1), 17940. <https://doi.org/10.1038/s41598-020-74954-5>
- Ghazanfar, A. A., Chandrasekaran, C., & Logothetis, N. K. (2008). Interactions between the Superior Temporal Sulcus and Auditory Cortex Mediate Dynamic Face/Voice Integration in Rhesus Monkeys. *Journal of Neuroscience*, *28*(17), 4457-4469. <https://doi.org/10.1523/JNEUROSCI.0541-08.2008>
- Ghazanfar, A. A., Maier, J. X., Hoffman, K. L., & Logothetis, N. K. (2005). Multisensory Integration of Dynamic Faces and Voices in Rhesus Monkey Auditory Cortex. *Journal of Neuroscience*, *25*(20), 5004-5012. <https://doi.org/10.1523/JNEUROSCI.0799-05.2005>

- Khandhadia, A. P., Murphy, A. P., Romanski, L. M., Bizley, J. K., & Leopold, D. A. (2021). Audiovisual integration in macaque face patch neurons. *Current Biology*. <https://doi.org/10.1016/j.cub.2021.01.102>
- Kreibig, S. D. (2010). Autonomic nervous system activity in emotion : A review. *Biological Psychology*, *84*(3), 394-421. <https://doi.org/10.1016/j.biopsycho.2010.03.010>
- Miller, E. K., & Desimone, R. (1994). Parallel neuronal mechanisms for short-term memory. *Science (New York, N.Y.)*, *263*(5146), 520-522. <https://doi.org/10.1126/science.8290960>
- Millidge, B., Seth, A., & Buckley, C. L. (2022). Predictive Coding : A Theoretical and Experimental Review. *arXiv:2107.12979 [cs, q-bio]*. <http://arxiv.org/abs/2107.12979>
- Perrodin, C., Kayser, C., Logothetis, N. K., & Petkov, C. I. (2014). Auditory and visual modulation of temporal lobe neurons in voice-sensitive and association cortices. *The Journal of Neuroscience: The Official Journal of the Society for Neuroscience*, *34*(7), 2524-2537. <https://doi.org/10.1523/JNEUROSCI.2805-13.2014>
- Rao, R. P. N., & Ballard, D. H. (1999). Predictive coding in the visual cortex : A functional interpretation of some extra-classical receptive-field effects. *Nature Neuroscience*, *2*(1), 79-87. <https://doi.org/10.1038/4580>
- Seltzer, B., & Pandya, D. N. (1994). Parietal, temporal, and occipital projections to cortex of the superior temporal sulcus in the rhesus monkey : A retrograde tracer study. *Journal of Comparative Neurology*, *343*(3), 445-463. <https://doi.org/10.1002/cne.903430308>
- Suzuki, W. A., & Eichenbaum, H. (2000). The neurophysiology of memory. *Annals of the New York Academy of Sciences*, *911*, 175-191. <https://doi.org/10.1111/j.1749-6632.2000.tb06726.x>
- Suzuki, W. A., Miller, E. K., & Desimone, R. (1997). Object and Place Memory in the Macaque Entorhinal Cortex. *Journal of Neurophysiology*, *78*(2), 1062-1081. <https://doi.org/10.1152/jn.1997.78.2.1062>

REVIEWER COMMENTS

Reviewer #1 (Remarks to the Author):

Exactly as stated by the authors, the manuscript improved considerably after this thorough and detailed revision, but the main message remains unaffected.

In my view this represents an exceedingly important message for everyone who is interested in visual and auditory cortex/processing. The results are thought-provoking and will trigger many follow-up studies. Especially interesting would be to perform the reverse associations, to use event related designs and to study what happens at neuronal level. I'm also convinced that the predictive coding theory requires updating, given this result.

I highly recommend publication and commend the authors again for their great work!

Reviewer #2 (Remarks to the Author):

I thank the authors for the additional work that they have performed following my requests. I am sorry having to report though that my concerns about the manuscript have, overall, increased. I assume the other two reviewers will remain their very positive attitude towards the manuscript and, at this point, will probably be as unhappy with me as the authors must be.

The main reason my concerns have increased is the fact that after registration of the functional activation maps to a standard atlas, the authors are now reporting auditory activation in visual cortical areas such as V2 and MT. If I had a hard time understanding how the STS results came about, I am now even more perplexed. How a paradigm like the current one would exert such long term, cross-modal yet specific effects in these areas at a significance level fully comparable or even exceeding those in known auditory areas, is impossible for me to understand. I did not find any explanation for this in rebuttal or new manuscript. In order to believe such results, we would need more than the broad activation maps of the manuscript.

It is for this reason that I had asked for a functional localizer of face areas. I commend the authors for attempting to do this. However, it did not work out. A face localizer requires a face and a non-face visual condition to be contrasted, but only the former one was measured. This results, expectedly, in yet another broad activation map from which no claim of face specificity beyond visual responsiveness can be derived. This made it impossible for the authors to reproduce a now rather extensive literature on face areas. Thus we do not gain new insights from the analysis. In particular, it is now impossible to see effects in areas with known functional properties and compare results across areas of different functional specialization.

Success here would have offered an opportunity to engage, both qualitatively and quantitatively, with single-unit physiology results published recently by Khandhadia et al. in two STS face areas, only one of which showed auditory effects at all, and the other ones rather modest ones. Thus in more than one way it is impossible to put these results into context.

These new concerns exist on top of older ones I have had and continue to have, namely that, given the specifics of the paradigm, in particular the long time scales, the pattern of activation combined with incredibly strong effects, are entirely unexpected. I would need more conformation to believe that these are not methodological side effects of fMRI or other

methods. A quantitative comparison with Khandhadia would have really helped here. That said, maybe this is the paper to overhaul decades of systems neuroscience results offering a revolutionary new perspective on brain function I am too conservative to see. Thus short of being able to pinpoint the source of a potential mistake, I do not wish to stand in the way of the manuscript getting published and do not want to further delay publication.

Should this be the editorial decision, the only thing I would advise the authors to do, is to reconsider their choice of main and supplemental figures. In my view several important analyses are relegated to supplemental material, while several main figures show very similar findings are thus less important.

Reviewer #3 (Remarks to the Author):

I reiterate my initial assessment that this could easily become a landmark paper in the fields of visual-auditory integration, social perception, and evolution of communication systems. The amount of analyses in response to the reviewers is admirable.

We would like to thank the reviewers for their appreciation of our work and their final assessment. By taking into account this feedback, our revised manuscript has been considerably improved. We are now addressing the final concerns below, according to the specific directions of the editor. We have additionally edited our manuscript to comply to the policies and formatting requirements of Nature Communications.

Reviewer #1 (Remarks to the Author):

Exactly as stated by the authors, the manuscript improved considerably after this thorough and detailed revision, but the main message remains unaffected.

In my view this represents an exceedingly important message for everyone who is interested in visual and auditory cortex/processing. The results are thought-provoking and will trigger many follow-up studies. Especially interesting would be to perform the reverse associations, to use event related designs and to study what happens at neuronal level. I'm also convinced that the predictive coding theory requires updating, given this result.

I highly recommend publication and commend the authors again for their great work!

We would like to thank reviewer 1 for this enthusiasm on our work and we absolutely agree with her/him on the multiple follow up experiments that are triggered by this work.

Reviewer #2 (Remarks to the Author):

I thank the authors for the additional work that they have performed following my requests. I am sorry having to report though that my concerns about the manuscript have, overall, increased. I assume the other two reviewers will remain their very positive attitude towards the manuscript and, at this point, will probably be as unhappy with me as the authors must be.

The main reason my concerns have increased is the fact that after registration of the functional activation maps to a standard atlas, the authors are now reporting auditory activation in visual cortical areas such as V2 and MT. If I had a hard time understanding how the STS results came about, I am now even more perplexed. How a paradigm like the current one would exert such long term, cross-modal yet specific effects in these areas at a significance level fully comparable or even exceeding those in known auditory areas, is impossible for me to understand. I did not find any explanation for this in rebuttal or new manuscript. In order to believe such results, we would need more than the broad activation maps of the manuscript.

The auditory activation in areas V2 and MST are clearly identifiable in figures 2, 3 and 4. These figures have not changed since the initial submission. As requested by reviewer 2, quantification of corresponding percentage signal change relative to fixation and associated statistical significance is described in supplemental tables T1, T2 and T3 for the face task (T1), F+ and F- face contexts separately (T2) and the social task (T3). In these tables, as requested by reviewer 2 in his previous comments, %SC are computed not over the specific significantly modulated voxels identified using the GLM described in figures 2, 3 and 4. Rather, they are computed over entire functional areas as defined using the CIVM atlas. This quantification supports the maps presented in figures 2, 3 and 4.

In the face context, visual %SC are significant in visual areas V1, C2, V3d, V3v, V3a, V4d, V4v, V4t and MT (n=11/18, where 18 corresponds to the number of considered visual areas, on the left and on the right hemisphere). Congruent auditory %SC are significant in V2, V4d and MT (n=4/18). Incongruent auditory %SC are never significant. In the face positive context, 10/18 of the visual areas have a significant %SC relative to fixation in response to visual blocks, and 8/18 of them have a significant %SC relative to fixation in response to congruent auditory stimuli. In the negative face context, 13/18 of the visual areas have a significant %SC relative to fixation in response to visual blocks, and 3/18 of them have a significant %SC relative to fixation in response to congruent auditory stimuli. In the social context, 11/18 of the visual areas have a significant %SC relative to fixation in response to visual blocks, and 7/18 of them have a significant %SC relative to fixation in response to congruent auditory stimuli. This is in line with the observation that the entire network activated by a given set of social visual stimuli (composed of prefrontal, temporal and striate and extrastriate visual areas) is also activated by the congruent social auditory stimuli.

The above paragraph is now added as is, as a supplemental note to supplemental tables T1, T2 and T3.

This supplemental note is now called in the main text as follows:

“Taken together, these results indicate that audio-visual semantic associations are implemented in a specific cortical network involved in the processing of both visual face and social stimuli as well as voice stimuli. This network is composed of prefrontal and temporal areas, but also, of visual striate and extrastriate visual areas (see supplemental note attached to supplementary tables T1, T2 and T3. An important question is thus whether these neuronal computations impact the behaviour or the physiology of the monkeys. In the following section, we investigate how heart rate changes in response to auditory-visual stimuli that are either congruent or incongruent with the social situation.”

It is for this reason that I had asked for a functional localizer of face areas. I commend the authors for attempting to do this. However, it did not work out. A face localizer requires a face and a non-face visual condition to be contrasted, but only the former one was measured. This results, expectedly, in yet another broad activation map from which no claim of face specificity beyond visual responsiveness can be derived. This made it impossible for the authors to reproduce a now rather extensive literature on face areas. Thus we do not gain new insights from the analysis. In particular, it is now impossible to see effects in areas with known functional properties and compare results across areas of different functional specialization. Success here would have offered an opportunity to engage, both qualitatively and quantitatively, with single-unit physiology results published recently by Khandhadia et al. in two STS face areas, only one of which showed auditory effects at all, and the other ones rather modest ones. Thus in more than one way it is impossible to put these results into context.

As already discussed in previous revision, this was not the scope of the experiment. Although the question raised by this reviewer is very relevant, we believe that our results bring something new to the literature. This is shared by the two other reviewers. Our additional independent control allows to test for possible double dipping in data analysis. The specific question raised by the reviewer 2 can be addressed by us or other in subsequent experiments, and obviously reproducing our observations using dual fMRI-ophys experiments would be THE thing to do. This is in itself a five-year project.

These new concerns exist on top of older ones I have had and continue to have, namely that, given the specifics of the paradigm, in particular the long time scales, the pattern of activation combined with incredibly strong effects, are entirely unexpected. I would need more conformation to believe that these are not methodological side effects of fMRI or other methods. A quantitative comparison with Khandhadia would have really helped here.

We agree with reviewer 2 that this would be a very interesting follow up study.

That said, maybe this is the paper to overhaul decades of systems neuroscience results offering a revolutionary new perspective on brain function I am too conservative to see. Thus short of being able to pinpoint the source of a potential mistake, I do not wish to stand in the way of the manuscript getting published and do not want to further delay publication. Should this be the editorial decision, the only thing I would advise the authors to do, is to reconsider their choice of main and supplemental figures. In my view several important analyses are relegated to supplemental material, while several main figures showing very similar findings are thus less important.

Reviewer 2 states that *several main figures show very similar findings [and] are thus less important*. The alternative view is that these figures indicate that our findings are reproducible over independent datasets. In the general current discussion on reproducibility of scientific results in neurosciences, this appears to us as an important point.

This being said, we however concur with reviewer 2 with the fact that some of the supplemental figures are also very important. In order to reconcile the suggestion of reviewer 2 with the editorial constraints by Nature Communication in terms of maximum number of figures, we have decided to include as a main figure previous supplemental figure S12. This figure captures two main observations that consolidate our findings: 1) the fact that context effects get stronger and stronger across run repetition and 2) the fact that increased distance to the last visual from block the current auditory block does not result in decreased context effects.

All main and supplemental figures have been renumbered accordingly.

Reviewer #3 (Remarks to the Author):

I reiterate my initial assessment that this could easily become a landmark paper in the fields of visual-auditory integration, social perception, and evolution of communication systems. The amount of analyses in response to the reviewers is admirable.

We would like to thank reviewer 3 for her/his appreciation of our work.